# Candida auris skin tropism and antifungal resistance are mediated by carbonic anhydrase Nce103

Trinh Phan-Canh [1,2,3], Cristina Coman[4], Michaela Lackner [5], Nina Troppmair[4,6], Christoph Müller [7], Diana Cerbu[3], Saskia Seiser[3], Philipp Penninger[1,2], Irina Tsymala[1,2], Narakorn Khunweeraphong[1,2,3], Tamires Bitencourt [1,5], Andrej Knarr[1,2], Sabrina Jenull[1,2], Hossein Arzani [1,2], Lisa-Maria Zenz[5], Giuseppe Ianiri [8], Weiqiang Chen[9], Anuradha Chowdhary [10], Harry L. T. Mobley[11], Markus Hartl [9], Doris Moser [12], Robert Ahrends[4], Adelheid Elbe-Bürger [3] ✉ & Karl Kuchler [1,2] ✉

The pronounced skin tropism and pan-antifungal resistance of *Candida auris* pose a serious global health threat. A key question in *C. auris* biology is how clinical isolates acquire amphotericin B resistance. Here we demonstrate that a carbonic sensing pathway (CSP) contributes to amphotericin B resistance by modulating mitochondrial energy functions in clinical *C. auris* isolates. Integrated transcriptomics and proteomics identify the carbonic anhydrase Nce103 and its transcription factors Rca1 and Efg1 as important regulatory components of the CSP. The conversion of $CO_2$ into bicarbonate sustains energy metabolism required for colonization and fitness on human skin and in nutrient-limited microenvironments. We also show that bacterial skin colonizers engage urease to release $CO_2$ that sustains *C. auris* fitness and skin colonization. These findings highlight therapeutic options to re-sensitize *C. auris* to antifungal treatments, as well as to prevent skin colonization by blocking the CSP.

*Candida auris* (*Candidozyma auris*) is an emerging human fungal pathogen causing disseminated infections of high mortality (30–72%) in individuals with underlying diseases or impaired immunity[1–3]. Since 2009, *C. auris* has spread to more than 50 countries, causing outbreaks in intensive care units and nursing homes[4,5]. Most importantly, *C. auris* is the first fungal pathogen showing untreatable pan-antifungal resistance traits to all clinically used antifungal entities, including azoles (up to 90%), amphotericin B (AMB, 30–60%), echinocandins (up to 8%) and flucytosine[6–8]. Hence, the high frequency of multidrug resistance (MDR)[6,7] in clinical isolates poses a serious challenge to conventional therapy. Although echinocandins are first-line therapy[9], their limited bioavailability in the urinary tract and central nervous system and acquired antifungal resistance traits often lead to therapeutic failure in relapsed invasive candidiasis[1]. In such cases, AMB has to be used as alternative treatment[1,9]. However, the increasing AMB resistance (AMB[R]) is another serious concern for *C. auris* infections.

[1]Max Perutz Labs Vienna, Vienna, Austria. [2]Center for Medical Biochemistry, Medical University of Vienna, Vienna, Austria. [3]Department of Dermatology, Medical University of Vienna, Vienna, Austria. [4]Department of Analytical Chemistry, University of Vienna, Vienna, Austria. [5]Institute of Hygiene and Medical Microbiology, Medical University of Innsbruck, Innsbruck, Austria. [6]Vienna Doctoral School in Chemistry, University of Vienna, Vienna, Austria. [7]Department of Pharmacy, Center for Drug Research, Ludwig-Maximilians-Universität München, Munich, Germany. [8]Department of Agricultural, Environmental and Food Sciences, University of Molise, Campobasso, Italy. [9]Mass Spectrometry Facility, Max Perutz Labs Vienna, Vienna, Austria. [10]Medical Mycology Unit, Department of Microbiology, Vallabhbhai Patel Chest Institute, University of Delhi, Delhi, India. [11]Department of Microbiology & Immunology, University of Michigan Medical School, Ann Arbor, MI, USA. [12]Department of Oral and Maxillofacial Surgery, Medical University of Vienna, Vienna, Austria. ✉e-mail: adelheid.elbe-buerger@meduniwien.ac.at; kuchlerkarl1@gmail.com

It is worth noting that molecular mechanisms underlying the high rate of AMB[R] in *C. auris* remain largely enigmatic[10,11]. AMB causes membrane leakage through ergosterol sequestration, thereby leading to a fungicidal disruption of the electrochemical gradient. The rapid induction of oxidative stress is another potential mode of action[12]. Resistance to AMB is very rare in yeast species due to highly deleterious fitness costs[13]. Most AMB[R] *C. auris* clinical isolates do not exhibit substantially impaired fitness and, in some cases, even acquire enhanced growth[12]. Rare mutations in the ergosterol pathway[10,14] gained during treatment promote AMB[R]. For instance, the *ERG6*[YY98V*] truncation confers resistance but also causes a high fitness loss[14]. Recently, an experimental evolution approach indicates that mutations in *ERG6*, *ERG10*, *ERG11*, *NCP1* and *HMG1* lead to acquired AMB[R] in *C. auris*[10]. However, the relevant mutations are exceedingly rare in clinical *C. auris* isolates, suggesting that elevated AMB[R] observed in most clinical cases may not primarily result from mutations in these genes but rather arise from synthetic genetic interaction with as yet unknown AMB[R] modifier genes.

In this Article, we used an advanced integrated proteotranscriptomics approach comparing phenotypes of drug-resistant and sensitive clinical isolates to shed light on potential mechanisms contributing to elevated AMB[R] in *C. auris*. The validation and reverse genetics show a critical function of a carbon dioxide sensing pathway (CSP) in establishing both fungal fitness for skin colonization and antifungal resistance. We also demonstrate that hospital-acquired bacterial pathogens such as *Proteus mirabilis* and *Klebsiella pneumoniae* can enhance *C. auris* fitness by urease-mediated release of carbon dioxide that is used by the fungal CSP to sustain energy metabolism and growth.

## Results

### Proteomics reveals multidrug resistance mechanisms in *C. auris*

Transcriptomics of MDR strains versus drug-sensitive clinical isolates revealed numerous genes implicated in carboxylic acid metabolism, mitochondrial function, translation and membrane transports as critical hallmarks of antifungal resistance traits in *C. auris*[15–17]. However, major effectors or sensing mechanisms implicated in antifungal resistance escaped discovery[15,16]. Hence, we hypothesized that proteomics may identify key pathways involved in *C. auris* MDR traits from differentially abundant proteins (DAPs). We subjected cell-free extracts from resistant (R) (462/P/14-R1) and sensitive (S) (2431/P/16–S) *C. auris* strains[15,17] (Fig. 1a,b) to shotgun proteomics. The data were then integrated with available RNA-sequencing (RNA-seq) datasets of strains with partially overlapping antifungal susceptibility profiles (R1, 1133/P/13R–R2 and S). As expected, several membrane transporter families were highly enriched in resistant *C. auris* strains (Fig. 1c and Extended Data Fig. 1a,b), including the Cdr1 multidrug efflux ATP-binding cassette (ABC) transporter, the Mdr1 major facilitator and the putative phosphatidylinositol transfer protein Pdr16 (ref. 18), all of which cause clinical azole resistance[16,19,20]. In addition, the isolates R1 and R2, but not the S strain, carried the mutational hotspot S639F in the Fks1 glucan synthase (Extended Data Fig. 1c)[15,21], thus explaining pronounced echinocandin resistance traits in R1 and R2. The overlay of proteo-transcriptomics datasets (Extended Data Fig. 1d–f) revealed numerous proteins, of which we chose several high-abundance factors for further validation, but at least ablation of *PGA31.2*, *PHR1* and *SMF12* did not change in minimal inhibitory concentrations (MICs) for all four antifungal classes (Extended Data Fig. 2a).

*NCE103* (B9J08_000363) encoding a carbonic anhydrase that converts $CO_2$ into bicarbonate ($HCO_3^-$) was the most abundant protein in R1 relative to S (Fig. 1d). In addition, the transcription factor Rca1, a key regulator of *NCE103*, was also enriched in strain R1 (Fig. 1d). Gene Ontology term analysis revealed the enrichment of carboxylic acid metabolism and cell redox homeostasis in R1 (Fig. 1e), implicating a potential link between drug resistance and mitochondrial function, the main cellular location of carboxylic acid metabolism[22]. In other fungal species, bicarbonate plays critical roles in energy metabolism and

morphogenesis, as it is acting through the cyclic adenosine monophosphate–dependent protein kinase A (cAMP/PKA) signaling pathway that converges at the regulator Efg1 (refs. 23–25). Taking these together, we hypothesized that the Rca1–Nce103–Efg1 axis is vital to control antifungal susceptibility (Fig. 1f).

### Rca1 and Efg1 controls AMB susceptibility across clinical isolates

Orthologues of the carbonic anhydrase basic leucine zipper (bZIP) regulator Rca1 and the helix–turn–helix / asexual–phases–specific (HTH-APSES)-type transcription factor Efg1 are only present in *Candida* spp. and *Saccharomyces* spp. genomes but not in *Aspergillus fumigatus* (Extended Data Fig. 2b,c). Thus, we first deleted *RCA1* and *EFG1* in different *C. auris* clinical isolates showing elevated MICs of ≥1 µg ml⁻¹ for AMB (Fig. 2a and Extended Data Fig. 3a). Indeed, loss of *RCA1* caused 2-fold and 4-fold reductions of the AMB[MIC] in a strain from clade III (AR384) and in different MDR strains from clade I, respectively. Deletion of *EFG1* resulted in a 2-fold reduction of the MIC for AMB in both clade I and clade III strains. Surprisingly, *efg1Δ* and the *rca1Δefg1Δ* double mutant showed a 2-fold increase in MICs compared to the *rca1Δ* deletion (Fig. 2a). This may result from rewiring multiple downstream pathways influencing AMB tolerance or to the flocculation phenotype of *efg1Δ* often observed in clinical isolates (Extended Data Fig. 3b)[26]. Moreover, both *rca1Δ* and *efg1Δ* mutants showed slightly increased susceptibility to 5-fluorocytosine, although not significant to caspofungin or voriconazole (Extended Data Fig. 3c). To further verify the role of the CSP, MIC experiments were repeated at high and low $CO_2$ to simulate the varying $CO_2$ levels on the skin and within deeper tissues, respectively. Remarkably, the *rca1Δ*, *efg1Δ*, and the sensitive strain S entirely restored AMB[R] in the presence of 5.5% $CO_2$ (Fig. 2a and Extended Data Fig. 3a) relative to the AMB[S] in ambient air (~0.04% $CO_2$). Consequently, both Rca1 and Efg1 modulate AMB susceptibility through carbonic sensing. The simultaneous deletion of *RCA1* and *EFG1* showed their epistatic relationship.

AMB susceptibility and $CO_2$-sensing phenotypes were similar in strain S and *rca1Δ* mutants (Fig. 2a), suggesting genetic mutations in *RCA1*. Indeed, using IGV tool (version 2.16.2)[27], we discovered a single non-synonymous nucleotide polymorphism *RCA1*, yielding a truncated Rca1[Y110*] variant lacking a the essential bZIP region. In addition, the Rca1[Y110*] variant was non-functional and unstable, as proteomics failed to detect any Rca1-derived peptides from strain S (Fig. 2b). Overall, the data indicated that existing loss-of-function mutation in *RCA1* resensitized *C. auris* clinical isolates by affecting Nce103 expression.

Fungi sense $CO_2$ by generating bicarbonate ions[28–30] or directly through the type 2C-related protein phosphatase Ptc2[31], with a single orthologue B9J08_003504 in *C. auris*[32]. However, deletion of *PTC2* in the R1 strain or in the *rca1Δ* mutant did not result in any changes of the AMB[MIC], excluding a role for Ptc2 in AMB susceptibility (Fig. 2a). Therefore, Rca1 and Efg1 most likely engage carbonic anhydrase for responding to $CO_2$-derived bicarbonate anions[30], which is supported by data showing that mutants lacking *RCA1* and *EFG1* exhibited hypersensitivity to acetazolamide, a specific Nce103 inhibitor (Extended Data Fig. 3d–f).

### The carbonic anhydrase Nce103 governs AMB susceptibility

The carbonic anhydrase gene *NCE103* is conserved across many fungal species including *C. auris* (Fig. 2c and Extended Data Fig. 3e,f). Nce103 was highly expressed in strain R1 at both transcriptional and protein level (Fig. 2b,d). Despite multiple attempts, we failed to delete *NCE103* even when supplying 5.5% $CO_2$ for selection. However, the CRISPR–Cas9 (clustered regularly interspaced short palindromic repeats and CRISPR-associated protein 9) method[33] allowed for constructing the *nce103Δ* deletion when supplementing with 10% $CO_2$. As Nce103 appeared as an essential protein for fitness in ambient air, its lack caused severe fitness defects, but growth gradually resumed upon $CO_2$ supplementation (Fig. 2a). The *nce103Δ* mutant exhibited the same AMB[MIC] values as the *rca1Δ* strain in 1.5% $CO_2$. The *rca1Δnce103Δ* double mutant

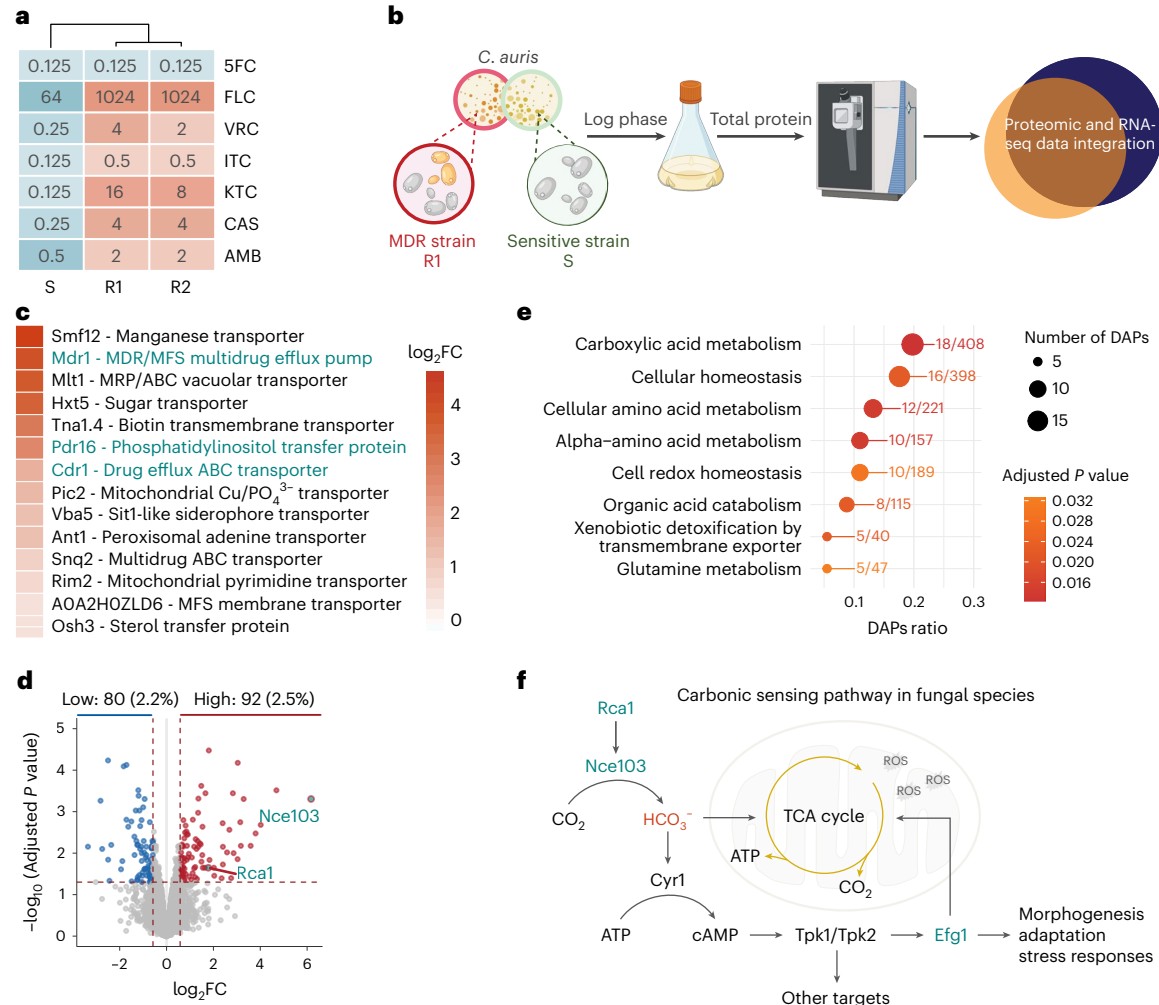

**Fig. 1 | Proteomics unravels multidrug resistance mechanisms in *C. auris*.**
**a**, Heat map showing MIC ($\mu$g ml$^{-1}$) values for multidrug-resistant (R1, R2) and drug-sensitive (S) clinical strains following the CLSI method. 5FC, 5-fluorocytosine; FLC, fluconazole; VRC, voriconazole; ITC, itraconazole; KTC, ketoconazole; CAS, caspofungin; AMB. **b**, Proteomic workflow and data analysis. **c**, Heat map shows top transporters that are of high abundance in strain R1. FC, fold change. **d**, Volcano plot of proteomics data shows DAPs between R1 and S strains. Statistical analyses were performed using moderated two-sided *t*-statistics with the limma-trend method, followed by multiple test correction using the Benjamini–Hochberg procedure. Differential expression analysis filters proteins with log$_2$ fold change (log$_2$FC) $\pm$ 0.58 and adjusted $P$ < 0.05 indicated as dashed line. **e**, Gene Ontology term enrichment analysis for DAPs between R1 and S strains. The dot plot presents data from Gene Ontology enrichment analysis using *clusterProfiler* with $P$ value calculated by hypergeometric distribution and Benjamini–Hochberg for multiple test correction. **f**, Scheme reveals a CSP controlling a link between metabolism and morphogenesis in fungal species. Panel **b** created with BioRender.com. Panel **f** adapted with permission from ref. 54, Springer Nature Limited.

did not show exacerbated susceptibility, while supplementing with 5.5% CO$_2$ fully restored AMB$^R$ in all strains (Fig. 2a). Therefore, Rca1 and Nce103 are epistatic and act in the same CSP.

Supplementing with 5.5% CO$_2$ enhanced AMB$^R$ in several strains (Fig. 2e). Therefore, we reasoned that messenger RNA expression or gene dosage of *NCE103* might affect AMB$^R$ traits. Indeed, a subset of clade I AMB-resistant strains exhibited high *NCE103* expression (Fig. 2f), and the correlation between *NCE103* mRNA levels and AMB$^R$ was modest but statistically significant (Pearson's $R^2$ = 0.19, $P$ = 0.045). Similarly, ectopically overexpressing *NCE103* (e*NCE103*) in multiple backgrounds fully restored AMB$^{MIC}$ to R1's MIC, although mRNA levels of e*NCE*103 were higher than in the R1 strain (Extended Data Fig. 3g).

The RNA-seq data hinted a strong 6-fold and 9-fold *NCE103* downregulation in *rca1$\Delta$* and *efg1$\Delta$* mutants, respectively (Fig. 2d). To answer how *RCA1* and *EFG1* regulate *NCE103*, we examined *NCE103* mRNA levels in *C. auris* cultured under low and high CO$_2$ conditions. Supplementing with 5.5% CO$_2$ resulted in a modest reduction in *NCE103* expression in the R1 strain (Fig. 2g). By contrast, deletion of *RCA1* and/or *EFG1* markedly repressed *NCE103*, showing a 4- to 9-fold

decrease under both CO$_2$ conditions. The proteomics data showed that Nce103 abundance under AMB stress was higher in strain S but failed to reach the levels in R1 (Extended Data Fig. 1f). These findings strongly support the notion that Rca1 and Efg1 regulate AMB susceptibility engaging the Nce103 carbonic anhydrase.

## Deletion of *NCE103* has marginal effects on lipidomes

The interference with ergosterol function is a primary mode of action for AMB. Hence, we analysed sterol lipid composition in mutants associated with the Rca1–Nce103–Efg1 axis (Extended Data Fig. 4a,b). Overall, AMB treatment altered relative ergosterol proportions, including sterol precursors, compared to the untreated group. Although lanosterol proportions increased by AMB exposure, differences in sterol composition between R and S strains were insignificant, consistent with a recent report[12]. The slight increase in relative lanosterol levels following AMB treatment was insignificant compared to sensitive strains (Extended Data Fig. 4b). Thus, alterations in membrane sterol composition are not the primary cause of CSP-mediated AMB susceptibility.

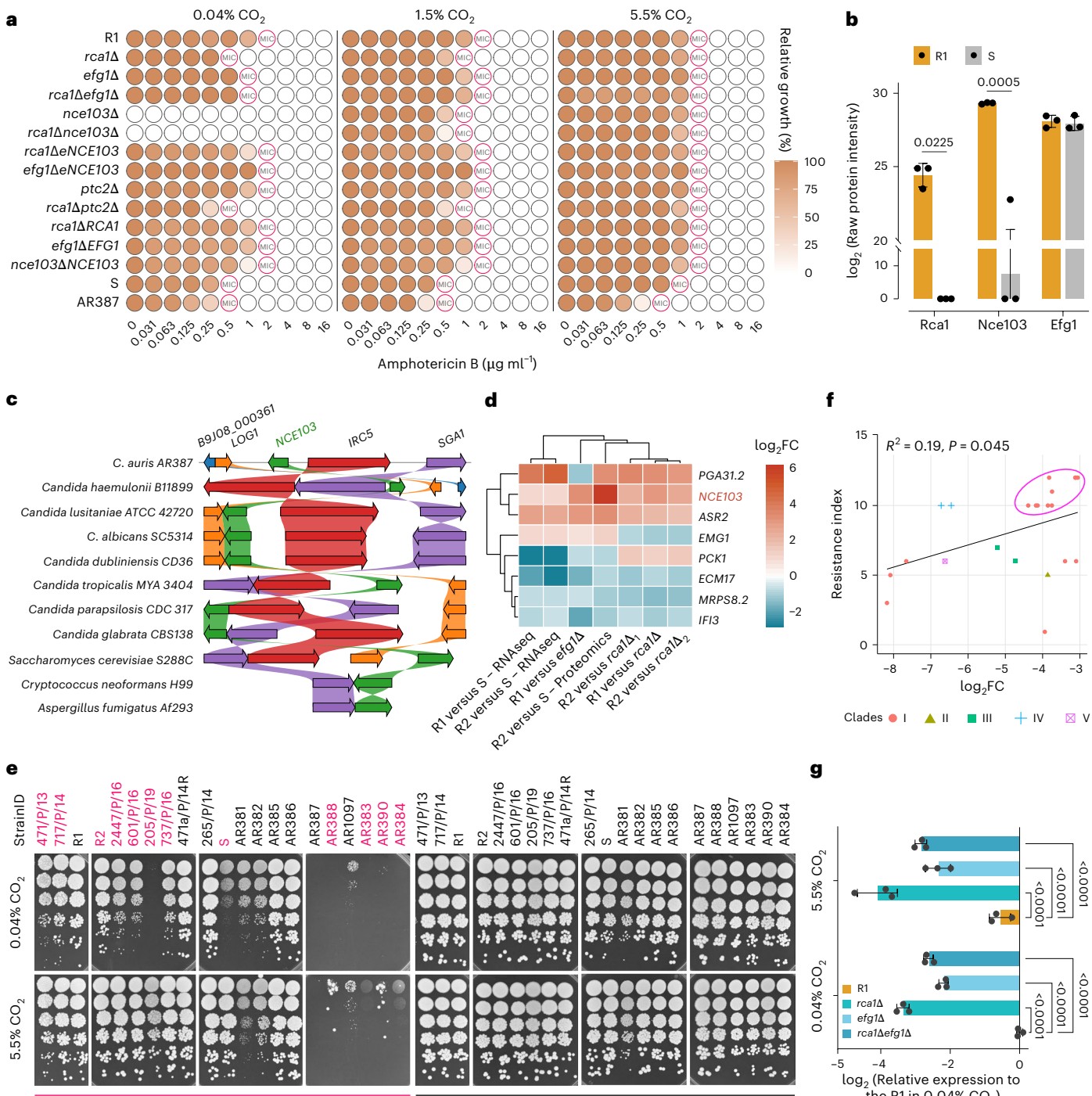

**Fig. 2 | A CSP controls AMB susceptibility. a**, Mutants in a CSP were subjected to MIC assays using the CLSI protocol (three biological replicates at indicated $CO_2$ concentrations). The deletion mutants $nce103\Delta$ and $rca1\Delta nce103\Delta$ fail to grow in ambient air (~0.04% $CO_2$). **b**, Protein levels of Rca1, Nce103 and Efg1 in strains R1 and S indicated by proteomics data ($n$ = 3 biological replicates per group). Adjusted $P$ values of DAPs from Limma analysis (Fig. 1d) are shown, and error bars indicate mean ± s.d. **c**, A synteny scheme shows conservation of carbonic anhydrase genes across fungal pathogens. **d**, Integrated heat map between proteomics and RNA-seq datasets reveal the upregulation of $NCE103$ in resistant strains. Cut-off: $log_2FC = \pm 0.58$. **e**, $CO_2$ enhances AMB tolerance in a subset of clinical *C. auris* strains (highlighted in bold pink). Spot dilution assays were performed using 5-fold serial dilutions of *C. auris* cells spotted onto YPD agar with or without AMB. **f**, Two-sided Pearson correlation between $NCE103$ mRNA

expression ($log_2FC$ ($NCE103/ACT1$)) and AMB resistance among clinical *C. auris* isolates. AMB resistance indexes were semi-quantified using spot dilution assays on AMB plates. Two AMB concentrations (1 µg ml⁻¹ and 2 µg ml⁻¹) were used to capture a range of susceptibilities. Two biological replicates were performed, yielding consistent results. **g**, $NCE103$ mRNA levels in *C. auris* mutants under low (0.04%) and high (5.5%) $CO_2$ conditions show that both Rca1 and Efg1 modulate $NCE103$ expression ($n$ = 3 biological replicates per group). Approximately 200 c.f.u. of *C. auris* were plated onto YPD agar and incubated at 37 °C under either 5.5% $CO_2$ or ambient air conditions for 2 days. Colonies were then collected for mRNA isolation. Two-way ANOVA followed by Dunnett's test was applied with R1 used as control group. Each data point is shown, and error bars indicate mean ± s.d. Orthologues and gene order information were retrieved from the *Candida* Gene Order Browser (cgob.ucd.ie) and fungi.ensembl.org.

Growing evidence suggests that inositol phosphorylceramides (IPC) contribute to AMB tolerance[34]. Therefore, we performed global lipidomics of the *nce103Δ* mutant and the reconstituted *nce103ΔNCE103* control, but we failed to see differences in lipid species between the reconstituted and mutant strains (Extended Data Fig. 4c). Likewise, sphingolipids such as IPC (Extended Data Fig. 4d) were unchanged. The fungal-specific IPC synthase inhibitor aureobasidin A (AbA) enhances the efficacy of AMB against *Cryptococcus neoformans*[34]. AbA similarly enhanced AMB activity in *C. auris*, but it showed no significant differences between WT and CSP mutants (Extended Data Fig. 4e). PCA analysis revealed that the *nce103Δ* samples were more dispersed with a slight separation from the WT strain following AMB treatment (Extended Data Fig. 4f), suggesting that dynamic lipid changes may still contribute to AMB susceptibility[12,18].

### The CSP may influence AMB^R by supporting metabolism and mitochondrial functions

To further investigate the impact of CSP on AMB susceptibility, we conducted semiquantitative spotting assays on R and S strains using various stress inhibitors (Fig. 3a). Remarkably, deletion of *RCA1*, *EFG1* and *NCE103* resensitized the R1 strain to cell membrane and osmotic stressors, including sodium dodecyl sulfate (SDS) and sodium chloride (NaCl), oxidative stress from hydrogen peroxide ($H_2O_2$), and the mitochondrial complex III block through antimycin A (AA). By contrast, overexpression of *NCE103* in the S strain reduced susceptibility to NaCl, $H_2O_2$, AA, the mutagen methyl methane sulfonate (MMS) and the mammalian target of rapamycin (mTOR) inhibitor rapamycin (Fig. 3a). The most striking effect was observed with AA (Extended Data Fig. 5a), correlating with the enhanced carboxylic acid metabolism upon *RCA1* or *EFG1* deletion (Fig. 3c).

Impaired mitochondrial functions can also promote accumulation of reactive oxygen species (ROS)[12], thus explaining the increased $H_2O_2$ susceptibility of *rca1Δ* and *efg1Δ* deletion mutants. Increased intracellular ROS can also exacerbate membrane lipid and DNA damage[12], which was strongly supported by the apparent hypersensitivity to SDS and MMS, respectively (Fig. 3a,b). Consequently, disruption of the CSP causes AMB hypersensitivity through mechanisms linked to mitochondrial function and ROS response. If this effect was primarily related to ROS, we hypothesized that antioxidants could reverse the AMB^R phenotype. To test this, we performed serial dilution spotting assays using *rca1Δ*, *efg1Δ* and *nce103Δ* mutants on media with or without AMB, vitamin C (VitC) or their combination (Fig. 3d). The supplementation with VitC enhanced AMB tolerance in all null mutants, as well as in the sensitive WT strain AR387 harbouring a fully functional CSP. Therefore, fluctuations in ROS levels can influence AMB^R traits in *C. auris*. However, there were no significant changes of AMB^MIC in mutants, although AR387 showed a 2-fold increase when supplementing with 250–1,000 μM VitC (Extended Data Fig. 6). Hence, ROS accumulation alone cannot fully explain AMB sensitivity resulting from a disabled CSP.

Because all deletion mutants in CSP were hypersusceptible to AA (Extended Data Fig. 5a), we hypothesized that supplementation with AA in the MIC assay should further reduce AMB^MIC from 2- to 4-fold as observed in null mutant strains. If mitochondrial function operates downstream of the CSP, AA should cause less reduction to the AMB^MIC in CSP null mutants than the WT. Hence, we subjected *rca1Δ*, *efg1Δ* mutants and e*NCE103* overexpression strains to checkerboard assays with AA and AMB. As expected, AA caused a 2-fold reduction of AMB^MIC in WT strains from five clades and in e*NCE103* overexpression strains, but not in null mutants (Fig. 3e and Extended Data Fig. 5c). Therefore, CSP may promote AMB^R by engaging mitochondrial functions.

### Targeting mitochondrial cytochrome *bc1* enhances AMB efficacy

Lansoprazole can synergize with AMB by inhibiting cytochrome *bc1* in *C. auris*[35], which is consistent with our data for AR387 and R1, yielding fractional inhibitory concentration index (FICI) values of 0.625 and 0.75,

respectively. Given that CSP disruption may be linked to mitochondrial dysfunction via the AA target cytochrome *bc1*, we attempted to delete *RIP1*, the catalytic subunit of this complex. While we were unable to ablate *RIP1* in R1 or its isogenic CSP mutants, we successfully obtained *rip1Δ* deletion mutants in the AR387 (B8441) background. The *rip1Δ* mutants had an approximately 2-fold reduction in AMB^MIC (Fig. 4a,b). Owing to the AA toxicity for humans, we used the fungal-selective cytochrome *bc1* inhibitor methyltetrazole analogue Inz-5[36] to assess its potential in enhancing AMB efficacy. Indeed, Inz-5 reduced AMB^MIC in several *Candida* species as well as in *C. neoformans*. Checkerboard assays suggested that effects of combinatorial treatments ranged from indifferent to synergistic based on the calculated FICI values (Fig. 4c). Therefore, the Rip1 mitochondrial cytochrome *bc1* component could be a promising target for potentiating AMB efficacy.

### The CSP is required for fungal fitness and colonization of nutrient-limited skin niches

The CSP influences not only AMB^R but also fungal fitness. In minimal medium (MM), both mutants and the WT S strain showed severe growth defects, even when supplementing with 5.5% $CO_2$ (Fig. 5a). Overexpression of *NCE103* in strain S strongly promoted fungal growth in MM and yeast–peptone (YP). Glucose as a carbon source enabled all mutants to reach the stationary growth phase with maximal optical density at 600 nm ($OD_{600nm}$) similar to the WT strain. Hence, glycolytic energy metabolism is linked to the Rca1–Nce103–Efg1 axis. Indeed, *efg1Δ* exhibited a severe fitness defect that was not fully rescued by supplementing with 5.5% $CO_2$ in YP medium. Impacting energy metabolism may trigger compensatory mechanisms, as reflected by the upregulation of carbohydrate transport in both *rca1Δ* and *efg1Δ* mutants (Extended Data Fig. 7a).

Although *rca1Δ* regrew at later time points in YP–glucose (YPD) medium, growth was delayed during the first 8 h compared to R1 (Extended Data Fig. 7b). Thus, CSP is particularly important during early culture stages at low cell densities, when metabolism-derived $CO_2$ is insufficient to support growth[37]. Indeed, initiating growth at low cell densities in RPMI (Roswell Park Memorial Institute) medium resulted in a marked growth delay of *rca1Δ* and *efg1Δ* deletions (Extended Data Fig. 7c). Next, we epitope-tagged native *NCE103* in the R1 strain with dTomato to quantify Nce103 levels in the growth phases. Nce103 was actively expressed during early stages of culture but downregulated during late exponential growth or when 5.5% $CO_2$ was used as supplement (Fig. 5b). These results indicate that *NCE103* is also critical for supporting initial growth when metabolism-derived $CO_2$ is limited.

Maintaining fitness is pivotal for host skin colonization as well as for the dissemination of *C. auris* in hospital settings. To assess whether CSP can impact fitness and growth of *C. auris* from different clades, we tested clinical isolates in minimal media (Fig. 5c). In fact, $CO_2$ supplementation enhanced growth of *C. auris* strains from five clades in nutrient-limiting conditions. Hence, the CSP may play key roles in maintaining fungal fitness when growing in or colonizing nutrient-limited environments.

### *C. auris* requires the CSP for growth on human skin

*C. auris* is a skin-tropic colonizer, and its ability to persist and penetrate from the skin's surface into deeper compartments is crucial for causing systemic infections[38]. Thus, we infected C57BL/6 mice with the WT (R1) strain and *rca1Δ*, *efg1Δ* and *nce103Δ* mutants by intradermal injection. Competitive quantitative fitness assays revealed that the R1 strain exhibited pronounced increased fitness compared to the *efg1Δ* mutant, but there were no remarkable differences between R1 and the *rca1Δ* or *nce103Δ* mutants (Fig. 5d). Therefore, the CSP does not contribute to fitness when *C. auris* reaches deeper skin compartments as by intradermal infection.

Although the *nce103Δ* mutant failed to grow in ambient air (Extended Data Fig. 7c), it can still establish long-term colonization

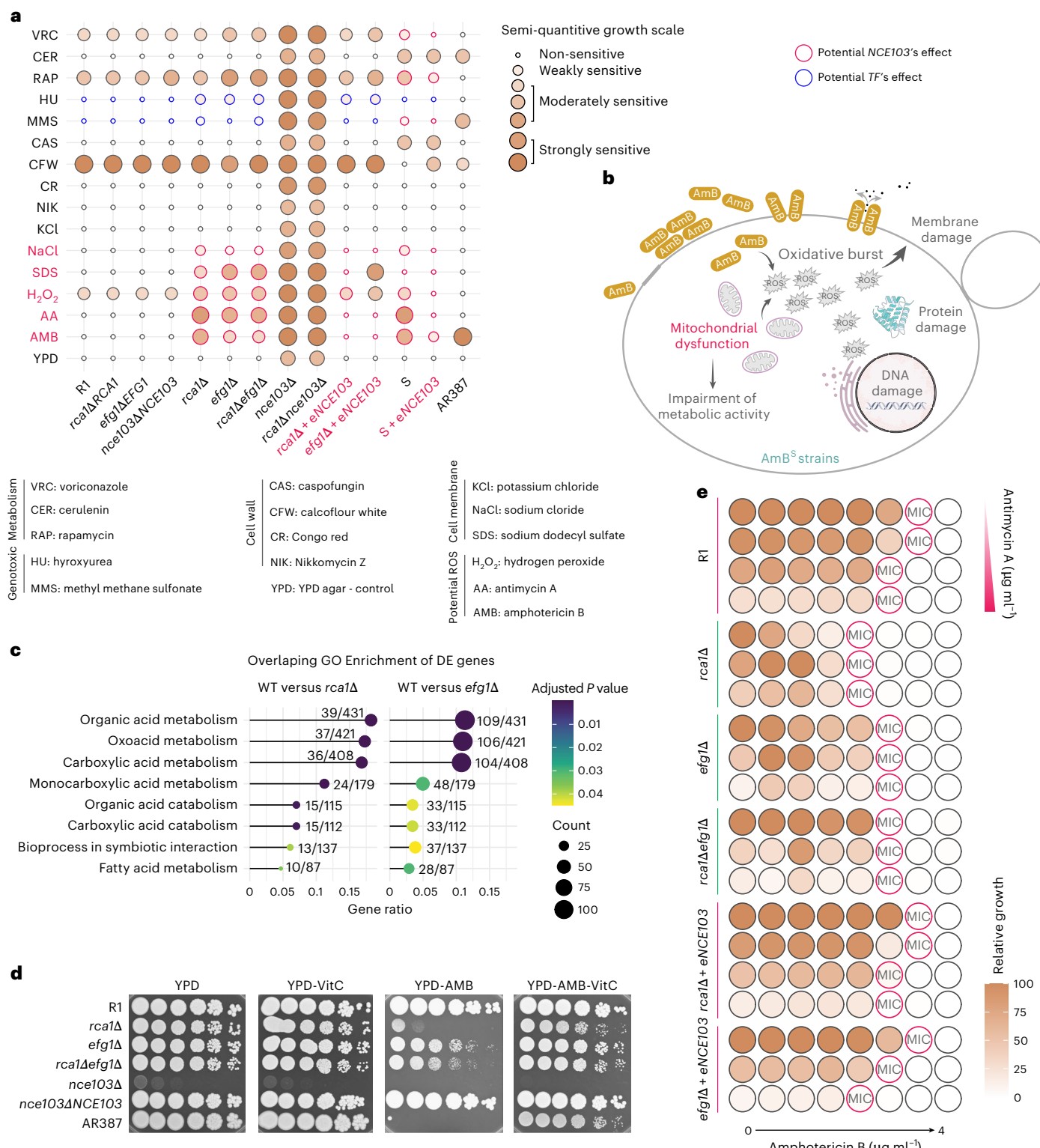

**Fig. 3 | Different mechanisms contribute to CSP-mediated antifungal resistance. a**, Semiquantitative spotting assays with different stress agents. Serial 5-fold dilutions of fungal cell suspensions were spotted onto YPD agar containing chemical agents as indicated. Semiquantitative data were collected by comparing relative growth properties of tested strains. **b**, Potential mechanisms implicated in CSP-mediated AMB resistance deduced from data shown in **a**. **c**, Gene Ontology enrichment for differentially expressed genes in *rca1*Δ and

*efg1*Δ mutants compared to WT. Data were filtered with cut-off log$_2$FC of ±0.58. Common Gene Ontology terms of *rca1*Δ and *efg1*Δ were visualized. **d**, Spotting assays for CSP mutants on media containing 3 µg ml⁻¹ AMB and 5 mM VitC. **e**, MIC assays for AMB alone and in combination with the mitochondrial inhibitor AA. The AA concentration ranged from 0 to 1 µg ml⁻¹ (top to bottom) for each strain. Three biological replicates yielded similar results in **a**, **d** and **e**. Credit: icons in **b**, Bioicons.com under a Creative Commons license CC BY 3.0.

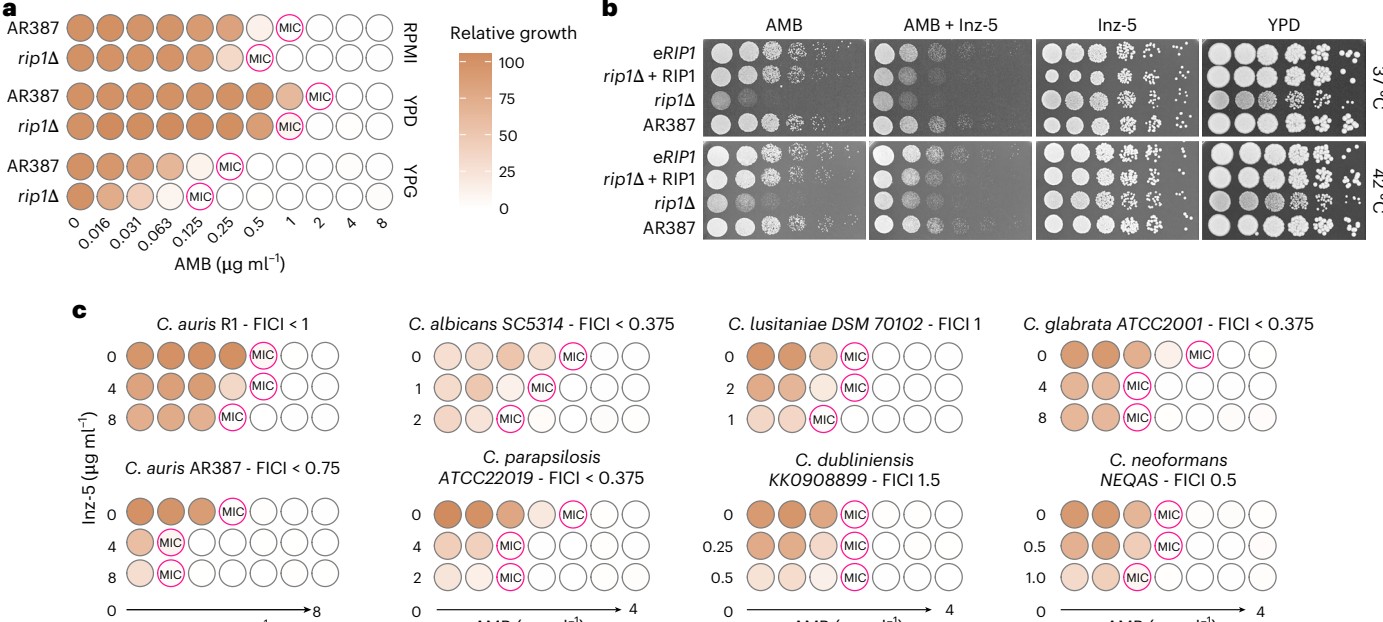

**Fig. 4 | Targeting mitochondrial cytochrome *bc1* enhances AMB susceptibility.** **a**, Deletion of *RIP1* encoding the active subunit of the cytochrome *bc1* complex increases AMB susceptibility. MIC assays were performed for *C. auris* in RPMI, YPD and YP–2% glycerol (YPG). **b**, The cytochrome *bc1* inhibitor Inz-5 partially phenocopies the sensitivity of the *rip1Δ* mutant in a serial dilution spotting assay, in both the presence and absence of AMB. **c**, Inz-5 reduces the AMB[MIC] by about 2- to 4-fold across multiple fungal pathogens in checkerboard assays. Two to three biological replicates were performed in **a**–**c**.

on intact mouse skin (Fig. 5e). However, *nce103Δ* is unable to efficiently colonize mouse in vivo and human skin ex vivo after 48 and 24 h, respectively, showing 10- to 100-fold lower burden compared to the WT strain (Fig. 5f,g). This confirmed the critical role of Nce103 for initial colonization of native skin tissues. After growing for 14 days on mouse skin, *C. auris* was mainly seen around or within hair follicle shafts[39], where higher $CO_2$ levels emerging from epithelial metabolism and bacterial skin microbiome components may further support fungal growth[40]. By contrast, *rca1Δ* and *efg1Δ* mutants exhibited a striking 10- to 100-fold decrease in colonization of both mouse and human skin compared to the WT control (Fig. 5e–g). Therefore, the CSP is crucial for the initial colonization of native human and mouse skin (Fig. 5h,i). Moreover, Efg1 is required for fitness, virulence and long-term persistence on mouse skin, as well as for subsequent infection, although independently of Nce103.

## $CO_2$ derived from bacterial metabolism can promote *C. auris* growth

As there were no significant differences observed in long-term persistence on mouse skin between the *nce103Δ* mutant and the WT, it suggests that $CO_2$ produced in hair follicles may contribute to the mutant's fitness. In addition, $CO_2$ could also be generated through microbiota metabolism. Specifically, *P. mirabilis* and *K. pneumoniae* are highly enriched bacteria in *C. auris*-positive skin samples[41]. Both bacteria are positive for urease, an enzyme that catalyses the breakdown of urea in the microenvironment into ammonia and $CO_2$ (Fig. 6a)[42]. Based on this, we hypothesized that bacterial urease activity supplies the fungal CSP, thereby supporting *C. auris* fitness on the skin microenvironment. Consistently, *P. mirabilis* and *K. pneumoniae* significantly enhanced the growth of CSP mutants in vitro (Fig. 6b and Extended Data Fig. 7d). However, this growth-promoting effect was lost when using urease-negative mutant *ureC*[43] of *P. mirabilis*, similar to results observed with urease-negative *Staphylococcus aureus* or the no-bacteria control. These findings strongly indicate that urease-positive bacteria that co-colonize on skin facilitate *C. auris* growth by releasing $CO_2$, the key substrate for the CSP.

## Discussion

One of the open unsolved questions in *C. auris* pathophysiology is why and how up to 60% of clinical isolates[6,7] can acquire high AMB[MIC]

**Fig. 5 | Rca1 and Efg1 are required for skin colonization and fungal fitness.** **a**, Growth curves were recorded at $OD_{600nm}$ for 72 h in MM, YP and YPD medium. **b**, Nce103 protein levels under different conditions. The *NCE103* locus in the R1 strain was tagged with dTomato; fluorescence signals were quantified by flow cytometry. **c**, Growth properties of *C. auris* from five clades are promoted by low and high $CO_2$ on MM. Tenfold serial dilutions of *C. auris* cell suspensions were spotted on MM agar and incubated at 37 °C for 5 days. **d**, Competitive fitness assays of mutants and the WT strain (R1) following intradermal infections after 3 days (n = 6 or 8 per group). The competitive index was determined using qPCR assays for gDNA isolated from a mixture of *C. auris* strains (left). For *nce103Δ* mutants, fitness was assessed by counting of colony-forming units after culturing in ambient air (control *nce103Δ* does not grow) and in 5.5% $CO_2$. **e**, Fungal burden in murine biopsies after colonizing intact back skin for 14 days (n = 6 for R1, *rca1Δ* and *efg1Δ*; n = 10 or 11 for other groups). **f**, Determination of the fungal burden in skin biopsies after 2 days of colonization on back skin (n = 5 or 6 per group). **g**, Fungal burden on human skin (9 biopsies from 3 human donors). Fungal

suspensions of the WT strain (R1) and the mutants were applied on human skin biopsies. After 24 h, biopsies were washed in PBS, and combined colony-forming units were quantified from washing supernatants and skin tissues, with the total representing the sum of colony-forming units in both fractions. Statistical analyses were performed on average colony-forming units data from each donor. **h,i**, Representative images of SEM (**h**) and periodic acid–Schiff staining (**i**) of human skin topically infected with *C. auris* for 24 h. Scale bar, 10 μm in **h** and 100 μm in **i**. *P* values were obtained by two-tailed paired *t*-test in **d**, one-way ANOVA in **e**, two-way ANOVA in **f**, and one- or two-way ANOVA for wash/tissue or total colony-forming units, respectively, in **g**, followed by Bonferroni or Tukey's multiple comparisons tests. Box plots indicate the median and interquartile range, with whiskers representing minimum and maximum values using data pooled from at least two independent experiments in **d**–**g**. Each data point and exact *P* < 0.05 are shown in **d**–**g**. Panel **g** (right) created with BioRender.com. Credit: icons in **d**–**g**, Bioicons.com under a Creative Commons license CC BY 3.0. Panel **g** (left) adapted with permission from ref. 53, Springer Nature Limited.

values. Although previous efforts focused on genetic variations[10,14], they cannot explain this phenomenon in the majority of AMB[R] isolates, especially as AMB[R] has been rarely seen in other *Candida* pathogens. Here, we took advantage of integrated proteomics and transcriptomics data from clinical strains to uncover a CSP that plays a crucial role in

AMB[R] as well as pathogen fitness and skin colonization. We show that the CSP engages the Nce103 carbonic anhydrase under the control of the transcription factors Rca1 and Efg1. We demonstrate that the Rca1–Nce103–Efg1 axis not only controls AMB susceptibility but is also essential for fungal fitness and skin colonization. Although 2- to

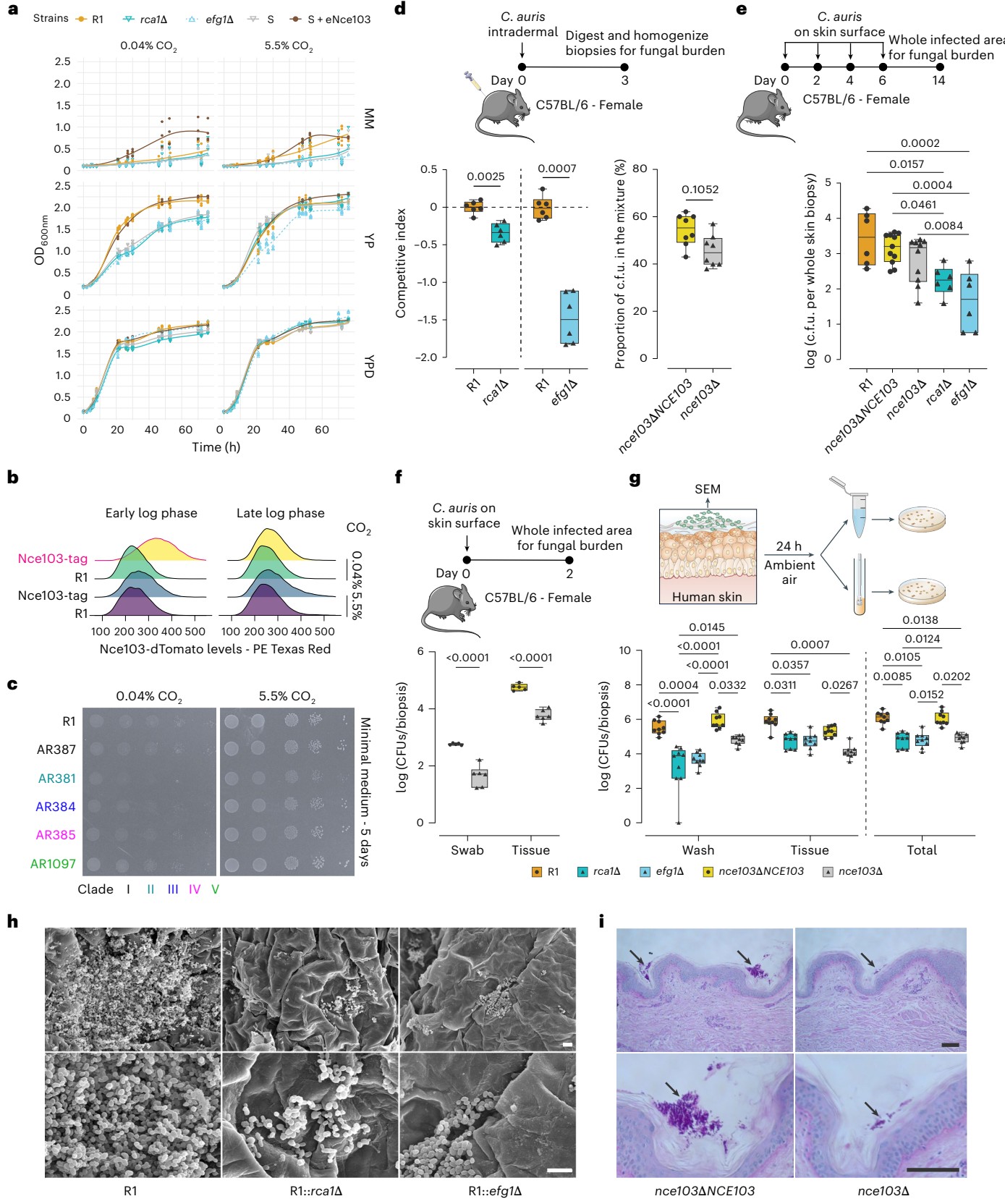

**a**

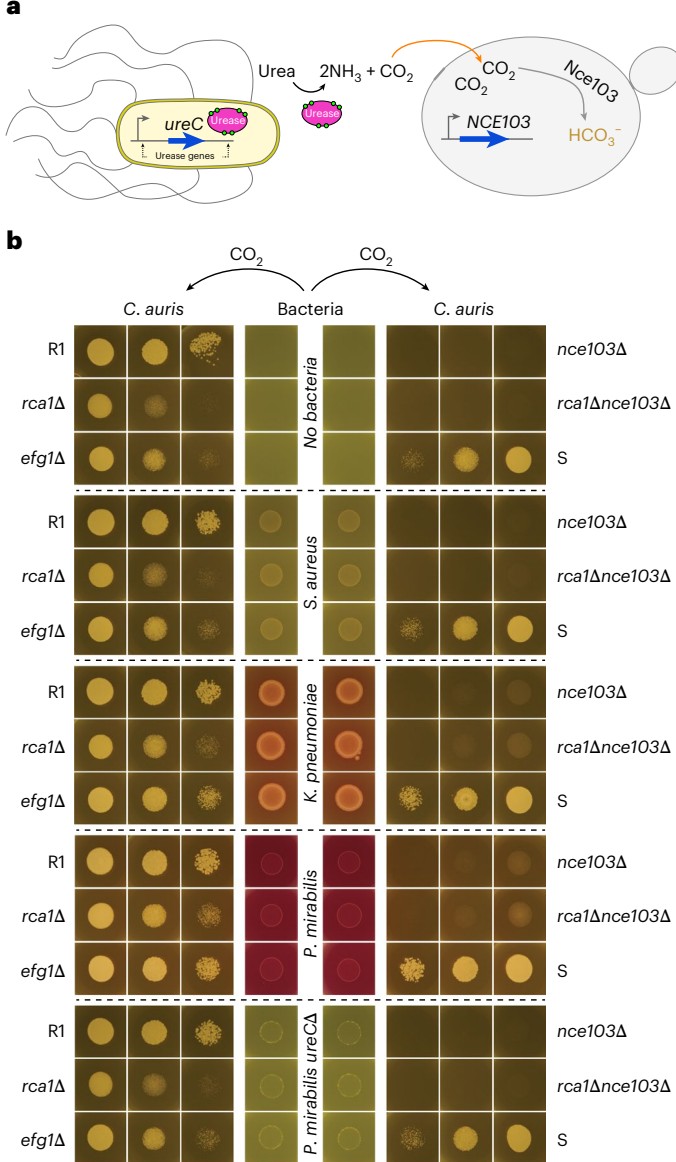

**b**

Fig. 6 | *C. auris* can scavenge $CO_2$ released by bacterial skin colonizers. **a**, A schematic illustration depicts how urease-positive bacteria supply $CO_2$ to the CSP of *C. auris*. **b**, Hospital-acquired bacteria can promote growth of *C. auris* by releasing $CO_2$ through the urease pathway. Experiments were performed on 12-well plates, where bacterial cell suspensions were spotted in the two middle columns and fungal cell suspensions in both outer columns. Each fungal strain was tested at three levels of 10-fold dilutions. The plates were covered, sealed in plastic bags and incubated at 37 °C for 24 h (Extended Data Fig. 7d). Representative data from at least three independent experiments.

4-fold reduction in AMB$^{MIC}$ seems a moderate effect, it is highly relevant for therapeutic outcomes[44–46]. We noticed that the CSP provides an intrinsic mechanism establishing a tolerogenic background to AMB in *C. auris*. However, higher concentration of $CO_2$ in deeper skin tissue helps *C. auris* bypass the need for a CSP and may thus regain AMB$^R$.

The downstream functions of the CSP link to mitochondrial functions, which in turn could regulate both fitness and AMB susceptibility. Indeed, acquired AMB$^R$ in clade II strains appearing after microevolution in vitro is associated with upregulation of the alternative oxidase Aox2. Deletion of *AOX2* in AMB$^R$-adapted strains reduced the MIC but had no significant effect on parental strains, implying that *AOX2* may contribute to controlling AMB susceptibility by involving interactions with other as yet unknown genetic factors[12]. However, our data suggest that the lack of *AOX2* in R1, *rca1Δ* or *efg1Δ* backgrounds does not change

the AMB$^{MIC}$ (Extended Data Fig. 5b), although *aox2Δ* mutants show severe sensitivity to AA. Similarly, deletion of the mitochondrial superoxide dismutase homologue *SOD2* yielded MIC comparable to *aox2Δ* mutants. These findings suggest that the ROS response may have a minor role in AMB$^R$ in *C. auris*. Alternatively, the effects may result from a combination of ROS[12], mitochondria-dependent stress response and other mechanisms. Indeed, targeting mitochondrial cytochrome *bc1* by ablating *RIP1* or by the lead compound Inz-5 enhances AMB efficacy across in species, identifying Rip1 as an antifungal target.

Although the precise mode of AMB action remains ill-posed even almost 70 years after approval, the consensus suggests the membrane lipid bilayer function as the primary AMB target. AMB binds or sequesters ergosterol-related lipids to exert its fungicidal activity[46,47]. Surprisingly though, our data reveal no significant differences in ergosterol levels or IPC-related lipids. Thus, we speculate that AMB–lipid interactions may cause dynamic changes in the relative lateral or transversal architecture of ergosterol and phospholipids in the plasma membrane without affecting overall lipid levels. For instance, 'sterol-sponges' may cause membrane instability and the breakdown of the electrochemical gradient[47]. Further, we hypothesize that the dynamic mosaic organization of lipid bilayers may be subject to control by CSP and/or mitochondrial function. Our lipidomic analysis supports this notion, as substantial fluctuations of lipid landscapes across biological replicates are evident in *nce103Δ* (Extended Data Fig. 4f). In addition, lack of *RCA1* or *EFG1* dysregulates multiple membrane transporters, many of which have been implicated in multidrug resistance phenomena (Extended Data Fig. 8a,b). Fluorescein diacetate-based assays suggest increased membrane fluidity, confirming protein-independent changes in bilayer permeability[17]. Although we observe increased fluorescein diacetate uptake in some experiments (Extended Data Fig. 8c), these results are variable across independent replicates similar to what we observe for lipidomics data in *nce103Δ*. Nonetheless, these observations strongly confirm a potential mechanistic link between AMB$^R$ and the dynamic membrane lipid homeostasis.

A model for fungal CSP wiring (Fig. 1f) suggests that Nce103 ultimately shunts HCO$_3^-$ into mitochondrial energy metabolism to support fatty acid and amino acid biosynthesis. Furthermore, it regulates mitochondrial functions through the cAMP–PKA–Efg1 signalling[24,48,49]. Deletion of *CYR1*, *RAS1* and *CDC25* in strain AR387 increases AMB tolerance[23,25] (Extended Data Fig. 3h), implying negative feedback regulation by downstream components within the CSP or a complex interplay with as yet unknown additional wiring in *C. auris*. Regarding *C. auris* fitness in the in vivo intradermal model and long-term colonization in vivo, Efg1 may involve additional independent mechanisms on top of the carbonic anhydrase Nce103. By sharp contrast, lack of *RCA1* or *EFG1* or *NCE103* does not cause significant differences in fungal burden recovered from various organs after systemic infections, even when mice were treated with AMB (Extended Data Fig. 7e,f). Moreover, differences in MIC changes between *rca1Δ*, *efg1Δ* and *rca1Δefg1Δ* mutants imply overlapping as well as independent mechanisms Rca1 and Efg1 engaging to regulate AMB susceptibility. This is a plausible scenario, as deletion of *RCA1* or *EFG1* dysregulates 35–50% of the putative transcription factors in the *C. auris* genome (Extended Data Fig. 7g), indicating additional roles. Although a CSP is well documented in other fungal species[24,28,29,48,50], its unique role in promoting AMB$^R$ has not been demonstrated. For instance, deletion of *Candida albicans RCA1* slightly increases susceptibility to 5-fluorocytosine while enhancing resistance to echinocandins and azoles[48]. Similarly, overexpression of the carbonic anhydrase Can2 in *C. neoformans* leads to hypersensitivity to polyenes and azoles[51].

The CSP plays a crucial role in fungal colonization and growth on skin niches, which is consistent with reports for other *Candida* species[29]. It is worth noting that the Rca1–Nce103–Efg1 axis enables *C. auris* growth in nutrient-limited environments such as human skin or abiotic surfaces of healthcare devices or medical equipment.

Indeed, *C. auris* and related species can grow on solid MM medium, particularly under elevated $CO_2$ levels (Fig. 5c and Extended Data Fig. 8d). This implies potential risks associated with carboxytherapy such as topical $CO_2$-releasing gels or subcutaneous applications[52] used in wound management. Although *C. auris* appears unable to breach intact epithelial barriers as present on skin, the pathogen is able to reach deeper skin compartments through microwounds, damaged skin or via hair follicle invaginations[38,39].

In hair follicles, metabolism-derived $CO_2$ levels from human cells or bacterial microbiota components may compensate for impaired CSP activity. For instance, the bacterial urease pathways may synergize with the CSP in controlling fungal fitness on skin tissues. We show that bacteria-derived $CO_2$ can support *C. auris* growth in vitro. In clinical swab samples from skin, *C. auris* is often enriched with nosocomial transmission-associated urease-positive bacteria such as *P. mirabilis*[42] and *K. pneumoniae*[41]. Nonetheless, it is critical to emphasize that Nce103 is essential for facilitating initial skin colonization, preceding (passive) skin penetration and subsequent systemic infections once reaching the vasculature (Extended Data Fig. 9)[53,54]. Our findings strongly support the notion about a possible skin microbiome contribution to facilitate or sustain fungal colonization[41], as bacteria can promote *C. auris* growth through their urease-mediated $CO_2$ release. This enzyme degrades urea, which is naturally present in sweat and commonly used in skincare products. While eccrine sweat glands are widely distributed in human skin, they are rare in fur-covered mammals such as mice[55]. Therefore, there is a need to establish suitable animal models and ex vivo human skin systems to investigate interkingdom interactions on the skin surface[56].

Taking these together, we answer a major question in *C. auris* biology related to AMB[R] and fungal virulence. We identify a CSP under the control of a dedicated bZIP transcription factor Rca1, which regulates the carbonic anhydrase Nce103 effector enzyme. Interestingly, Efg1 may be serving as both regulator of and downstream effector for the CSP. We further demonstrate that abrogation of the CSP may impair mitochondrial function, which debilitates fitness and AMB[R] traits. Therefore, the *C. auris* CSP enables growth on human skin and potentially exploits carbon dioxide that is released in the skin microenvironment by microbiome components (Extended Data Fig. 9). Finally, our data suggest that targeting microbiome-derived $CO_2$ production via bacterial urease inhibition may offer a strategy to limit *C. auris* skin colonization.

## Methods

### Ethics statement

Animal experiments adhered to ethical approval from the ethics committee of the Medical University of Vienna and the Federal Ministry of Science and Research, Vienna, Austria (BMBWF-66.009/0436-V/3b/2019). Adult wild-type C57BL/6 mice (*Mus musculus*) were housed in specific pathogen-free conditions, with controlled temperature (20–22 °C) and humidity (45–65%), in a 12 h light/dark cycle at the animal facility of the Max Perutz Labs Vienna. Mice breeding and maintenance was in accordance with ethical animal license protocols complying with the applicable Austrian law. Abdominal human skin samples were obtained from anonymous healthy adult female donors with approved consent following the Declaration of Helsinki and ethics committee approvals of the Medical University of Vienna (ECS 1969/2021). Donors were included based on availability, without random selection or specific covariate criteria such as age, sex or medical history. Given the small number and the use of samples solely for ex vivo colonization assays, potential selection bias is not expected to impact the study outcomes.

### Fungal growth conditions, media formulations and dose–response assays

A list of fungal strains used in this study is provided in Supplementary Table 2. *C. auris* was routinely grown in YPD medium at 30 °C, with

shaking at 200 r.p.m. For growth curve experiments, we used MM or YP with different carbon sources. All medium compositions are provided in Supplementary Table 3. All chemical reagent vendors and identifiers are provided in Supplementary Table 4.

Growth curves were derived for different culture media formulations as described in Supplementary Table 3. Fungal cells from an overnight culture in YPD at 30 °C, with shaking at 200 r.p.m., were washed 3 times with distilled water ($dH_2O$), followed by measurement of the $OD_{600nm}$. Cell suspensions were adjusted to 0.2 $OD_{600nm}$ in $dH_2O$. Aliquots of 100 µl of testing media (two times concentration) were dispensed into non-treated 96-well plates (Starlab). Subsequently, 100 µl of fungal suspensions was added to each well. Plates were incubated in static incubators at 37 °C with/without 5.5% $CO_2$. The $OD_{600nm}$ was recorded every 2 h within 72 h by Victor Nivo plate reader (PerkinElmer).

Dose–response assays were done adhering to the Clinical and Laboratory Standards Institute (CLSI) standard M27-A3 (ref. 57) protocol in 96-well plates. Briefly, yeast cells from YPD agar plates 3–5 days old were inoculated overnight in liquid YPD medium. Fungal cultures were adjusted to $OD_{600nm}$ of 0.1 in distilled water. A 25 µl aliquot of fungal suspensions was diluted into 10 ml RPMI 1640 (Gibco) buffered with 35 g l[-1] 3-(N-morpholino)propanesulfonic acid (MOPS) (AppliChem), pH 7 (referred as RPMI). The 96-well plates containing 100 µl media with drug compounds were prepared by 2-fold serial dilutions. Then 100 µl of the fungal suspensions was aliquoted into each well. Negative controls lacked inoculum, whereas positive controls included inoculum without adding agents. Plates were incubated 24 h at proper conditions following experimental requirements. Optical density at $OD_{600nm}$ was recorded in Victor Nivo Microplate Reader. MIC was defined as the well showing at least 50% growth inhibition compared to the no-drug control for all drugs, except for AMB, where a 90% growth inhibition threshold was used. Checkerboard assays were performed similar to dose–response assays, using 2-fold serial dilutions of AMB on the *y* axis and AA, Inz-5 or AbA on the *x* axis. FICI values were calculated as reported before[58].

### Semiquantitative agar assays

Fungal cells were grown until the exponential growth phase in YPD liquid medium. Cells were counted by a CASY cell counter then adjusted to $2 \times 10^7$ cells per ml in PBS. A 5-fold serial dilution of fungal suspension was performed in a 96-well plate. Then 3 µl of each dilution was spotted onto YPD agar without/with stress agents targeting metabolism (8 µg ml[-1] voriconazole, cerulenin, 0.5 µg ml[-1] rapamycin), genotoxic stress (150 mM hydroxyurea, 0.06% MMS), cell wall (4 µg ml[-1] caspofungin, 30 µg ml[-1] calcofluor white, 0.02% Congo red, 64 µg ml[-1] nikkomycin Z), cell membrane (1 M potassium chloride, 1 M NaCl, 0.1% SDS), mitochondrial functions and ROS inducers (15 mM $H_2O_2$, 5 µg ml[-1] AA, 8 µg ml[-1] Inz-5, 0.5–3 µg ml[-1] AMB). Plates were incubated at designated conditions. Pictures were taken after 2 days. Semi-quantitative phenotypic traits were collected by comparing the growth fitness of different strains under specific stress conditions[59,60].

### Proteomics

Several colonies of *C. auris* growing on YPD agar were picked and re-grown overnight in RPMI. Fungal suspensions were transferred to 50 ml fresh RPMI to reach $OD_{600nm}$ of 0.1 in baffled flasks. After 5 h incubation at 30 °C with agitation of 200 r.p.m., AMB was added at a final concentration of 0.5 or 0 µg ml[-1] for an additional 2 h. Yeast cells were pelleted and washed 3 times in cold PBS (Sigma-Aldrich). Cells were resuspended in tubes containing 300 mg of glass beads (Sigma-Aldrich) and 1 ml of *Candida* lysis buffer (1% sodium deoxycholate, 100 mM Tris–HCl, 150 mM NaCl, 1 mM PMSF, 1 mM EDTA, 1 tablet per 50 ml complete protease inhibitor), followed by bead beating (FastPrep-MPI) (6 m s[-1] for 45 s, done 3 times). The supernatant was collected by centrifuging through the small hole created at the bottom of the tube with a G26 needle tip. Protein was precipitated by

4 volume acetones, at −20 °C, overnight. Three biological replicates were performed for each experimental group. Protein pellets were subjected to proteomic analysis using a standard workflow (see details in Supplementary Information).

### RNA isolation, quantitative PCR and RNA-seq

Fungal cultures were centrifuged, and cell pellets were rapidly frozen in liquid nitrogen. Dry cell pellets were then stored at −80 °C for later use. Total RNA was extracted using the TRIZOL method, followed by DNase I treatment using the same protocol described previously[8,61] (see details in Supplementary Information).

RNA quality and purification was checked with Nanodrop and conventional PCR-based quantification of $ACT1$ mRNA. For quantitative PCR (qPCR), first-strand complementary DNA was synthesized from RNA with Reverse Transcription System Kit (Promega). Subsequently, 15 ng cDNA was utilized for qPCR amplification, using the 2x Luna Universal master mix (NEB). For competitive assays, genomic DNA was used directly for qPCR. The data were analysed using the cloud-based system provided by Bio-Rad accessible at BR.io.

For RNA-seq analysis, fungal cells grown overnight were diluted into 15 ml fresh YPD to reach $OD_{600nm}$ of 0.1 in baffled flasks. Flasks were then incubated at 37 °C until they attained $OD_{600nm}$ of 2.5. Fungal cells were collected by centrifugation, and RNA was isolated using TRIZOL method. At least three biological replicates were subjected to RNA-seq. The library and sequencing were performed at the commercial Novogene Sequencing Facility (UK). Briefly, mRNA was enriched by poly-T oligo-attached magnetic beads followed by double-stranded cDNA library preparation. The quality-controlled RNA libraries were pooled and sequenced with 150 bp paired-end reads on the Illumina NovaSeq 6000 platform.

RNA-seq bioinformatics data analysis used a workflow established previously[15]. Differential expression analysis was conducted using EdgeR v3.40.2[62]. The false discovery rate was controlled by adjusting $P$ values using the Benjamini–Hochberg correction. Gene Ontology term enrichment analysis (enrichGO) and gene set enrichment analysis were performed with clusterProfiler (version 4.0)[63] using annotation data retrieved from fungiDB (www.fungidb.org).

### Generation of *C. auris* mutants

*C. auris* deletion mutants were generated by gene replacement as described before using a gene-specific deletion cassette with a dominant marker constructed using the three-way stitching PCR method[64]. Briefly, approximately 500 bp upstream and downstream flanking regions of the target gene were amplified from gDNA of *C. auris* strains. The *NAT1* or *NeoR* selection marker flanked by the *TEP* promoter and terminator was amplified from plasmid pTS50[64] or pTO149[33]. DNA fragments were purified by GeneJET gel extraction kit (ThermoScientific), followed by a stitching PCR to obtain gene deletion cassettes for genomic targeting.

*NCE103* null mutants were constructed using a CRISPR–Cas9 system kindly provided by the O'Meara lab[33]. The Cas9 amplicon was amplified from plasmid pTO135[33]. The DNA-repairing cassette was generated by fusion PCR to replace the *NCE103* coding region by *NeoR* marker amplified from plasmid pTO149[33]. A DNA cassette containing guide RNA (gRNA)–trans-activating CRISPR RNA (tracrRNA) was designed with Benchling[65]. A mixture of three DNA amplicons were transformed into *C. auris* using the routine electroporation protocol[64,66].

For overexpression of *NCE103*, we placed the *NCE103* locus under the control of the *ENO1* promoter from *C. auris*. The cassette containing *urNEUT1-pENO1-NCE103-pTEP1-NAT1(or NeoR)-drNEUT1* was cloned into a long intergenic region (*NEUT1*) located between B9J08_000423 and B9J08_000424 on chromosome 1. For the conditional deletion of *NCE103* and *CYR1* in AR387, a plasmid pCB323 containing the *tet-Off* system[67], assembled with the *NAT1* marker optimized for *C. auris*, was used. The *tet-Off* system was flanked by approximately 500 bp of the

left and right regions and directed into *NCE103* locus assembled in plasmid pCT06. The native promoter of *NCE103* was replaced with this *tet-Off* system, allowing for regulated gene expression under tetracycline supplementation.

For C-terminal *NCE103*-dTomato epitope tagging, plasmid pCT02 was constructed using 500 bp upstream and downstream flanking sequences of the native *NCE103* locus. The *dTomato* gene was inserted at the C-terminus immediately before the stop codon of the *NCE103* gene, followed by a NeoR selectable marker. All plasmids were assembled using the NEBuilder HiFi DNA Assembly Master Mix (NEB - M5520) according to the manufacturer's protocol.

PCR-amplified DNA cassettes were transformed into *C. auris* using the electroporation protocol as described before[64,66]. Several independent transformants were cultivated for 4 h in YPD/sorbitol (1:1) before being plated onto YPD agar containing 200 μg ml⁻¹ nourseothricin or 1,200 μg ml⁻¹ G418 plus 1,000 mg ml⁻¹ molybdate[68] and incubated at 30 °C for 3–4 days. Colony PCR with OneTaq 2X master mix (NEB - M0482) was used to control for loss of gene as well as correct genomic integration[64]. For *nce103Δ*, outgrowth and antibiotic selection plates were incubated in 10% $CO_2$. The outgrowth medium (YPD/sorbitol, 1:1) was degassed and preincubated overnight in 10% $CO_2$ before use. All oligonucleotide primers and plasmids are listed in Supplementary Tables 5 and 6.

### Lipidomic analysis

Fungal cells from exponential growth phase were cultured in 15 ml RPMI from $OD_{600nm} = 0.3$ at 37 °C with shaking at 200 r.p.m. After 5 or 16 h, cells were treated with 1 μg ml⁻¹ AMB for an additional 2 h. After treatment, cells were washed three times with sterile $dH_2O$ and flash-frozen in liquid nitrogen for subsequent analyses. Sterol lipids were extracted using a previously reported protocol and quantified by gas chromatography coupled to an Agilent 5977B quadrupole mass spectrometer (MS), with cholestane (Merck, C8003) as the internal standard[69,70]. Lipid extraction was performed using a routine chloroform-based extraction protocol[71] and further analysed by liquid chromatography coupled with MS/MS. Data sets from shotgun and targeted analysis were combined to calculate molar percentages (mol%) of individual lipid species[72–74] (see details in Supplementary Information).

### Mice infection and colonization assays

WT C57BL/6 mice, aged 8 to 14 weeks at the start of experiments, were used. Female mice were used for skin experiments, while both sexes were used for systemic infections via the tail vein.

For intradermal infections, an equal cellular mixture of $5 × 10^6$ WT and mutant cells in 15 μl PBS was injected into the shaved back skin after anaesthesia with ketamine–xylazine (100 mg ketamine per kg body weight and 4 mg xylazine per kg body weight). On day 3 after infection, mice were euthanized via cervical dislocation, and 8 mm skin biopsies from infected sites were collected. Skin biopsies were homogenized as described above, followed by plating on YPD agar containing ampicillin, tetracycline and chloramphenicol (YPD-CAT). Plates were incubated at 37 °C with 5.5% $CO_2$ for 48 h. Fungal cells were collected from plates for gDNA extraction and qPCR to identify the ratio of mutants and WT in each sample using the primers for *RCA1, EFG1* and *NAT1*. Fungal mixtures used for infection were plated to control the actual infection dose. Relative gDNA abundance of WT (primers for *RCA1 or EFG1*) and mutants (primers for *NAT1*) were calculated based on *ACT1* as a control gene. The relative gDNA abundance was adjusted with abundance ratio of input samples before calculating competitive values as $\log_2$ (relative abundance values to WT).

For systemic infections, WT mice (21.5 g) were injected via the lateral tail vein with $2 × 10^6$ fungal cells in 100 μl PBS[75]. On day 5 after infection, mice were euthanized via cervical dislocation, and colony-forming units (c.f.u.) in the spleen, kidney, liver and brain were quantified by plating. For AMB treatments, amphotericin B deoxycholate (Fungizone,

Gibco) was administered by intraperitoneal injection of 5 mg per kg body weight on days 1 and 2 after infection.

For skin colonization assays, mice were anaesthetized with ketamine–xylazine followed by hair removal from back skin using an electric shaver to clear an area ~9 $cm^2$. Four applications were conducted on the same area using suspensions containing $2 \times 10^8$ fungal cells (WT and mutant strains) in 100 µl PBS, applied every 2 days. Mice were killed by cervical dislocation on day 14, and fungal cells on the skin surface were collected by a cotton swab moistened with 80 µl PBS, then immediately plated on YPD-CAT agar. Skin at infection area was excised and dissociated in 500 µl of enzyme solution containing 1 mg $ml^{-1}$ Collagenase Type II (Gibco), 1 mg $ml^{-1}$ DNase I (Roche), incubated at 37 °C, 5.5% $CO_2$. After 1.5 h, samples were homogenized with a homogenizer (IKA 3386000) for 30 s, followed by plating onto YPD-CAT agar plates to quantify colony-forming units. For short-term colonization, *Candida* cell suspensions were applied to the back skin once on day 0. Fungal burden was assessed on day 2 via cotton swab samples and whole biopsy digestion.

Native adult human skin samples were obtained within 1 to 2 h after plastic surgery procedures[38]. Initially, the skin underwent a cleansing process using PBS before being punctured with an 8 mm KAI Biopsy Punch tool (Heintel 29045435). Skin biopsies were then placed into a 12-well plate containing Dulbecco's modified Eagle's medium supplemented with 10% fetal bovine serum and 1% penicillin–streptomycin. Fungal cells from exponential growth phase cultures were washed twice with PBS, and subsequently adjusted to $10^5$ cells per ml using a CASY cell counter (Roche). Fungal cell suspensions (3 µl) were applied topically to the centre of the biopsies, with PBS serving as non-infected control and incubated at 30 °C under ambient air. After 24 h, biopsies were collected and washed in 1 ml of PBS by vortex-mixing for 10 s. Subsequently, biopsies were sectioned into four pieces and subjected to digestion with a 500 µl enzyme solution containing 1 mg $ml^{-1}$ Collagenase Type II (Gibco) and 1 mg $ml^{-1}$ DNase I (Roche) and incubated for 3 h at 37 °C, 5.5% $CO_2$. Next, 500 µl PBS was added, and biopsies were homogenized with a mechanical tissue homogenizer (IKA 3386000) for 30 s. Washing suspension and homogenized biopsies were diluted and plated onto YPD-CAT agar for fungal burden assessment.

For all experiments, initial infection doses were adjusted using a CASY cell counter after 30 s of sonication and further verified by counting of colony-forming units. Cell density was also confirmed by recording $OD_{600nm}$ whenever *efg1Δ* was used in the experiments. Before infection, cell suspensions were vigorously vortex-mixed to minimize clumping and maintain planktonic cell suspensions that ensure uniform distribution.

### Scanning electron microscopy and histological staining

The preparation of human skin biopsies for scanning electron microscopy (SEM) analysis was performed essentially as described previously[76]. In brief, biopsies were collected after 24 h and fixed in Karnovsky's fixative (2% paraformaldehyde, 2.5% glutaraldehyde in 0.1 M phosphate puffer pH 7.4 from Morphisto) for at least 24 h. Samples were then washed two times in 0.1 M phosphate buffer at 4 °C (pH 7.3) for 2 min each, dehydrated in a graded ethanol series (50%, 70%, 80%, 90%, 95% and 100%) for 20 min each and immersed for 30 min in pure hexamethyldisilazane (Sigma-Aldrich) followed by air-drying. For SEM analyses, samples were sputter-coated with gold (Fisons Instruments Polaron Sputter Coater, SC7610) and examined with a scanning electron microscope (JSM 6310, Jeol) at an acceleration voltage of 15 kV.

Skin biopsies were collected 24 h after infection and fixed in 7.5% formaldehyde (SAV Liquid Production), before embedding in histology-grade paraffin (Sanova Pharma). Paraffin sections were subsequently immersed in xylene (ThermoFisher), followed by sequential treatments with 96% ethanol and 70% ethanol. Periodic acid–Schiff staining was performed in 5% periodic acid for 10 min, followed by a 20 min immersion in Schiff's reagent (both from Merck). Counterstaining was carried out using haematoxylin (Sigma-Aldrich) for 1 min, followed by washing and treatment with 3% hydrochloric acid in ethanol. Finally, the sections were dehydrated, mounted with synthetic mounting medium (Eukitt, Sigma-Aldrich), and visualized under a bright-field microscope (Olympus AX70).

### Bacterial-fungal co-culture assays

Bacteria from a Luria–Bertani agar plate were inoculated into 5 ml of Luria–Bertani liquid medium and incubated overnight at 37 °C with shaking at 200 r.p.m. The culture was then measured at $OD_{600nm}$ and diluted in PBS to an $OD_{600nm}$ of 0.5 for co-culture experiments. *C. auris* in the exponential phase was washed twice with $dH_2O$ and counted using a CASY cell counter. The fungal cells were adjusted to $2 \times 10^7$ cells per ml in $dH_2O$ and further serially diluted 10-fold for co-culture experiments.

In a 12-well plate, 2 ml of solid agar media was added per well for bacterial and fungal culture as shown in Extended Data Fig. 7d. For bacterial culture, Christensen urea agar (0.1% peptone, 0.1% glucose, 0.5% NaCl, 0.2% $KH_2PO_4$, 0.0012% phenol red, 2% urea, 1.5% agar, final pH 6.8 ± 0.2) was used and distributed to lines 2 and 3 of the plates. For *Candida* culture, YPD agar buffered to pH 5 with $Na_2HPO_4$ and citric acid was used. The same bacterial strain was spotted on lines 2 and 3, while different *C. auris* WT and mutant strains were spotted on lines 1 and 4, ensuring equal distances between fungal strains and bacterial cultures. The plates were sealed with plastic tape, placed in sealed plastic bags, and incubated at 37 °C for 24 h.

### Statistical methods

Statistical analyses were conducted with GraphPad Prism version 9.0 and rstatix R package version 0.7.2 (ref. [77]). Unless otherwise specified, data represent means ± standard deviation from at least three biological replicates. For mouse experiments, 5–11 mice were used for each group. For human skin experiments, data were collected from three independent human donors. Significance was determined using *t*-test or analysis of variance (ANOVA), followed by Bonferroni or Benjamini–Hochberg or Tukey's or Dunnett's post hoc tests for multiple comparisons. Cut-off *P* values indicating significance are given in figure legends.

### Reporting summary

Further information on research design is available in the Nature Portfolio Reporting Summary linked to this article.

## Data availability

The proteomics data were deposited to the ProteomeXchange Consortium via the PRIDE partner repository[78] with the dataset identifier PXD048342. RNA-seq data are available from the Gene Expression Omnibus database with the accession number GSE253332. Lipidomics datasets are provided in Supplementary Table 8. Source data are provided with this paper.

## Code availability

The code for RNA-seq analysis workflow is available via Github at https://github.com/kakulab/CSP2024. The workflow for processing MaxQuant output tables is available via Zenodo at https://doi.org/10.5281/zenodo.5758974 (ref. [79]).

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

## Acknowledgements

We thank A. Petryshyn, S. Volkan and all other lab members for their technical support and experimental advice. Moreover, we thank N. Chauhan (Center for Discovery and Innovation) for providing clinical isolates; T. O'Meara (University of Michigan Medical School) for providing the plasmids pTO135, pTO149 and pTO136; C. Chen (Chinese Academy of Sciences) and S. Znaidi (Memorial University of Newfoundland) for providing plasmid pCB323; M. Bradley (Queen Mary University of London) for the gift of Cy5-conjugated AMB; A. Brakhage (Leibniz Institute for Natural Product Research and Infection Biology-Hans-Knöll-Institute) for providing *Trichophyton benhamiae* (WT and *stuA* mutants); C. Freystätter (Medical University of Vienna) for providing the skin samples. We thank C. Nobile (University of California, Merced) and A.E.-B. for serving as members on the PhD Thesis Committee of T.P.-C; P.-L. Luu (Thong Nhat Hospital, Ho Chi Minh City) for guiding in bioinformatics. We are grateful to human volunteers donating native skin tissues. K.K. was supported by grants from the Austrian Science Fund (ChromFunVir - P-32582-B08, Candidomics - P-33425, and in part by FWF-SFB70-08). A.E.-B. received funds from the Austrian Science Fund - FWF (P-31485-B30) and the LEO Foundation (LF-OC-23-001332). T.P.-C. was supported by a Student Fellowship, Ernst Mach Grant, a Bernd Rode Award

2024 from ASEAN – European Academic University Network (ASEA-UNINET) network and the ESCMID Research Grant 2023 from the European Society of Clinical Microbiology and Infectious Diseases; T.P.-C. was also supported in part by the LEO Foundation (LF-OC-23-001332). H.A., I.T. and T.P.-C. were supported by the FWF-funded PhD training program (TissueHome – FWF-DOC32-B28). N.K. was supported in part by the LEO Foundation (LF-OC-23-001332). A.C. is a fellow of the Canadian Institute for Advanced Research, Fungal Kingdom. N.N.T., C.C. and R.A. received support from the Faculty of Chemistry, University of Vienna - Mass Spectrometry Centre, with funding from the Vienna Doctoral School in Chemistry and DigiOmics4Austria: BMBWF '(Digitale) Forschungsinfrastrukturen'.

## Author contributions

T.P.-C. and K.K. conceived the project and designed experiments. K.K. provided PhD supervision and mentoring. K.K., A.E.-B. and T.P.-C. acquired funding. T.P.-C. conducted most experiments, data analysis and visualization. A.E.-B. coordinated ethics approval and human skin sample collection and contributed to data interpretation. S.S., D.C. and D.M. performed human skin infections and SEM. P.P., I.T., T.B. and H.A. contributed to mouse experiments. N.N.T., C.C. and R.A. performed lipidomics analysis. W.C. and M.H. performed proteomics analyses. A.K., G.I. and S.J. supported cloning experiments. N.K. performed computational docking. L.-M.Z., M.L. and C.M. performed sterol analysis and antifungal susceptibility testing. H.L.T.M. contributed to bacterial experiments. A.C. provided clinical strains. T.P.-C. and K.K. wrote the manuscript with important contributions from A.E.-B. and input from all other authors.

## Funding

## Competing interests

The authors declare no competing interests.

## Additional information

**Extended data** is available for this paper at https://doi.org/10.1038/s41564-025-02189-z.

**Correspondence and requests for materials** should be addressed to Adelheid Elbe-Bürger or Karl Kuchler.

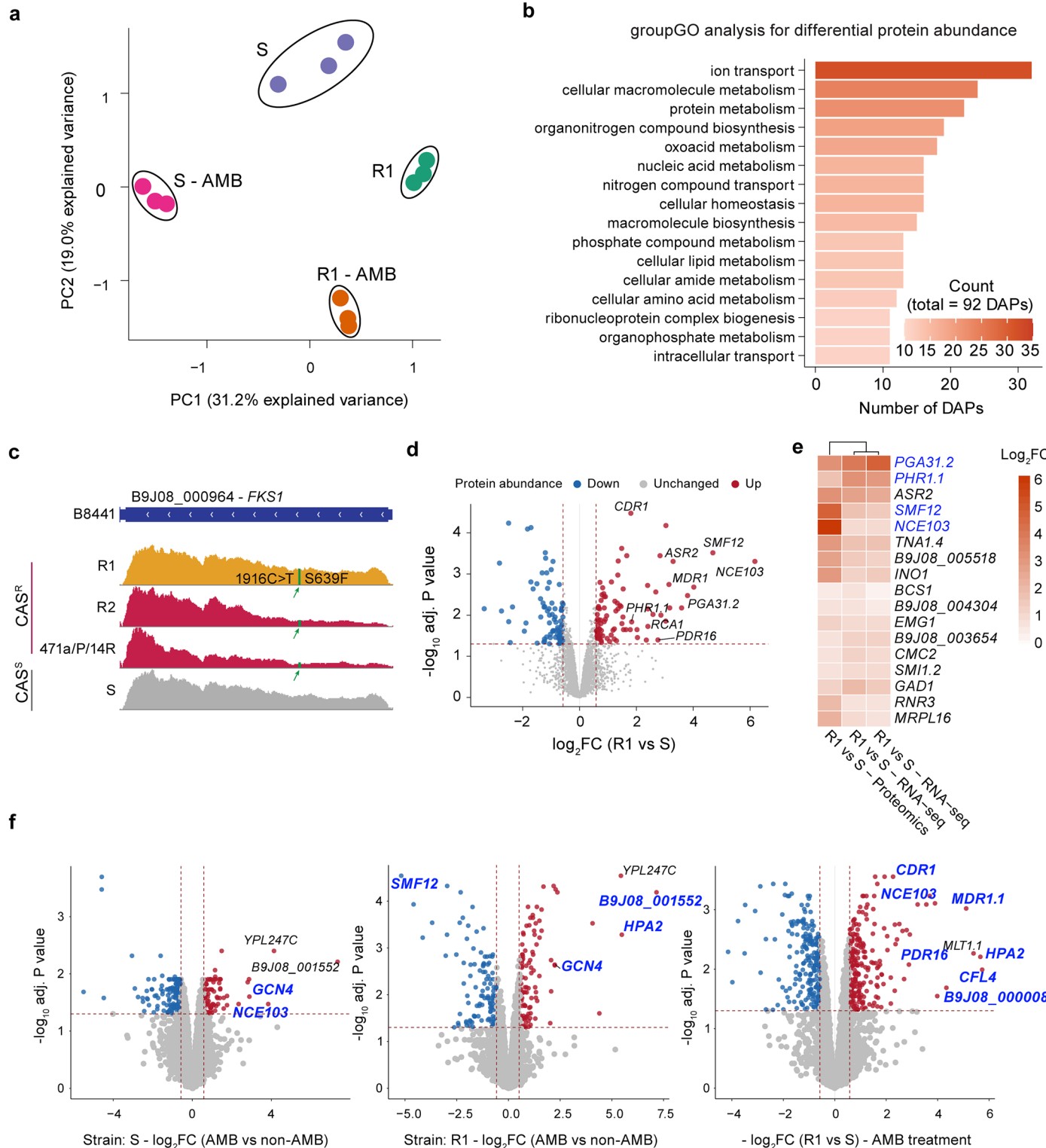

**Extended Data Fig. 1 | Proteomics reveals potential drug-resistant mechanisms. a.** PCA analysis of proteomics data from resistant and sensitive strains with / without AMB treatment (3 biological replicates for each strain). **b.** group GO analysis for differentially abundant proteins (DAPs) using the clusterProfiler package suggests that ion transport (32/92 DAPs) proteins are elevated in the resistant strain. **c.** The IGV identifies a hotspot S639F mutant variant (green) in Fks1 responsible for caspofungin resistance in R1, R2, 471a/P/14R[15]. **d.** Volcano plot showing comparative proteomic data between R1 and S strains. **e.** Integrated heatmap showing the overlap of DAPs with highly expressed genes[15,17] between resistant and sensitive *C. auris* strains. **f.** Volcano plots showing comparative proteomics data of R1 and S strains under AMB-treatment. Genes in blue bold were functionally validated in this study or by the available literature. Among them, *PDR16* and *NCE103* contribute to AMB[R]. Deletion of *HPA2*, *CFL4*, *SMF12*, *GCN4*, and *B9J08_000008* did not alter AMB MICs. Statistical analyses were performed using moderated two-sided t-statistics with the limma-trend method, followed by multiple test correction using the Benjamini–Hochberg (BH) method (**d**, **f**). Differential expression analysis of proteomics and RNA-seq datasets used a cut-off ± 0.58 for log₂FC and an adjusted p-value < 0.05 (**d-f**).

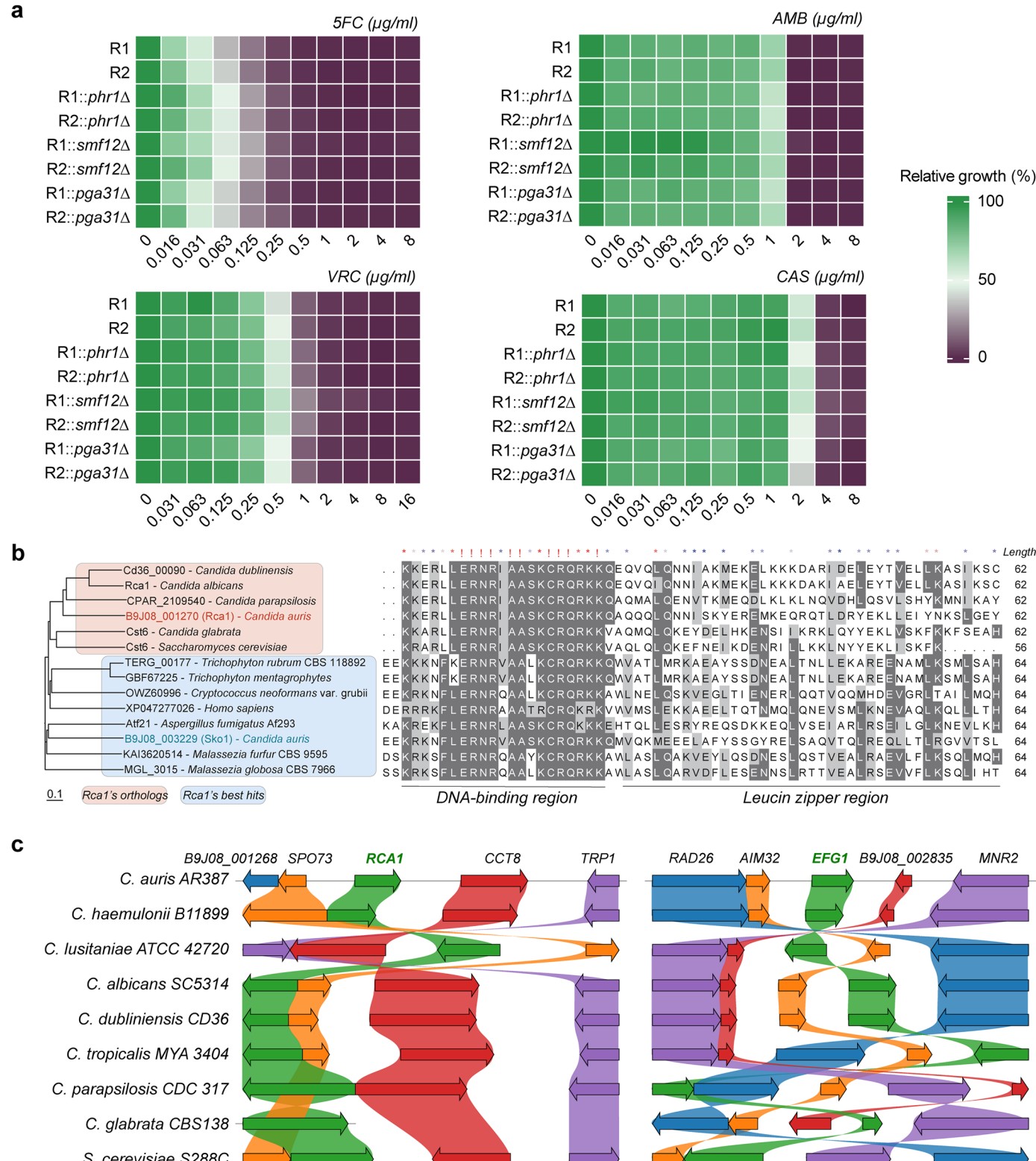

**Extended Data Fig. 2 | Deletion of genes encoding highly expressed cell surface proteins in resistant strains does not alter susceptibility to antifungal drugs.** **a.** Dose response assays for 5-fluorocytosine (5FC), amphotericin B (AMB), voriconazole (VRC) and caspofungin (CAS) using the CLSI protocol. **b.** Multiple sequence alignments (MSA) reveal the evolutionary conservation of the bZip domain in Rca1 across yeast species. *C. auris* genomic sequences of *RCA1* were subjected to *blastn* searches against the indicated species. A reciprocal *blastn* search was performed to confirm orthologous genes. bZip domains for MSA were predicted with ScanProsite tool (prosite.expasy.org/scanprosite). **c.** Synteny maps illustrate orthologous relationships of *RCA1* and *EFG1*, including their proximally neighboring genes in *C. auris* compared to other fungal pathogens. Orthologues and gene order information were retrieved from the *Candida* Gene Order Browser (cgob.ucd.ie) and fungi.ensembl.org.

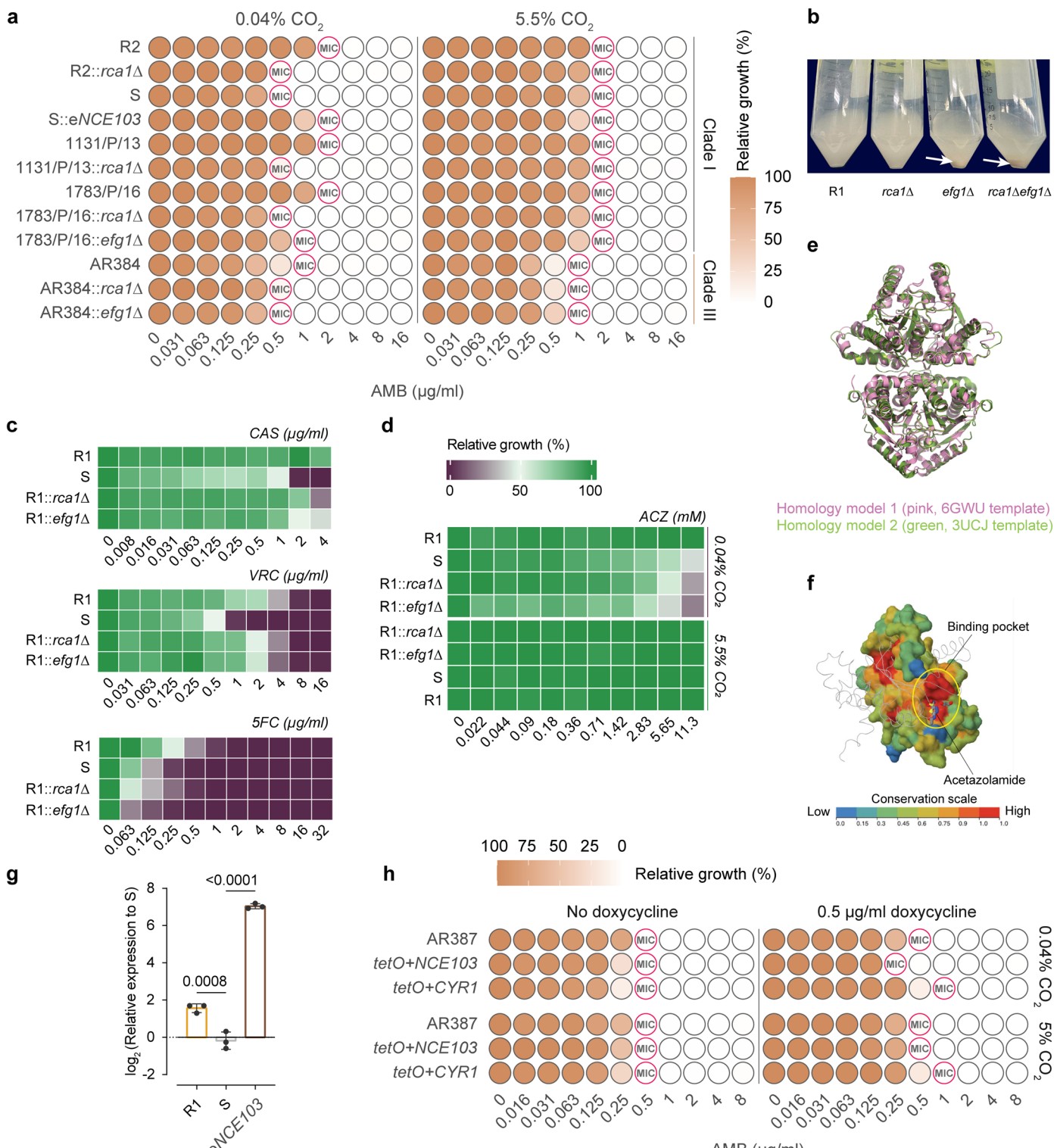

**Extended Data Fig. 3 | The Rca1-Nce103-Efg1 axis controls AMB susceptibility.**
**a**, **c**, **d**, **h**. CLSI-based dose-response assays (n = 3 biological replicates) of
*C. auris* strains for AMB (**Fig. a, h**), CAS, VRC, 5FC (**Fig. c**) and acetazolamide
(ACZ; **Fig. d**). **b**. Flocculation phenotypes of *C. auris* WT and mutants in dH$_2$O
after 15 minutes at room temperature. Cells were cultured overnight in liquid
YPD, washed twice with dH$_2$O, and adjusted to an OD$_{600nm}$ of 10 prior to the
assay. **e**. Overlay of Nce103 homology models generated using the coordinates
for *C. albicans* Nce103 (PDB: 6GWU), and the Coccomyxa (PDB: 3UCJ) carbonic

anhydrase crystal structures. **f**. The homology model of *C. auris* Nce103 shows
the conservation of the binding pocket for the ACZ carbonic anhydrase inhibitor.
**g**. Relative expression of *NCE103* to the WT S strain (n = 3 biological replicates).
mRNAs isolated from fungal cells in the exponential growth phase were
quantified with qPCR (*ACT1* as loading control). Statistical analysis used the one-
way ANOVA with Dunnett's multiple comparisons test, and p values < 0.05 are
shown. Each data point is shown, error bars indicate mean ± SD.

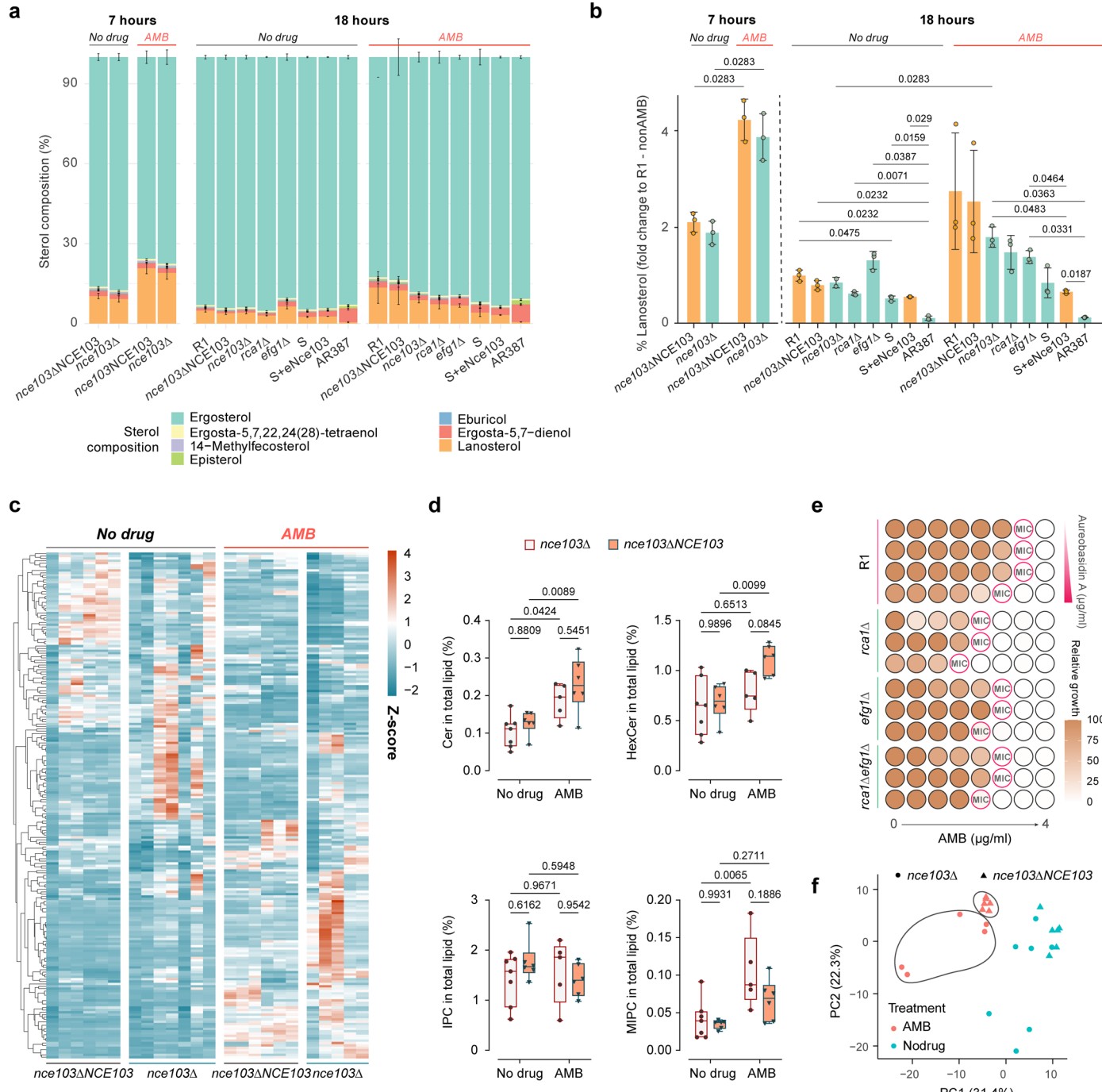

**Extended Data Fig. 4 | Deletion of *NCE103* has marginal effects on lipidomes.** Fungal cells in the exponential growth phase were cultured in RPMI medium for 5 or 16 hours, followed by treatment with 1 µg/ml AMB for an additional two hours. Cell pellets were then processed for lipid and sterol analysis. **a.** Sterol composition analysis with GC-MS from *C. auris* cells collected in the early and late logarithmic growth phases in RPMI liquid medium. **b.** Fold-change of lanosterol between mutants and WT strains with/without AMB treatment. Statistical comparisons were conducted between strains within each treatment, and separately between treatments for each strain. Statistics used two-sided Welch's t-test for each comparison with multiple test correction by Benjamini-Hochberg; exact p values < 0.05 are shown. Error bars indicated mean ± SD based on three biological replicates (A, B). **c, d, f.** Global lipid profiling of *C. auris* cultured in RPMI medium with or without AMB treatment (biological replicates from two independent experiments: n = 5 for AMB-*nce103*Δ; n = 7 for no-drug-*nce103*Δ;

n = 6 for other groups). Fungal cells were grown in RPMI for 5 hours prior to treatment with 1 µg/mL AMB for 2 hours, followed by lipidomic analysis (see Methods for details). Data represents the relative abundance of each lipid species relative to total lipids. **c.** Heatmap displays lipid species across samples, scaled as Z-scores. **d.** Proportion of sphingolipid classes within the total lipid pool. Certain classes were summed up data from their related species. Cer: ceramide; HexCer: hexosylceramide; IPC: inositol phosphorylceramide; MIPC: mannosylinositol phosphorylceramide. The box plot indicates the median and interquartile range, with whiskers representing minimum and maximum values. Two-way ANOVA with Šídák's correction was used for multiple comparisons across strains and treatments, and exact p values are shown. **e.** Susceptibility of *C. auris* to AMB and aureobasidin A (0–1 µg/mL) in RPMI at 37 °C. n = 2 biological replicates. **f.** The lipidomics dataset was subjected to PCA analysis using the *prcomp* function in R.

**a**

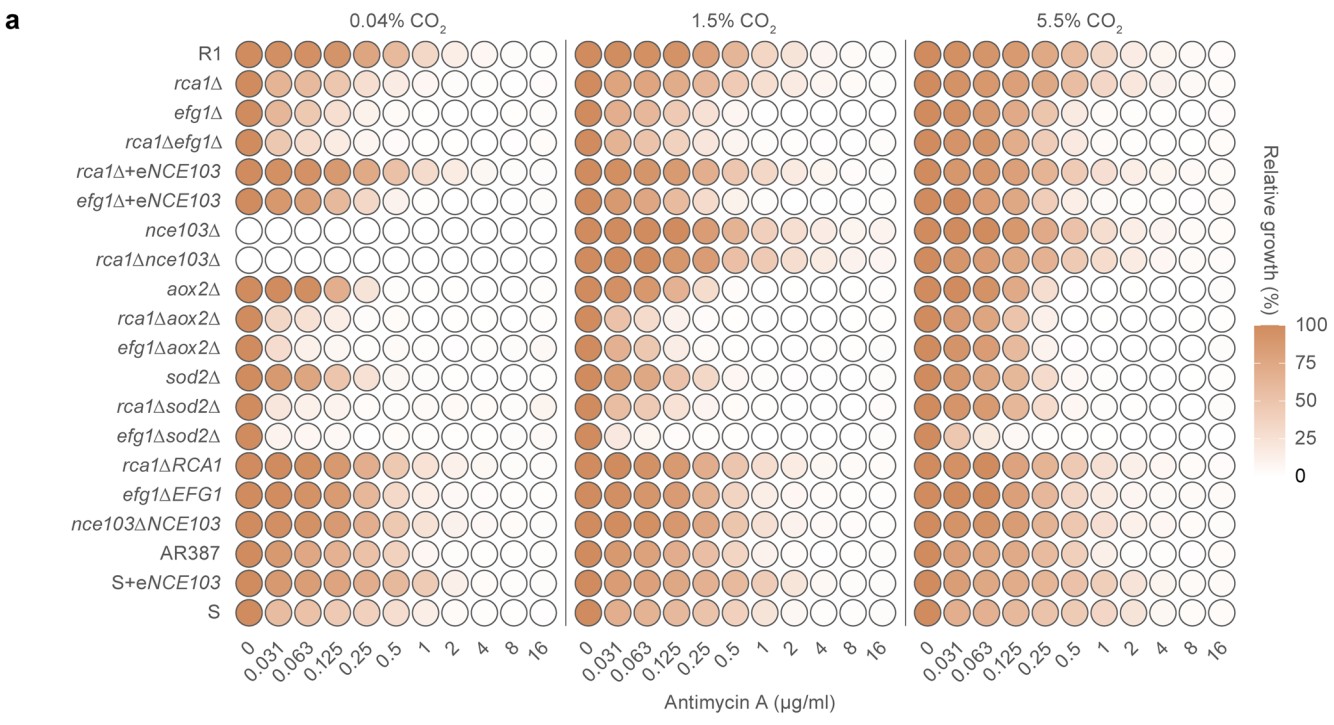

**b**

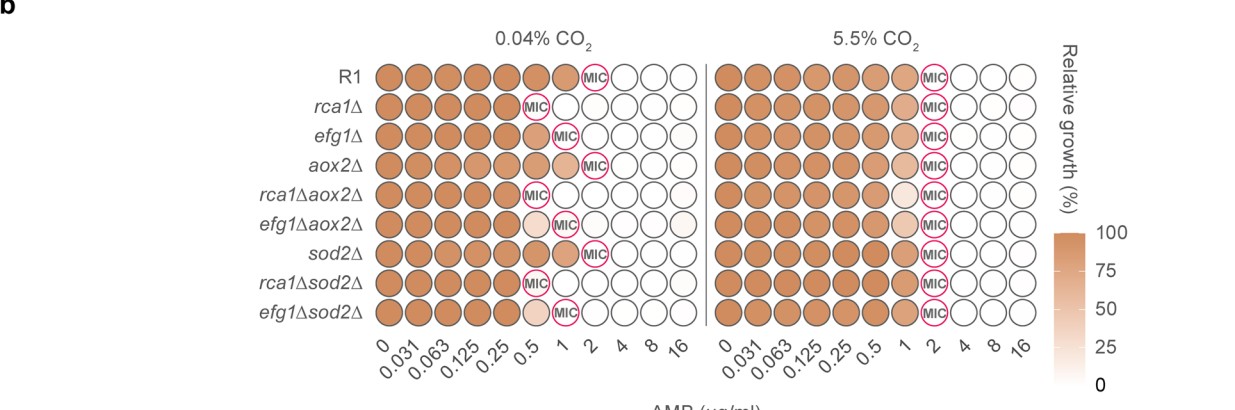

**c**

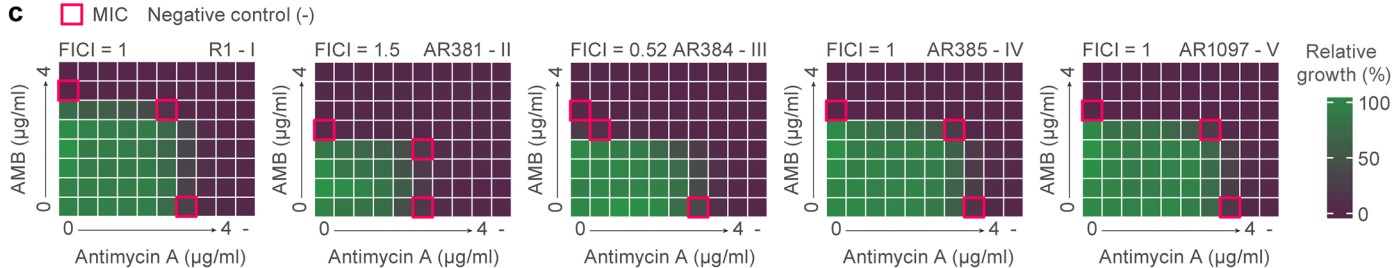

**Extended Data Fig. 5 | Mitochondrial function regulates AMB susceptibility.**
**a-b**. Dose-response assays of *C. auris* strains (three biological replicates) for antimycin A (**a**) and AMB (**b**) at indicated $CO_2$ concentrations using the CLSI protocol. The *nce103Δ* and *rca1Δ nce103Δ* mutants fail to grow in ambient air

( ~ 0.04% $CO_2$). **c**. Checkerboard assay for AMB and antimycin A for representative strains from clades I to V. FICI: fractional inhibitory concentration index. At least two biological replicates yielded similar results.

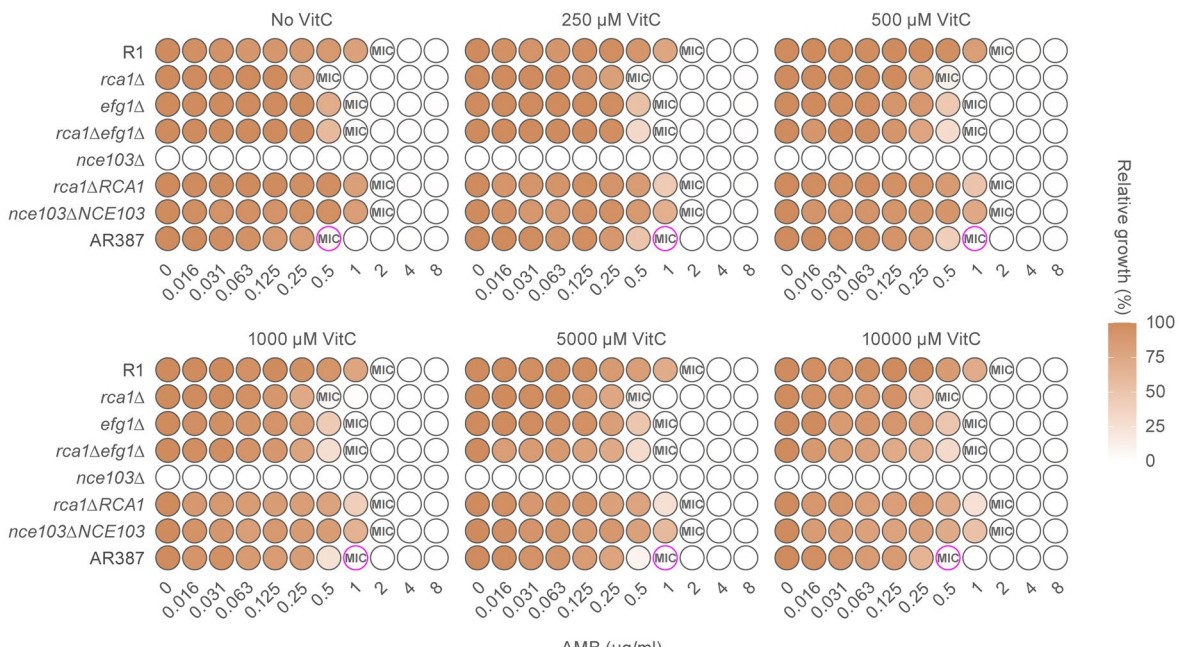

**Extended Data Fig. 6 | Vitamin C (VitC) only marginally restores AMB^MIC.** Dose response assays for AMB in RPMI containing VitC. The *nce103Δ* mutant fails to grow in ambient air ( ~ 0.04% $CO_2$); pink circles indicate the MIC reading where VitC exhibited protective effects (AR387). Two independent replicates yielded similar results.

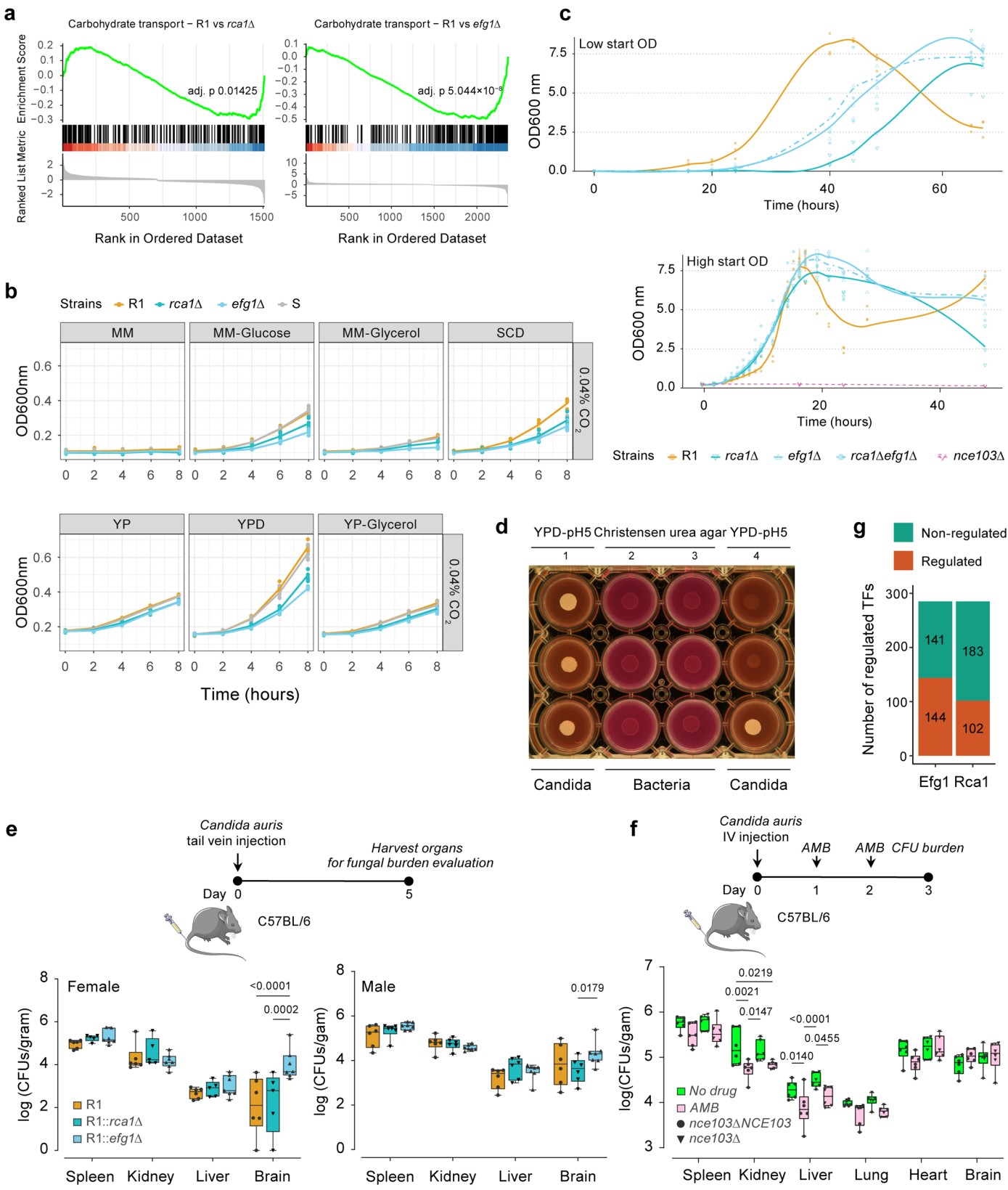

**Extended Data Fig. 7 | See next page for caption.**

**Extended Data Fig. 7 | Carbohydrate metabolism affects fitness of *C. auris* *rca1Δ and efg1Δ* mutants. a**. Gene set enrichment analysis (GSEA) using gseGO (*clusterProfiler*) shows upregulation of carbohydrate uptake upon deletion of *RCA1* or *EFG1*. Enrichment Scores (ES) indicate upregulated genes in null mutants. **b**. Growth kinetics within the first 8 hours of *C. auris* in different growth media. MM: minimal medium, SCD: synthetic complete dextrose medium; YP: yeast extract-peptone. Detailed media formulations are described in Supplementary Table 3. **c**. Standard growth curves of *C. auris* were recorded in RPMI medium after inoculation at either high ($OD_{600nm}$ = 0.2) or low ($OD_{600nm}$ = 0.001) starting cell densities. **d**. A representative picture illustrates the experimental setup for a bacterial-fungal co-culture assay in a 12-well plate. **e**. C57BL/6 mice were injected with $2\times10^6$ fungal cells through a lateral tail vein; fungal loads in organs were assessed on day five post-infection (n = 5 or 6 or 7 mice per group). **f**. C57BL/6 female mice (n = 6) were infected via lateral tail vein injection with fungal cells. AMB was administered intraperitoneally every 24 hours on day 1 and 2 post-infection. p-values were calculated by Two-way ANOVA with Tukey's multiple comparisons test and exact p values < 0.05 are shown (**e-f**). The box plot indicates the median and interquartile range, with whiskers representing minimum and maximum values; each data point is shown (**e-f**). **g**. Number of putative transcription factors regulated by the Rca1 and Efg1; data shows all DEGs with adjusted p-value < 0.05 from RNA-seq datasets of *rca1Δ* and *efg1Δ*.

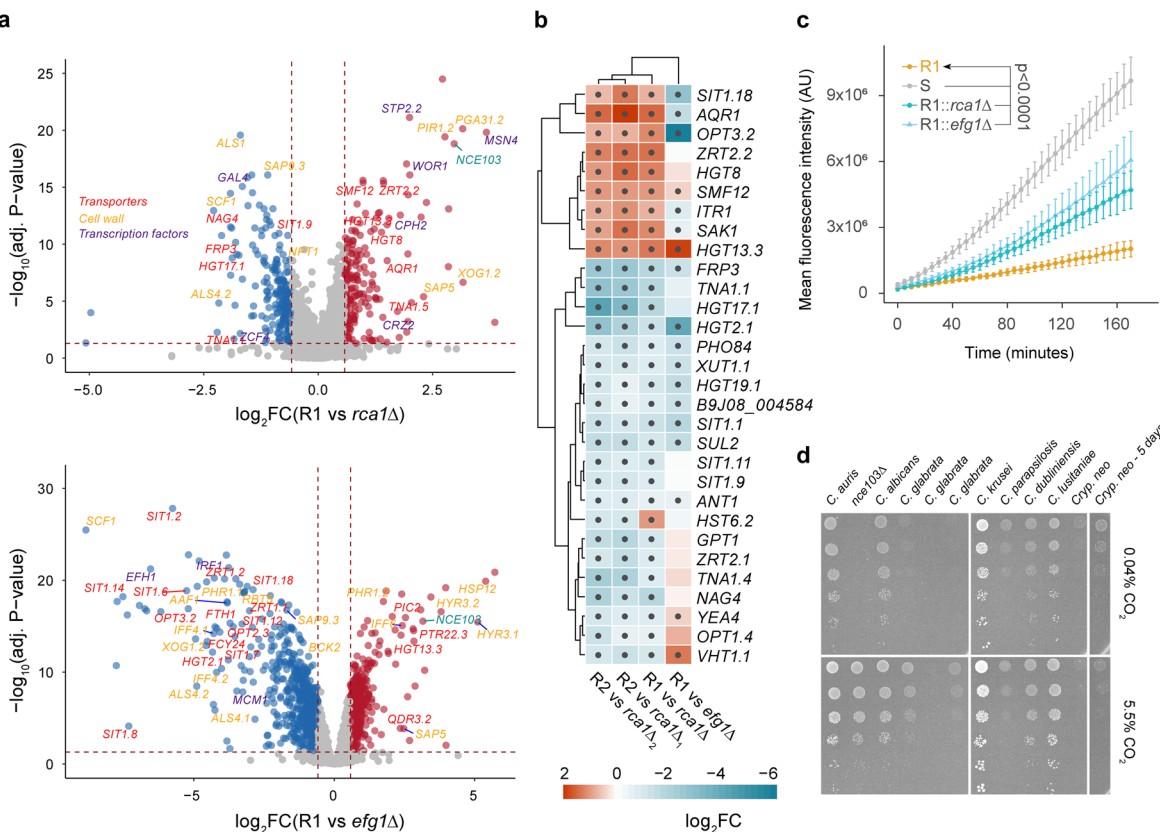

**Extended Data Fig. 8 | Differential expression of cell wall- and membrane-related genes in *rca1*Δ and *efg1*Δ mutants. a**. Volcano plots showing comparative transcriptomics data between R1, *rca1*Δ, *efg1*Δ strains. Dash lines indicate a cut-off ± 0.58 for log$_2$FC and adjusted p-value < 0.05. **b**. Heatmap showing the overlap of differentially expressed transporters between WT and mutant strains. Dots (●) indicate DEGs with adjusted p-value < 0.05. Differential expression analysis was conducted using EdgeR, including the two-sided quasi-likelihood F-test (glmQLFTest) approach (**a-b**). **c**. Membrane lipid permeability was assessed by an FDA uptake assay. Two-way ANOVA (two-sided) was performed for time × strain, followed by Tukey's multiple comparison tests using R1 as control group, with exact p values reported (based on data from 3 biological replicates in 1 independent experiment). Error bars indicate mean ± SD. See detail method in Supplementary Information. **d**. CO$_2$ supports fitness of most *Candida* pathogens but not of *Cryptococcus neoformans (Cryp. neo)*. Strains used in this experiment: R1::*nce103*Δ; *C. albicans* SC5314; *C. glabrata* ATCC 2001; *C. glabrata* KK01; *C. glabrata* KK04; *C. glabrata* KK05; *C. krusei* ATCC6258; *C. parapsilosis* ATCC22019; *C. dubliniensis* KK0908899; *C. lusitaniae* DSM 70102; *C. neoformans* NEQAS strain.

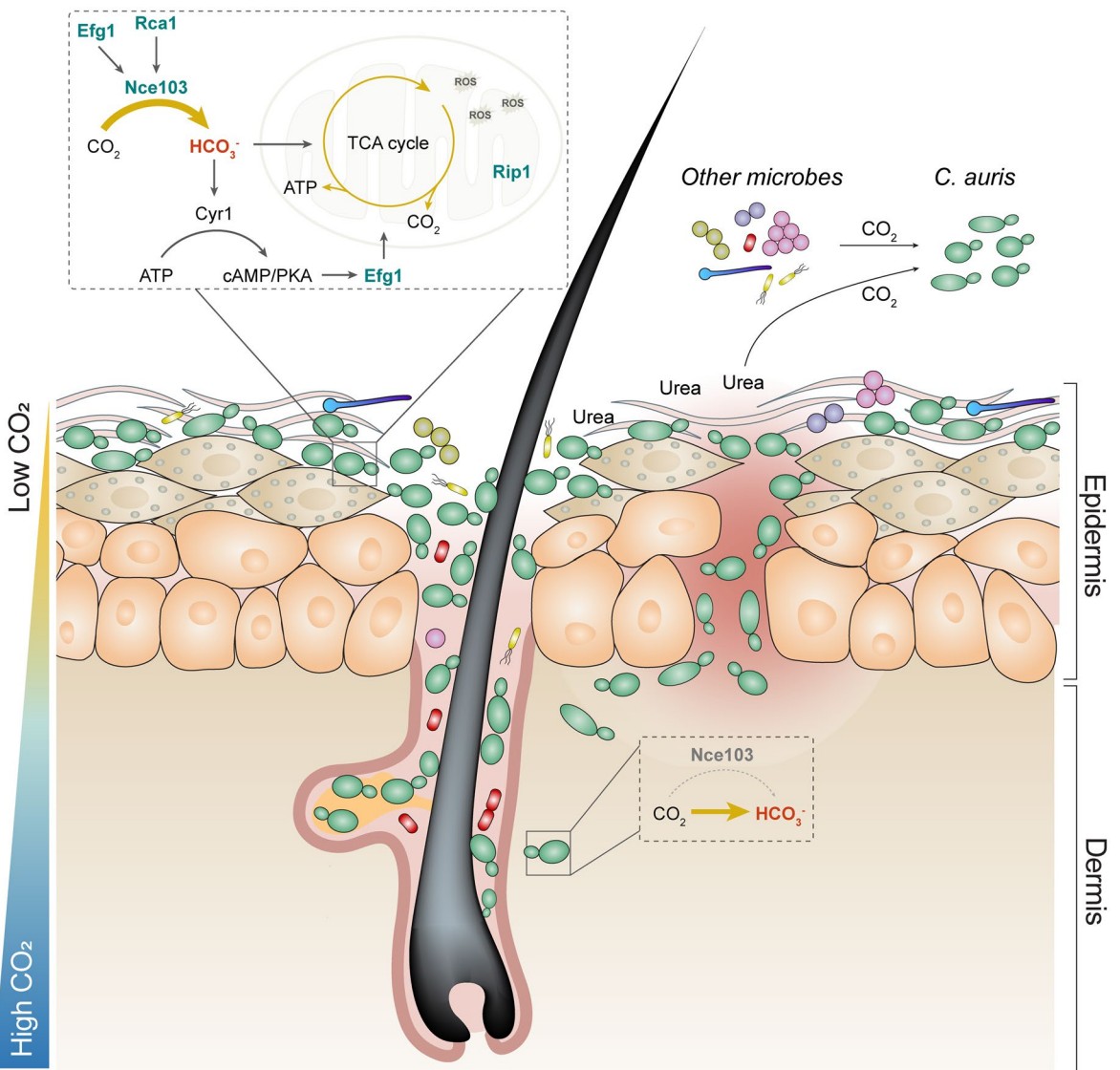

**Extended Data Fig. 9 | The skin niche allows for *Candida auris* colonization and growth.** When *C. auris* colonizes the nutrient-limited skin surface, it encounters ~0.04% $CO_2$ in ambient air. Under such low $CO_2$ levels, *C. auris* generates $HCO_3^-$ requiring the carbonic anhydrase Nce103, which in turn is regulated by Rca1 as well as Efg1. The resulting bicarbonate feeds mitochondrial energy metabolism through the tricarboxylic acid cycle (TCA) cycle, as well as signaling processes through the cAMP/PKA/Efg1 pathway. The CSP pathway controls drug resistance and skin colonization by sustaining mitochondrial functions, thus improving fitness and antifungal tolerance. Importantly, the microbiome containing urease-positive bacteria offers additional sources for $CO_2$ to sustain metabolism and *C. auris* persistence as well as fitness. Figure adapted with permission from ref. 53, Springer Nature Limited, and ref. 54, Springer Nature Limited.

# Reporting Summary

## Statistics

For all statistical analyses, confirm that the following items are present in the figure legend, table legend, main text, or Methods section.

| n/a | Confirmed | |
|---|---|---|
| ☐ | ☒ | The exact sample size (*n*) for each experimental group/condition, given as a discrete number and unit of measurement |
| ☐ | ☒ | A statement on whether measurements were taken from distinct samples or whether the same sample was measured repeatedly |
| ☐ | ☒ | The statistical test(s) used AND whether they are one- or two-sided *Only common tests should be described solely by name; describe more complex techniques in the Methods section.* |
| ☒ | ☐ | A description of all covariates tested |
| ☐ | ☒ | A description of any assumptions or corrections, such as tests of normality and adjustment for multiple comparisons |
| ☐ | ☒ | A full description of the statistical parameters including central tendency (e.g. means) or other basic estimates (e.g. regression coefficient) AND variation (e.g. standard deviation) or associated estimates of uncertainty (e.g. confidence intervals) |
| ☐ | ☒ | For null hypothesis testing, the test statistic (e.g. *F*, *t*, *r*) with confidence intervals, effect sizes, degrees of freedom and *P* value noted *Give P values as exact values whenever suitable.* |
| ☒ | ☐ | For Bayesian analysis, information on the choice of priors and Markov chain Monte Carlo settings |
| ☒ | ☐ | For hierarchical and complex designs, identification of the appropriate level for tests and full reporting of outcomes |
| ☐ | ☒ | Estimates of effect sizes (e.g. Cohen's *d*, Pearson's *r*), indicating how they were calculated |

*Our web collection on statistics for biologists contains articles on many of the points above.*

## Software and code

Policy information about availability of computer code

| Data collection | BD FACSDiva™ Software for Flow Cytometry. For lipidomics, LipidCreator (version 1.2.0), Analyst (version 1.7.2; AB Sciex), Skyline (version 22.2.0.312) and KNIME (version 5.2.5) were used. |
|---|---|
| Data analysis | All software and code used in this study are described in the Methods section, Supplementary Table 4, and Supplementary Information. Briefly, MaxQuant software (version 1.6.17.0), Cassiopeia_LFQ (https://doi.org/10.5281/zenodo.5758974), and the LIMMA R package were used for proteomics analysis. For RNA-seq analysis, the following tools were utilized: FastQC v0.11.9, NextGenMap v0.5.5, BEDtools v2.29.1, HTSeq v0.11.2, SAMtools v1.15.1, and EdgeR v3.40.2. R packages including msa v1.32, ape v5.7, and ggtree v3.9.1 were used for multiple sequence alignment and neighbor-joining tree construction. GraphPad Prism v9.0, along with R packages rstatix v0.7.2 and ggplot2 v3.4.2, were used for statistical analysis and data visualization. FlowJo v10.8 was used for flow cytometry data analysis. Code for RNA sequencing analysis and data integration is available at https://github.com/kakulab/CSP2024. |

For manuscripts utilizing custom algorithms or software that are central to the research but not yet described in published literature, software must be made available to editors and reviewers. We strongly encourage code deposition in a community repository (e.g. GitHub). See the Nature Portfolio guidelines for submitting code & software for further information.

## Data

Policy information about availability of data

All manuscripts must include a data availability statement. This statement should provide the following information, where applicable:
- Accession codes, unique identifiers, or web links for publicly available datasets
- A description of any restrictions on data availability
- For clinical datasets or third party data, please ensure that the statement adheres to our policy

---

Data Availability
The proteomics data was deposited to the ProteomeXchange Consortium via the PRIDE partner repository113 with the dataset identifier PXD048342. RNA-seq data are available from the Gene Expression Omnibus (GEO) database with the accession number GSE253332. Lipidomics datasets are provided in supplementary data. Data used for figure generation are deposited as a Source Data file. Correspondence and requests for materials should be addressed to Karl Kuchler (kuchlerkarl1@gmail.com).
Code Availability
The code for RNA-seq analysis workflow was deposited on Github available through https://github.com/kakulab/CSP2024. The workflow for processing MaxQuant output tables was deposited at https://doi.org/10.5281/zenodo.5758974.

---

## Research involving human participants, their data, or biological material

Policy information about studies with human participants or human data. See also policy information about sex, gender (identity/presentation), and sexual orientation and race, ethnicity and racism.

| | |
|---|---|
| Reporting on sex and gender | Human skin samples were obtained from healthy adult female donors. |
| Reporting on race, ethnicity, or other socially relevant groupings | Not applicable. |
| Population characteristics | Healthy adult female donors |
| Recruitment | Participants were recruited from the patient pool of the Department of Plastic and Reconstructive Surgery, Medical University of Vienna, Währinger Gürtel 18-20, 1090 Vienna, Austria. Before their surgery, all participants were informed about the relevance of this study by the investigating physician of the Department of Plastic and Reconstructive Surgery. There were no additional risks or burdens for the study participants. Human skin biopsy samples were obtained from three available donors. Donors were included based on availability, without random selection or specific covariate criteria (e.g., age, sex, medical history). Given the small number and the use of samples solely for ex vivo colonization assays, potential selection bias is not expected to impact the study outcomes. |
| Ethics oversight | Abdominal human skin samples were obtained from anonymous healthy adult female donors with approved consent following the Declaration of Helsinki and ethics committee approvals of the Medical University of Vienna (ECS 1969/2021). |

Note that full information on the approval of the study protocol must also be provided in the manuscript.

# Field-specific reporting

Please select the one below that is the best fit for your research. If you are not sure, read the appropriate sections before making your selection.

☒ Life sciences ☐ Behavioural & social sciences ☐ Ecological, evolutionary & environmental sciences

For a reference copy of the document with all sections, see nature.com/documents/nr-reporting-summary-flat.pdf

# Life sciences study design

All studies must disclose on these points even when the disclosure is negative.

| | |
|---|---|
| Sample size | Sample sizes of experimental groups were chosen based on community standards (e.g., 3 experimental repeats if not otherwise specified; at least 5 mice per group were used for in vivo experiments). |
| Data exclusions | No data were excluded from analyses workflows |
| Replication | Replication in the form of biological repeats is reported in the manuscript. For the study of human skin colonization, three skin biopsy samples were obtained from each of three different donors. Mouse experiments were replicated independently at least twice, and results were pooled. For in vitro experiments, at least three biological replicates were performed unless otherwise specified. In some cases, two biological or independent replicates were conducted instead of three due to the inclusion of multiple independent clinical strains; in such cases, conclusions were also confirmed by additional experiments (e.g., genetic ablation in addition to phenotypic assays). |

| Randomization | Female mice were randomly assigned to study groups. Skin biopsy samples and mice were randomly selected for infection with wild-type or mutant fungal strains. |
| --- | --- |
| Blinding | Blinding was not applied during allocation, outcome assessment, data collection, or analysis, as no potential sources of bias were identified. |

# Reporting for specific materials, systems and methods

We require information from authors about some types of materials, experimental systems and methods used in many studies. Here, indicate whether each material, system or method listed is relevant to your study. If you are not sure if a list item applies to your research, read the appropriate section before selecting a response.

## Materials & experimental systems

| n/a | Involved in the study |
| --- | --- |
| ☒ | Antibodies |
| ☒ | Eukaryotic cell lines |
| ☒ | Palaeontology and archaeology |
| ☐ | ☒ Animals and other organisms |
| ☒ | Clinical data |
| ☒ | Dual use research of concern |
| ☒ | Plants |

## Methods

| n/a | Involved in the study |
| --- | --- |
| ☒ | ChIP-seq |
| ☐ | ☒ Flow cytometry |
| ☒ | MRI-based neuroimaging |

## Animals and other research organisms

Policy information about studies involving animals; ARRIVE guidelines recommended for reporting animal research, and Sex and Gender in Research

| Laboratory animals | Adult wild-type female C57BL/6J mice (Mus musculus) were housed in specific pathogen-free conditions, with controlled temperature (20-22 °C) and humidity (45-65%), in a 12-h light/dark cycle at the Max Perutz Labs Vienna. Mice breeding and maintenance was in accordance with ethical animal license protocols complying with the current Austrian law. Animals used for experiments were wild-type C57BL/6J mice aged 8 to 14 weeks at the start of experiments. |
| --- | --- |
| Wild animals | No wild animals were used. |
| Reporting on sex | All skin experiments were performed with female mice. Tail vein infections were performed in both female and male mice. |
| Field-collected samples | No field-collected samples were used. |
| Ethics oversight | Animal experiments adhered to ethical approval from the ethics committee of the Medical University of Vienna and the Federal Ministry of Science and Research, Vienna, Austria (BMBWF-66.009/0436-V/3b/2019). |

Note that full information on the approval of the study protocol must also be provided in the manuscript.

## Plants

| Seed stocks | Not applied |
| --- | --- |
| Novel plant genotypes | Not applied |
| Authentication | Not applied |

# Flow Cytometry

## Plots

Confirm that:

☒ The axis labels state the marker and fluorochrome used (e.g. CD4-FITC).

☐ The axis scales are clearly visible. Include numbers along axes only for bottom left plot of group (a 'group' is an analysis of identical markers).

☐ All plots are contour plots with outliers or pseudocolor plots.

☐ A numerical value for number of cells or percentage (with statistics) is provided.

## Methodology

| | |
|---|---|
| Sample preparation | Sample preparations are described in the Methods section. Briefly, fungal cells were collected, washed before subjecting to flow cytometry analysis. |
| Instrument | BD LSRFortessa™ Cell Analyzer |
| Software | BD FACSDiva™ Software, FlowJo™ Software 10.8 |
| Cell population abundance | Approximately 80-95% of fungal cells were obtained after excluding cell debris. |
| Gating strategy | Fungal cells were gated using FCS-A and SSC-A parameters, followed by single-cell gating using FCS-A and FCS-H. As only a single color was used, complex gating strategies were not required. |

☐ Tick this box to confirm that a figure exemplifying the gating strategy is provided in the Supplementary Information.

