## [Peer Review File · Nature Microbiology]

Candida auris skin tropism and antifungal resistance are mediated by carbonic anhydrase Nce103

Corresponding Author: Professor Karl Kuchler

Version 0:

Reviewer comments:

Reviewer #1

(Remarks to the Author)

The study by Cahn TP, et al. is focused on CO₂ sensing mechanism of *C. auris* that is unique in promoting advantage in the pathogen for acquiring antifungal resistance, colonization and how the bacterial skin colonizers can promote the *C. auris* growth by CO₂ production. It is very interesting to note that *C. auris* utilizes carbonic sensing pathways to gain fitness for resistance and colonization; 1.) Antifungal resistance due to CO₂ sensing, 2.) Tropisms of *C. auris* in skin tissue due to CO₂ sensing mechanism and 3.) How the bacterial skin colonizers promote *C. auris* growth by CO₂ production. Understanding the mechanisms governing antifungal resistance and skin colonization is very important to understand the pathogenesis of this emerging fungal pathogen. This study for the first time provided a unique mechanisms. Although it's very interesting to look into the CO₂ sensing in *C. auris*, performing some additional experiments would enhance the study.

Major comments

- 1) Currently, the author does not have data to show that Rca1 regulates Nce103, and it acts on Efg1 to control antifungal susceptibility. Performing additional experiments to establish a connection between NCE103 and Rca1 and Efg1 mechanism in *C. auris* is important.
- 2) Is CO₂ sensing mechanism contributing fitness unique to *C. auris* unique or similar phenomenon observed in other *Candida* spp as well such as *C. albicans*?
- 3) The mutant strains appear to have growth defects in the MM and YPD Fig. 4A. In this case the CFU data from Fig. 4D and 4E is difficult to interpret. Performing a traditional growth curve assay using RPMI-MOPS to show that the growth defects in these mutants are not significant to interfere in the fungal burden from the murine skin would be useful.
- 4) Unclear if AmpB sensitive isolates have genetic mutation in either Rca1, Nce103 and/or Efg1? Is it possible to perform a correlation analysis between the genetic mutation harbored in Rca1, Nce103 or Efg1 and the AmpB susceptibility?

Minor comments

- 1.) Line 150 and 376... unique NCE103 gene in *C. auris*. The Nce103 orthologs are present in other *Candida* spp. So, the Nce103 gene is not unique in *C. auris*, and this statement could be reframed throughout the manuscript. Additionally, synteny schema of Nce103 gene in *C. auris* with other *Candida* spp. will give more insights into the gene level homology.
- 2.) Some of the mutants are aggregative Fig. S3G. In this case, it is difficult to enumerate the infection dose, and to quantify the recovered fungal burden by CFU plating. Consider sonicating the fungal cells before infection and CFU plating for precise quantification (Ref: Bing J, 2024 Nat Commun (2024); DOI: 10.1038/s41467-024-46786-8).
- 3.) The experiments with the bacterial co-culture with the *C. auris* strains are reasonable. The urease positive bacterial skin colonizers promote *C. auris* growth by CO₂ production. But the effect of ammonia and PH change in the skin microenvironment may also influence the *C. auris* growth. Additional in vitro or in vivo assay to show the net effect on the *C. auris* growth by these urease positive bacterial skin colonizers would be interesting.
- 4.) Fig. 2B. The susceptibility of these clinical isolates to AmpB is missing and needs to be discussed in the results.
- 5.) The flocculation phenotype of *rca1Δefg1Δ* mutants in the Fig. S3G is not shown. If the cells are aggregative, they could be separated by sonication for MIC assay (Ref: Bing J, 2024 Nat Commun (2024); DOI: 10.1038/s41467-024-46786-8).
- 6.) Line 109 to 114 is unclear. The results do not match the figures. For instance, *rca1Δefg1Δ* is only in Fig. 2A but, Fig. S3A was cited in line 112 to 114.
- 7.) Some of the figure legends are incomplete (Fig. S7D).

(Remarks on code availability)

Reviewer #2

(Remarks to the Author)

Here the authors combined proteomics and transcriptome approaches and identified the carbonic anhydrase Nce103 and its regulators Rca1 and Efg1 as components of the carbonic sensing pathway that contributes to *Candida auris* AMB resistance (modest 2 to 4 fold increase). This is a quite exciting finding as it is different from the known ergosterol biosynthetic pathway. As expected, high level of CO₂, including CO₂ released by bacterial ureases, restored AMB tolerance in the nce103 mutant. The data to support NCE103's role in the modestly increased AMB resistance is strong. The authors further showed that CSP contributes to skin colonization, and connected CSP with mitochondrial dysfunction and ROS generation. The evidence to support later conclusions was not as convincing.

General concerns:

1. I agree that Rca1 and Efg1 regulate NCE103. However, as the authors stated, these regulators likely control many other factors/pathways in addition to the carbonic anhydrase, the enzyme that converts CO₂ to bicarbonate. It is, therefore, not accurate to attribute the phenotypes of Rca1 and Efg1 mutations to the CSP pathway in some of the occasions. For example, Figure 4D: the competitive index for the rca1 mutant or the efg1 mutant during skin infection is very low, particularly for the efg1 mutant. In contrast, the difference between the control and the nce103 mutant is not even statistically significant. So the role of CSP's specific contribution to the intradermal infection is inconclusive at the best. There are other similar experiments/conclusions in this study that should be modified.
2. The role of NCE103 and its homologs in fungal growth in ambient air has been demonstrated previously. The importance of bicarbonate for metabolism is also known. The fact that high levels of CO₂ can restore nce103 mutant phenotype is expected. This is also consistent with no so drastically reduced fungal burden of the nce103 mutant in the skin infection models (Figure 4D-E). This is also consistent with the lack of virulence phenotype of cryptococcal carbonic anhydrase mutants. Thus, it feels a bit forced for the authors to conclude that CSP is critical for skin infection.
3. The authors showed that the sterol composition is similar between AMB resistant and sensitivity strains. It is recently shown that the inositolphosphorylceramide level has significant impact on AMB susceptibility. Given that both ergosterol and IPC are important components of fungal membrane and they affect various transporters and susceptibility to various drugs, the potential impact of CSP on both lipid components should be examined. The hypothesis that CSP may change the composition of membrane, which may indirectly affect mitochondrial function, should be considered. The evidence that CSP disruption impairs mitochondrial functions is not convincing based on the ROS evidence. The sensitivity to AA could also be an indirect effect because of altered membrane.
4. One thing that is missing is the experiment to determine if AMB therapy will make a difference in vivo when the Nce103 is blocked.

Specific concerns:

1. Figure 3E: why growth of the rca1, the efg1, and the double mutant is the best at intermediate AMB concentration with a relatively lower dose of Antimycin A?
2. Figure S5A: (1) Both S strain and nce103 mutant in R1 strain background showed restored resistance to AMB at 5.5% CO₂ level. However, efg1 sensitivity to AMB is not restored by CO₂, even with overexpression of NCE103. (2) The AMB sensitivity of aox2 and sod2 mutants did not change much in the presence of high levels of CO₂, arguing against the idea that they are affected by the bicarbonate levels.
3. It would be useful to survey the AMB resistant isolates used in Figure 2C to examine how common NCE103 is involved in AMB resistance (e.g. by qPCR) as RCA1 and EFG1 ultimately need to alter the function of NCE103 for their CSP specific effect. Given that Nce103 is essential in ambient air, the likely impact is on its expression level.
4. In Table S1, CLSI and EUCAST methods generated similar MICs (mostly within 2-4 fold). However, there are some large differences between the two methods. For example, MICs for VRC of the sensitive strain showed greater than 32 fold difference. Similarly, MICs for CAS of the strains AR285 and AR1097 are more than 32 fold different. Explanation?
5. Figure S3C: The authors stated that "both rca1Δ and efg1Δ mutants showed slightly increased susceptibility to 5-fluorocytosine (5FC), although not significant to caspofungin (CAS) or voriconazole (VRC) (Fig. S3C)". The CAS test did not reach the concentration that causes significant reduced growth for R1, and thus it is hard to tell. However, it looks like that there should be at least 2 or 4 fold difference with rca1 or efg1 mutations. Based on the graph of relative growth, both rca1Δ and efg1Δ mutants showed dramatically increased sensitivity to 5FC (8 fold for efg1 and 4 fold for rca1). This is more than they observed with AMB, the focus of this story. Are these changes in drug resistance/sensitivity related to bicarbonate levels?
6. Ptc2 functions in sensing high levels of CO₂ to control fungal response to high levels of CO₂ (like morphogenesis in *Candida albicans*). It is obviously not the same as the CSP pathway where conversion of CO₂ to bicarbonate is critical when the CO₂ level is very low. It is unclear how including this adds to this study.
7. Have the authors tested eNCE103 in the R1 strain without other mutations as a control for all the eNCE103 strains in various mutant backgrounds?
8. "Although there was no significant enhancement of AMB MIC in mutants, strain AR387 showed a 2-fold increase when treated with 250-1000 μM VitC. These findings suggest that, in addition to the ROS response, mitochondrial functions may control downstream effects resulting from CSP disruption." It is not clear to me how the evidence leads to the conclusion that ROS is an important factor in CSP's impact on mitochondrial function.
9. Figure 4A. Have the authors tested preculturing the strains in high level of CO₂ before the growth assay? Will such pre-adaptation reduce the difference in MM media?
10. Figure 4D-E: Could authors clarify if the nce103ΔNCE103 used in animal models is a complemented strain with NCE103 driven by its native promoter or it is an NCE103 overexpression strain in the nce103Δ mutant background?

(Remarks on code availability)

They used standard pipelines for proteomics and RNA seq analyses.

Reviewer #3

(Remarks to the Author)

The manuscript investigates the role of the carbonic sensing pathway (CSP) in antifungal resistance and fitness of *Candida auris*, focusing on the carbonic anhydrase Nce103 and its regulators Rca1 and Efg1. The authors provide compelling evidence that CSP is crucial for amphotericin B resistance (AMBR), mitochondrial function, and fungal adaptation to skin environments. They propose that bacterial urease-derived CO₂ may enhance *C. auris* colonization, suggesting potential therapeutic strategies targeting CSP components.

The experimental design is robust, combining transcriptomics, proteomics, gene deletion, pharmacological inhibition, and in vivo/in vitro models. The findings are largely supported by data, and conclusions appear justified. The manuscript does not have apparent fatal flaws that would prohibit publication, though some aspects require additional clarification. In particular, the proteomics and RNA-seq experiment section would benefit from further details about the following:

- What statistical test was employed for the differential expression analysis?
- Was the data normalised and/or log transformed before statistics? If yes, what method of normalisation was used?
- After splitting the data into single CV and processing each one separately through MaxQuant, how were the results combined and normalised?
- Add a reference for the contaminants database.
- The actual code for proteomics data analysis is not disclosed in github or elsewhere.
- Although figure S1 mentions biological triplicates, more details on replicates for proteomics and RNA-seq in the methods section would be helpful.

I was not able to access the data in PRIDE as the provided username and password did not work.

On line 76, I suggest you remove the statement that "proteomics may identify key genes" and leave it at key pathways.

On line 454, change wording to "Fungal cells were grown..."

In fig. 1B., the authors have used a picture of a randomly-selected Fusion Lumos mass spectrometer. It is advisable (but not strictly necessary) to use a picture of the machine that was actually used in the experiments, that is the Exploris 480.

Other potential points to address may include:

- Are the fitness defects of *rca1*Δ and *efg1*Δ mutants fully attributable to mitochondrial dysfunction?
- How do CSP-related AMB resistance mechanisms compare to those in clinical isolates with known resistance mutations?
- Could the CSP be targeted in therapeutic interventions without disrupting host CO₂ metabolism?
- Were urease-negative bacterial strains tested for other factors that might affect fungal growth?

In conclusion, the study presents novel insights into the CSP's role in AMBR and fungal colonization. The integrated proteo-transcriptomics approach is an appropriate preliminary screen, and the in vivo skin colonization model strengthens translational relevance. The conclusions align well with the data and the claim that CSP can be a therapeutic target is reasonable, but potential off-target effects of inhibitors should be acknowledged. The study is strong and has substantial merit for publication after addressing the minor points as mentioned above.

(Remarks on code availability)

I have not reviewed the RNA-seq code because this is not my area of expertise. I would be happy to review the proteomics data analysis code if the authors provide it. I would also be happy to review the PRIDE submission if you give me a working username and password.

Decision Letter:

18th March 2025

Dear Karl,

Thank you for your patience while your manuscript "The carbonic anhydrase Nce103 is required for fitness and skin tropism of *Candida auris*" was under peer-review at Nature Microbiology. It has now been seen by 3 referees, whose expertise and comments you will find at the end of this email. Although they find your work of some potential interest, they have raised a number of concerns that will need to be addressed before we can consider publication of the work in Nature Microbiology.

In particular, referee #1 says the connection between NCE103 and Rca1 and Efg1 mechanism in *C. auris* should be further analyzed. Further, the referee asks if CO₂ sensing is unique for *C. auris* or common for other *Candida* spp, and suggests to perform a growth curve assay using RPMI-MOPS to assess growth defects. Referee #1 also asks to perform a correlation analysis between the genetic mutation harbored in Rca1, Nce103 or Efg1 and the AmpB susceptibility. Also, the referee suggests the addition of an in vitro or in vivo assay to show the net effect on the *C. auris* growth by these urease positive bacterial skin colonizers. Referee #2 says that the connections between phenotypes of Rca1 and Efg1 and CSP are not supported, and says that the role of CSP for skin infection is overstated. Furthermore, the referee says the effects of ergosterol and IPC on CSP should be assessed. Referee #2 also asks for validation of CSP affecting the mitochondrial function. Referee #2 furthermore states that it is unclear if AMB therapy will have effects in vivo when Nce103 is blocked. Referee #3 asks for clarifications on the methods, and on the statistics used. Further, the referee says that the actual code for proteomics data analysis is not disclosed in github, and that they could not access the PRIDE database. Editorially, we will require all referee comments and concerns to be addressed in full.

Should further experimental data allow you to address these criticisms, we would be happy to look at a revised manuscript.

Please include a data availability statement as a separate section after Methods but before references, under the heading "Data Availability". This section should inform readers about the availability of the data used to support the conclusions of your study. This information includes accession codes to public repositories (data banks for protein, DNA or RNA sequences, microarray, proteomics data etc...), references to source data published alongside the paper, unique identifiers such as URLs to data repository entries, or data set DOIs, and any other statement about data availability. At a minimum, you should include the following statement: "The data that support the findings of this study are available from the corresponding author upon request", mentioning any restrictions on availability. If DOIs are provided, we also strongly encourage including these in the Reference list (authors, title, publisher (repository name), identifier, year). For more guidance on how to write this section please see: <http://www.nature.com/authors/policies/data/data-availability-statements-data-citations.pdf>

* If you have not done so already we suggest that you begin to revise your manuscript so that it conforms to our Article format instructions at <http://www.nature.com/nmicrobiol/info/final-submission>. Refer also to any guidelines provided in this letter.

When submitting the revised version of your manuscript, please pay close attention to our [href="https://www.nature.com/nature-portfolio/editorial-policies/image-integrity">Digital Image Integrity Guidelines.](https://www.nature.com/nature-portfolio/editorial-policies/image-integrity) and to the following points below:

EXTENDED DATA FIGURES

Link Redacted

Note: This url links to your confidential homepage and associated information about manuscripts you may have submitted or be reviewing for us. If you wish to forward this e-mail to co-authors, please delete this link to your homepage first.

Nature Microbiology is committed to improving transparency in authorship. As part of our efforts in this direction, we are now requesting that all authors identified as 'corresponding author' on published papers create and link their Open Researcher and Contributor Identifier (ORCID) with their account on the Manuscript Tracking System (MTS), prior to acceptance. This applies to primary research papers only. ORCID helps the scientific community achieve unambiguous attribution of all scholarly contributions. You can create and link your ORCID from the home page of the MTS by clicking on 'Modify my Springer Nature account'. For more information please visit www.springernature.com/orcid.

If you wish to submit a suitably revised manuscript we would hope to receive it within 6 months. If you cannot send it within this time, please let us know.

Yours sincerely,

Reviewer Expertise:

Referee #1: *C. auris*, skin infections

Referee #2: Fungal pathogenesis, fungal genetics, antifungal drugs

Referee #3: Proteomics, mass Spectrometry

Reviewer Comments:

Reviewer #1 (Remarks to the Author):

The study by Cahn TP, et al. is focused on CO₂ sensing mechanism of *C. auris* that is unique in promoting advantage in the pathogen for acquiring antifungal resistance, colonization and how the bacterial skin colonizers can promote the *C. auris* growth by CO₂ production. It is very interesting to note that *C. auris* utilizes carbonic sensing pathways to gain fitness for resistance and colonization; 1.) Antifungal resistance due to CO₂ sensing, 2.) Tropisms of *C. auris* in skin tissue due to CO₂ sensing mechanism and 3.) How the bacterial skin colonizers promote *C. auris* growth by CO₂ production. Understanding the mechanisms governing antifungal resistance and skin colonization is very important to understand the pathogenesis of this emerging fungal pathogen. This study for the first time provided a unique mechanisms. Although it's very interesting to look into the CO₂ sensing in *C. auris*, performing some additional experiments would enhance the study.

Major comments

- 1) Currently, the author does not have data to show that Rca1 regulates Nce103, and it acts on Efg1 to control antifungal susceptibility. Performing additional experiments to establish a connection between NCE103 and Rca1 and Efg1 mechanism in *C. auris* is important.
- 2) Is CO₂ sensing mechanism contributing fitness unique to *C. auris* unique or similar phenomenon observed in other *Candida* spp as well such as *C. albicans*?
- 3) The mutant strains appear to have growth defects in the MM and YPD Fig. 4A. In this case the CFU data from Fig. 4D and 4E is difficult to interpret. Performing a traditional growth curve assay using RPMI-MOPS to show that the growth defects in these mutants are not significant to interfere in the fungal burden from the murine skin would be useful.
- 4) Unclear if AmpB sensitive isolates have genetic mutation in either Rca1, Nce103 and/or Efg1? Is it possible to perform a correlation analysis between the genetic mutation harbored in Rca1, Nce103 or Efg1 and the AmpB susceptibility?

Minor comments

- 1.) Line 150 and 376... unique NCE103 gene in *C. auris*. The Nce103 orthologs are present in other *Candida* spp. So, the Nce103 gene is not unique in *C. auris*, and this statement could be reframed throughout the manuscript. Additionally, synteny schema of Nce103 gene in *C. auris* with other *Candida* spp. will give more insights into the gene level homology.
- 2.) Some of the mutants are aggregative Fig. S3G. In this case, it is difficult to enumerate the infection dose, and to quantify the recovered fungal burden by CFU plating. Consider sonicating the fungal cells before infection and CFU plating for precise quantification (Ref: Bing J, 2024 Nat Commun (2024); DOI: 10.1038/s41467-024-46786-8).
- 3.) The experiments with the bacterial co-culture with the *C. auris* strains are reasonable. The urease positive bacterial skin colonizers promote *C. auris* growth by CO₂ production. But the effect of ammonia and PH change in the skin microenvironment may also influence the *C. auris* growth. Additional in vitro or in vivo assay to show the net effect on the *C. auris* growth by these urease positive bacterial skin colonizers would be interesting.
- 4.) Fig. 2B. The susceptibility of these clinical isolates to AmpB is missing and needs to be discussed in the results.
- 5.) The flocculation phenotype of *rca1Δefg1Δ* mutants in the Fig. S3G is not shown. If the cells are aggregative, they could be separated by sonication for MIC assay (Ref: Bing J, 2024 Nat Commun (2024); DOI: 10.1038/s41467-024-46786-8).
- 6.) Line 109 to 114 is unclear. The results do not match the figures. For instance, *rca1Δefg1Δ* is only in Fig. 2A but, Fig. S3A was cited in line 112 to 114.
- 7.) Some of the figure legends are incomplete (Fig. S7D).

Reviewer #2 (Remarks to the Author):

Here the authors combined proteomics and transcriptome approaches and identified the carbonic anhydrase Nce103 and its regulators Rca1 and Efg1 as components of the carbonic sensing pathway that contributes to *Candida auris* AMB resistance (modest 2 to 4 fold increase). This is a quite exciting finding as it is different from the known ergosterol biosynthetic pathway. As expected, high level of CO₂, including CO₂ released by bacterial ureases, restored AMB tolerance in the *nce103* mutant. The data to support NCE103's role in the modestly increased AMB resistance is strong. The authors further showed that CSP

contributes to skin colonization, and connected CSP with mitochondrial dysfunction and ROS generation. The evidence to support later conclusions was not as convincing.

General concerns:

1. I agree that Rca1 and Efg1 regulate NCE103. However, as the authors stated, these regulators likely control many other factors/pathways in addition to the carbonic anhydrase, the enzyme that converts CO₂ to bicarbonate. It is, therefore, not accurate to attribute the phenotypes of Rca1 and Efg1 mutations to the CSP pathway in some of the occasions. For example, Figure 4D: the competitive index for the rca1 mutant or the efg1 mutant during skin infection is very low, particularly for the efg1 mutant. In contrast, the difference between the control and the nce103 mutant is not even statistically significant. So the role of CSP's specific contribution to the intradermal infection is inconclusive at the best. There are other similar experiments/conclusions in this study that should be modified.
2. The role of NCE103 and its homologs in fungal growth in ambient air has been demonstrated previously. The importance of bicarbonate for metabolism is also known. The fact that high levels of CO₂ can restore nce103 mutant phenotype is expected. This is also consistent with no so drastically reduced fungal burden of the nce103 mutant in the skin infection models (Figure 4D-E). This is also consistent with the lack of virulence phenotype of cryptococcal carbonic anhydrase mutants. Thus, it feels a bit forced for the authors to conclude that CSP is critical for skin infection.
3. The authors showed that the sterol composition is similar between AMB resistant and sensitivity strains. It is recently shown that the inositolphosphorylceramide level has significant impact on AMB susceptibility. Given that both ergosterol and IPC are important components of fungal membrane and they affect various transporters and susceptibility to various drugs, the potential impact of CSP on both lipid components should be examined. The hypothesis that CSP may change the composition of membrane, which may indirectly affect mitochondrial function, should be considered. The evidence that CSP disruption impairs mitochondrial functions is not convincing based on the ROS evidence. The sensitivity to AA could also be an indirect effect because of altered membrane.
4. One thing that is missing is the experiment to determine if AMB therapy will make a difference in vivo when the Nce103 is blocked.

Specific concerns:

1. Figure 3E: why growth of the rca1, the efg1, and the double mutant is the best at intermediate AMB concentration with a relatively lower dose of Antimycin A?
2. Figure S5A: (1) Both S strain and nce103 mutant in R1 strain background showed restored resistance to AMB at 5.5% CO₂ level. However, efg1 sensitivity to AMB is not restored by CO₂, even with overexpression of NCE103. (2) The AMB sensitivity of aox2 and sod2 mutants did not change much in the presence of high levels of CO₂, arguing against the idea that they are affected by the bicarbonate levels.
3. It would be useful to survey the AMB resistant isolates used in Figure 2C to examine how common NCE103 is involved in AMB resistance (e.g. by qPCR) as RCA1 and EFG1 ultimately need to alter the function of NCE103 for their CSP specific effect. Given that Nce103 is essential in ambient air, the likely impact is on its expression level.
4. In Table S1, CLSI and EUCAST methods generated similar MICs (mostly within 2-4 fold). However, there are some large differences between the two methods. For example, MICs for VRC of the sensitive strain showed greater than 32 fold difference. Similarly, MICs for CAS of the strains AR285 and AR1097 are more than 32 fold different. Explanation?
5. Figure S3C: The authors stated that "both rca1Δ and efg1Δ mutants showed slightly increased susceptibility to 5-fluorocytosine (5FC), although not significant to caspofungin (CAS) or voriconazole (VRC) (Fig. S3C)". The CAS test did not reach the concentration that causes significant reduced growth for R1, and thus it is hard to tell. However, it looks like that there should be at least 2 or 4 fold difference with rca1 or efg1 mutations. Based on the graph of relative growth, both rca1Δ and efg1Δ mutants showed dramatically increased sensitivity to 5FC (8 fold for efg1 and 4 fold for rca1). This is more than they observed with AMB, the focus of this story. Are these changes in drug resistance/sensitivity related to bicarbonate levels?
6. Ptc2 functions in sensing high levels of CO₂ to control fungal response to high levels of CO₂ (like morphogenesis in *Candida albicans*). It is obviously not the same as the CSP pathway where conversion of CO₂ to bicarbonate is critical when the CO₂ level is very low. It is unclear how including this adds to this study.
7. Have the authors tested eNCE103 in the R1 strain without other mutations as a control for all the eNCE103 strains in various mutant backgrounds?
8. "Although there was no significant enhancement of AMB MIC in mutants, strain AR387 showed a 2-fold increase when treated with 250-1000 μM VitC. These findings suggest that, in addition to the ROS response, mitochondrial functions may control downstream effects resulting from CSP disruption." It is not clear to me how the evidence leads to the conclusion that ROS is an important factor in CSP's impact on mitochondrial function.
9. Figure 4A. Have the authors tested preculturing the strains in high level of CO₂ before the growth assay? Will such pre-adaptation reduce the difference in MM media?
10. Figure 4D-E: Could authors clarify if the nce103ΔNCE103 used in animal models is a complemented strain with NCE103 driven by its native promoter or it is an NCE103 overexpression strain in the nce103Δ mutant background?

Reviewer #2 (Remarks on code availability):

They used standard pipelines for proteomics and RNA seq analyses.

Reviewer #3 (Remarks to the Author):

The manuscript investigates the role of the carbonic sensing pathway (CSP) in antifungal resistance and fitness of *Candida auris*, focusing on the carbonic anhydrase Nce103 and its regulators Rca1 and Efg1. The authors provide compelling evidence that CSP is crucial for amphotericin B resistance (AMBR), mitochondrial function, and fungal adaptation to skin environments. They propose that bacterial urease-derived CO₂ may enhance *C. auris* colonization, suggesting potential therapeutic strategies

targeting CSP components.

The experimental design is robust, combining transcriptomics, proteomics, gene deletion, pharmacological inhibition, and in vivo/in vitro models. The findings are largely supported by data, and conclusions appear justified. The manuscript does not have apparent fatal flaws that would prohibit publication, though some aspects require additional clarification. In particular, the proteomics and RNA-seq experiment section would benefit from further details about the following:

- What statistical test was employed for the differential expression analysis?
- Was the data normalised and/or log transformed before statistics? If yes, what method of normalisation was used?
- After splitting the data into single CV and processing each one separately through MaxQuant, how were the results combined and normalised?
- Add a reference for the contaminants database.
- The actual code for proteomics data analysis is not disclosed in github or elsewhere.
- Although figure S1 mentions biological triplicates, more details on replicates for proteomics and RNA-seq in the methods section would be helpful.

I was not able to access the data in PRIDE as the provided username and password did not work.

On line 76, I suggest you remove the statement that "proteomics may identify key genes" and leave it at key pathways.

On line 454, change wording to "Fungal cells were grown..."

In fig. 1B., the authors have used a picture of a randomly-selected Fusion Lumos mass spectrometer. It is advisable (but not strictly necessary) to use a picture of the machine that was actually used in the experiments, that is the Exploris 480.

Other potential points to address may include:

- Are the fitness defects of *rca1Δ* and *efg1Δ* mutants fully attributable to mitochondrial dysfunction?
- How do CSP-related AMB resistance mechanisms compare to those in clinical isolates with known resistance mutations?
- Could the CSP be targeted in therapeutic interventions without disrupting host CO₂ metabolism?
- Were urease-negative bacterial strains tested for other factors that might affect fungal growth?

In conclusion, the study presents novel insights into the CSP's role in AMBR and fungal colonization. The integrated proteo-transcriptomics approach is an appropriate preliminary screen, and the in vivo skin colonization model strengthens translational relevance. The conclusions align well with the data and the claim that CSP can be a therapeutic target is reasonable, but potential off-target effects of inhibitors should be acknowledged. The study is strong and has substantial merit for publication after addressing the minor points as mentioned above.

Reviewer #3 (Remarks on code availability):

I have not reviewed the RNA-seq code because this is not my area of expertise. I would be happy to review the proteomics data analysis code if the authors provide it. I would also be happy to review the PRIDE submission if you give me a working username and password.

Version 1:

Reviewer comments:

Reviewer #1

(Remarks to the Author)

Authors performed additional experiments and addressed majority of critical comments. I recommend the manuscript for publication in this journal !

Reviewer #2

(Remarks to the Author)

I applaud the authors' effort in addressing the reviewers' concerns. I again believe that the finding of CSP contribution to *Candida auris* AMB resistance is a quite exciting as it is different from the known ergosterol biosynthetic pathway. That said, I do not think this discovery will have much impact much on antifungal therapy given the modest contribution of CSP to AMB resistance and the fact that once *C. auris* is disseminated, the high levels of CO₂ would compensate for the loss of CSP. The potential combination therapy of ABA with AMB and also anti-mitochondrial drugs with AMB have been proposed in other studies. It is important for the authors to acknowledge those in the manuscript without diminishing the novelty the findings reported here.

There are few specific concerns after reading the revised version.

1. "One thing that is missing is the experiment to determine if AMB therapy will make a difference in vivo when the Nce103 is blocked." I was surprised that the authors did systemic candidiasis model to address this concern. This is clearly a wrong animal model to test the idea as authors demonstrated so well that the elevated CO₂ levels in systemic infection would restore the AMB resistance due to lack of NEC103 in vitro. They need to test the AMB therapy in a different animal model, as they have done in Figure 6.

2. I am hoping that the authors can specifically conclude that CSP's specific contribution to the intradermal infection is inconclusive. The way it is written is misleadingly vague. The authors have proposed some likely reasons and those should be included in the revised manuscript rather than just in the rebuttal text for the reviewers.

3. I think it would be useful to include the FDA assay in the manuscript to bolster the conclusion that CSP may affect membrane integrity, particularly given that the authors did not find much changes in ergosterol or other lipids in the lipidomics experiments.
4. Although I am not the reviewer 1, I think the authors should include some of the data in the manuscript rather than just presenting in this rebuttal letter for the reviewers here. Examples include Fig. R2, R3.

Reviewer #3

(Remarks to the Author)

I would like to thank the authors for addressing all of my concerns and for their detailed answers. A few very minor points:

1. I was able to access the PRIDE submission this time and I am satisfied with the shared raw mass spec files and MaxQuant outputs. However, both the zenodo link and the PRIDE submission contain the generic *Cassiopeia*-LFQ code, not the actual script that was implemented for analysing this particular data set. I'll leave it to the editor to decide if that is acceptable.
2. It was interesting to see the additional lipidomics experiments. They are described very well, although unlike the other omics, these data was not shared.
3. What does the blue line in fig. S1F indicate? Could you please write it in the legend?

I look forward to seeing this paper published.

Decision Letter:

19th August 2025

Dear Karl,

Thank you for your patience while your manuscript "The carbonic anhydrase Nce103 is required for fitness and skin tropism of *Candida auris*" was under peer-review at Nature Microbiology. It has now been seen by 3 referees, whose expertise and comments you will find at the end of this email. You will see from their comments below that while they find your work of interest, some important points are raised. We are very interested in the possibility of publishing your study in Nature Microbiology, but would like to consider your response to these concerns in the form of a revised manuscript before we make a final decision on publication.

In particular, you will see that referee #2 says previous studies on combination therapy of ABA with AMB need to be acknowledged. The referee also says that the relevant data and text from the rebuttal need to be added to the manuscript. Please note that we editorially overrule the request for additional animal infection experiments. Please respond to referee #2's request through additional careful text changes (and adding the data from the rebuttal to the paper). Please also respond to the data comments from referee #3. The rest of the referees' reports are clear and the remaining issues should be straightforward to address.

If you have not done so already please begin to revise your manuscript so that it conforms to our Article format instructions at <http://www.nature.com/nmicrobiol/info/final-submission/>

The usual length limit for a Nature Microbiology Article is six display items (figures or tables) and 4,000 words. We have some flexibility, and can allow a revised manuscript at 4,500 words, but please consider this a firm upper limit. There is a trade-off of ~250 words per display item, so if you need more space, you could move a Figure or Table to Supplementary Information.

Some reduction could be achieved by focusing any introductory material and moving it to the start of your opening 'bold' paragraph, whose function is to outline the background to your work, describe in a sentence your new observations, and explain your main conclusions. The discussion should also be limited. Methods should be described in a separate section following the discussion, we do not place a word limit on Methods.

Nature Microbiology titles should give a sense of the main new findings of a manuscript, and should not contain punctuation. Please keep in mind that we strongly discourage active verbs in titles, and that they should ideally fit within 90 characters each (including spaces).

Please include a data availability statement as a separate section after Methods but before references, under the heading "Data Availability". This section should inform readers about the availability of the data used to support the conclusions of your study.

This information includes accession codes to public repositories (data banks for protein, DNA or RNA sequences, microarray, proteomics data etc...), references to source data published alongside the paper, unique identifiers such as URLs to data repository entries, or data set DOIs, and any other statement about data availability. At a minimum, you should include the following statement: "The data that support the findings of this study are available from the corresponding author upon request", mentioning any restrictions on availability. If DOIs are provided, we also strongly encourage including these in the Reference list (authors, title, publisher (repository name), identifier, year). For more guidance on how to write this section please see: <http://www.nature.com/authors/policies/data/data-availability-statements-data-citations.pdf>

To improve the accessibility of your paper to readers from other research areas, please pay particular attention to the wording of the paper's opening bold paragraph, which serves both as an introduction and as a brief, non-technical summary in about 150 words. If, however, you require one or two extra sentences to explain your work clearly, please include them even if the paragraph is over-length as a result. The opening paragraph should not contain references. Because scientists from other sub-disciplines will be interested in your results and their implications, it is important to explain essential but specialised terms concisely. We suggest you show your summary paragraph to colleagues in other fields to uncover any problematic concepts.

If your paper is accepted for publication, we will edit your display items electronically so they conform to our house style and will reproduce clearly in print. If necessary, we will re-size figures to fit single or double column width. If your figures contain several parts, the parts should form a neat rectangle when assembled. Choosing the right electronic format at this stage will speed up the processing of your paper and give the best possible results in print. We would like the figures to be supplied as vector files - EPS, PDF, AI or postscript (PS) file formats (not raster or bitmap files), preferably generated with vector-graphics software (Adobe Illustrator for example). Please try to ensure that all figures are non-flattened and fully editable. All images should be at least 300 dpi resolution (when figures are scaled to approximately the size that they are to be printed at) and in RGB colour format. Please do not submit Jpeg or flattened TIFF files. Please see also 'Guidelines for Electronic Submission of Figures' at the end of this letter for further detail.

Figure legends must provide a brief description of the figure and the symbols used, within 350 words, including definitions of any error bars employed in the figures.

When submitting the revised version of your manuscript, please pay close attention to our [href="https://www.nature.com/nature-research/editorial-policies/image-integrity">Digital Image Integrity Guidelines. and to the following points below:](https://www.nature.com/nature-research/editorial-policies/image-integrity)

EXTENDED DATA FIGURES

Please include a statement before the acknowledgements naming the author to whom correspondence and requests for materials should be addressed.

Finally, we require authors to include a statement of their individual contributions to the paper -- such as experimental work, project planning, data analysis, etc. -- immediately after the acknowledgements. The statement should be short, and refer to authors by their initials. For details please see the Authorship section of our joint Editorial policies at http://www.nature.com/authors/editorial_policies/authorship.html

* include a point-by-point response to any editorial suggestions and to our referees. Please include your response to the editorial suggestions in your cover letter, and please upload your response to the referees as a separate document.

* ensure it complies with our format requirements for Letters as set out in our guide to authors at www.nature.com/nmicrobiol/info/gta/

* resubmit electronically if possible using the link below to access your home page:

Link Redacted

*This url links to your confidential homepage and associated information about manuscripts you may have submitted or be reviewing for us. If you wish to forward this e-mail to co-authors, please delete this link to your homepage first.

Please ensure that all correspondence is marked with your Nature Microbiology reference number in the subject line.

Nature Microbiology is committed to improving transparency in authorship. As part of our efforts in this direction, we are now requesting that all authors identified as 'corresponding author' on published papers create and link their Open Researcher and Contributor Identifier (ORCID) with their account on the Manuscript Tracking System (MTS), prior to acceptance. This applies to primary research papers only. ORCID helps the scientific community achieve unambiguous attribution of all scholarly contributions. You can create and link your ORCID from the home page of the MTS by clicking on 'Modify my Springer Nature account'. For more information please visit www.springernature.com/orcid.

We hope to receive your revised paper within three weeks. If you cannot send it within this time, please let us know.

Yours sincerely,

Reviewer Expertise:

Referee #1: *C. auris*, skin infections

Referee #2: Fungal pathogenesis, fungal genetics, antifungal drugs

Referee #3: Proteomics, mass Spectrometry

Reviewers Comments:

Reviewer #1 (Remarks to the Author):

Authors performed additional experiments and addressed majority of critical comments. I recommend the manuscript for publication in this journal !

Reviewer #2 (Remarks to the Author):

I applaud the authors' effort in addressing the reviewers' concerns. I again believe that the finding of CSP contribution to *Candida auris* AMB resistance is a quite exciting as it is different from the known ergosterol biosynthetic pathway. That said, I do not think this discovery will have much impact much on antifungal therapy given the modest contribution of CSP to AMB resistance and the fact that once *C. auris* is disseminated, the high levels of CO₂ would compensate for the loss of CSP. The potential combination therapy of ABA with AMB and also anti-mitochondrial drugs with AMB have been proposed in other studies. It is important for the authors to acknowledge those in the manuscript without diminishing the novelty the findings reported here.

There are few specific concerns after reading the revised version.

1. "One thing that is missing is the experiment to determine if AMB therapy will make a difference in vivo when the Nce103 is blocked." I was surprised that the authors did systemic candidiasis model to address this concern. This is clearly a wrong animal model to test the idea as authors demonstrated so well that the elevated CO₂ levels in systemic infection would restore the AMB resistance due to lack of NEC103 in vitro. They need to test the AMB therapy in a different animal model, as they have done in Figure 6.
2. I am hoping that the authors can specifically conclude that CSP's specific contribution to the intradermal infection is inconclusive. The way it is written is misleadingly vague. The authors have proposed some likely reasons and those should be included in the revised manuscript rather than just in the rebuttal text for the reviewers.
3. I think it would be useful to include the FDA assay in the manuscript to bolster the conclusion that CSP may affect membrane integrity, particularly given that the authors did not find much changes in ergosterol or other lipids in the lipidomics experiments.
4. Although I am not the reviewer 1, I think the authors should include some of the data in the manuscript rather than just presenting in this rebuttal letter for the reviewers here. Examples include Fig. R2, R3.

Reviewer #3 (Remarks to the Author):

I would like to thank the authors for addressing all of my concerns and for their detailed answers. A few very minor points:

1. I was able to access the PRIDE submission this time and I am satisfied with the shared raw mass spec files and MaxQuant outputs. However, both the zenodo link and the PRIDE submission contain the generic Cassiopeia-LFQ code, not the actual script that was implemented for analysing this particular data set. I'll leave it to the editor to decide if that is acceptable.

2. It was interesting to see the additional lipidomics experiments. They are described very well, although unlike the other omics, these data was not shared.

3. What does the blue line in fig. S1F indicate? Could you please write it in the legend?

I look forward to seeing this paper published.

Version 2:

Decision Letter:

Our ref: NMICROBIOL-25020560B

9th September 2025

Dear Karl,

Thank you for submitting your revised manuscript "The carbonic anhydrase Nce103 is required for fitness and skin tropism of *Candida auris*" (NMICROBIOL-25020560B). Editorially, we find that the paper has improved in revision, and therefore we'll be happy in principle to publish it in Nature Microbiology, pending minor revisions to comply with our editorial and formatting guidelines.

We are now performing detailed checks on your paper and will send you a checklist detailing our editorial and formatting requirements in about two weeks. Please do not upload the final materials and make any revisions until you receive this additional information from us.

Thank you again for your interest in Nature Microbiology. Please do not hesitate to contact me if you have any questions.

Sincerely,

Version 3:

Decision Letter:

14th October 2025

Dear Karl,

I am pleased to accept your Article "Candida auris skin tropism and antifungal resistance are mediated by carbonic anhydrase Nce103" for publication in Nature Microbiology. Thank you for having chosen to submit your work to us and many congratulations.

Due to the importance of these deadlines, we ask you please us know now whether you will be difficult to contact over the next

month. If this is the case, we ask you provide us with the contact information (email, phone and fax) of someone who will be able to check the proofs on your behalf, and who will be available to address any last-minute problems.

Authors may need to take specific actions to achieve compliance with funder and institutional open access mandates. If your research is supported by a funder that requires immediate open access (e.g. according to [Plan S principles](https://www.springernature.com/gp/open-science/plan-s-compliance) or the [NIH public access policy](https://www.springernature.com/gp/open-science/us-federal-agency-compliance)) then you should select the gold OA route, and we will direct you to the compliant route where possible. Because authors warrant under our subscription licensing terms that they haven't committed to licensing any version of their article under a licence inconsistent with the terms of our agreement – including the applicable embargo period – publication under the subscription model isn't suitable for authors whose funders require no embargo.

Congratulations once again and I look forward to seeing the article published.

With kind regards,

P.S. Click on the following link if you would like to recommend Nature Microbiology to your librarian <http://www.nature.com/subscriptions/recommend.html#forms>

** Visit the Springer Nature Editorial and Publishing website at http://editorial-jobs.springernature.com?utm_source=ejP_NMicro_email&utm_medium=ejP_NMicro_email&utm_campaign=ejP_NMicro for more information about our career opportunities. If you have any questions please click [here](mailto:editorial.publishing.jobs@springernature.com).

RESPONSE TO REFEREES

Reviewer #1 (Remarks to the Author):

The study by Cahn TP, et al. is focused on CO₂ sensing mechanism of *C. auris* that is unique in promoting advantage in the pathogen for acquiring antifungal resistance, colonization and how the bacterial skin colonizers can promote the *C. auris* growth by CO₂ production. It is very interesting to note that *C. auris* utilizes carbonic sensing pathways to gain fitness for resistance and colonization; 1.) Antifungal resistance due to CO₂ sensing, 2.) Tropisms of *C. auris* in skin tissue due to CO₂ sensing mechanism and 3.) How the bacterial skin colonizers promote *C. auris* growth by CO₂ production. Understanding the mechanisms governing antifungal resistance and skin colonization is very important to understand the pathogenesis of this emerging fungal pathogen. This study for the first time provided a unique mechanisms. Although it's very interesting to look into the CO₂ sensing in *C. auris*, performing some additional experiments would enhance the study.

Response: We appreciate the positive feedback from Reviewer #1. We agree that the CO₂-sensing pathway in *C. auris* is intriguing, influencing both fungal fitness and drug resistance, as well as organ tropism. Our work demonstrates how the CSP mitigates typical fitness costs often associated with AMB resistance by CO₂ fixation taking advantage of the carbonic anhydrase. To address technical limitations, we have conducted additional control experiments and updated figures accordingly, with supplementary data to be included in the Peer Review report as outlined below.

Major comments

1) Currently, the author does not have data to show that Rca1 regulates Nce103, and it acts on Efg1 to control antifungal susceptibility. Performing additional experiments to establish a connection between NCE103 and Rca1 and Efg1 mechanism in *C. auris* is important.

Response: We do acknowledge the need for data linking Rca1, Nce103, and Efg1 in *C. auris*. In fact, we performed a CHIP experiment and planned a ChIP-seq experiment, but functional challenges were encountered when tagging Rca1 and Efg1 prevented reliable results (**Fig. R1A, B**). Although we have not been able to demonstrate a direct interaction of Rca1 with cognate *cis*-acting sites in the *NCE103* promoter regions. This is not uncommon, as many transcriptional regulators can control target gene expression engaging additional factors. Moreover, the affinity of transcription factor-DNA interactions (on- versus off-rate constants) may also affect results in ChIP or ChIP-seq experiments. Additionally, epitope-tagging might leave functionality intact, but affect regulatory complex dynamics at recognition sites.

As an alternative, we performed *in silico* analysis using YeTFaSCo (de Boer and Hughes, 2012), identifying conserved transcription factor binding sites in the *NCE103* promoter for Rca1 and Efg1 (**Fig. R1C**). Additionally, Rca1, Efg1, and Nce103 are unique homologues in the *Candida auris* genome, suggesting conserved functions across *Candida* species. Based on previous reports and our own findings, we propose that Rca1 is a key regulator of *NCE103* (Cottier et al., 2012; Pohlers et al., 2017; Vandeputte et al., 2012). To further support a regulatory impact by Rca1 and/or Efg1, we performed qPCR analysis on *RCA1* and/or *EFG1* mutants, showing significantly impaired *NCE103* expression under both low and high CO₂ conditions (**Fig. R1D**). We provide the results here for the inspection by Reviewer #1 and the Editor (**Fig. R1**), but we do not add all data into the revised manuscript in the sense of brevity.

Changes in revised manuscript, lines 164-171: Additionally, our RNA-seq data revealed a strong 6-fold and 9-fold downregulation of *NCE103* upon deletion of *RCA1* and *EFG1*, respectively (**Fig. 2D**). To further investigate how *RCA1* and *EFG1* regulate *NCE103* expression, we examined *NCE103* mRNA levels in *C. auris* cultured under low and high CO₂ conditions. As expected, supplementation of 5.5% CO₂ resulted in a modest reduction in *NCE103* expression in the R1 strain. In contrast, deletion of *RCA1* and/or *EFG1* led to a marked repression of *NCE103* expression, with an approximate 4- to 9-fold decrease under both CO₂ conditions (**Fig. 2G**). In our proteomics experiment, AMB stress induced Nce103 abundance in strain S, but failed to reach Nce103 levels in R1 (**Fig. S1F**).

Figure R1. A. AlphaFold prediction of the Rca1–9×myc fusion protein indicates that the 9×myc epitope tag forms a flexible “arm” suggesting that a 9×myc tag should be recognized by an anti-myc antibody. **B.** ChIP–qPCR data was not successful, as the negative-control site (within the *NCE103* ORF) also yielded signals, not just the promoter-specific probes. **C.** Motif-binding predictions were performed for the *NCE103* promoter regions against orthologues of Rca1 (Cst6) and Efg1 (Sok2), identifying putative TF-binding sites for both regulators. **D.** *NCE103* mRNA levels in *C. auris* mutants under low- (ambient) and high- (5.5%) CO₂ conditions show that both Rca1 and Efg1 modulate *NCE103* expression. Approximately 200 CFUs of *C. auris* were plated onto YPD agar and incubated at 37 °C under either 5.5% CO₂ or ambient air conditions for 2 days. Colonies were then harvested for mRNA isolation. Two-way ANOVA followed by Tukey’s multiple comparison test was applied.

2) Is CO₂ sensing mechanism contributing fitness unique to *C. auris* unique or similar phenomenon observed in other *Candida* spp as well such as *C. albicans*?

Response: Indeed, the CO₂-sensing mechanism contributes to fitness in other fungal pathogens as well (Avelar et al., 2024; Bahn et al., 2005; Cottier et al., 2012; Hall et al., 2010; Vandeputte et al., 2012). To further address this point, we performed spotting assays using multiple *Candida* species and *Cryptococcus neoformans* on minimal medium agar. Results showed enhanced growth of most *Candida* species in 5.5% CO₂ (Fig. R2), while high CO₂ slightly inhibited *Cryptococcus neoformans*, consistent with previous findings related CO₂ stress for this species (Krysan et al., 2019). We provide the results here for the inspection of Reviewer #1 and the Editor (Fig. R2), but we do not add the data into the revised manuscript in the sense of brevity.

Figure R2. CO₂ promotes the fitness of most *Candida* spp pathogens but not *Cryptococcus neoformans* (*Cryp. neo*). Strains used in this experiment: R1::*nce103Δ*; *C. albicans* SC5314; *C. glabrata* ATCC 2001; *C. glabrata* KK01; *C. glabrata* KK04; *C. glabrata* KK05; *C. krusei* ATCC6258; *C. parapsilosis* ATCC22019; *C. dupliniensis* KK0908899; *C. lusitanae* DSM 70102; *Cryptococcus neoformans* NEQAS strain

3) The mutant strains appear to have growth defects in the MM and YPD Fig. 4A. In this case the CFU data from Fig. 4D and 4E is difficult to interpret. Performing a traditional growth curve assay using RPMI-MOPS to show that the growth defects in these mutants are not significant to interfere in the fungal burden from the murine skin would be useful.

Response: Thank you for this thoughtful suggestion. We recorded growth curves starting from an initial OD_{600nm} of ~0.1–0.2, finding no significant differences between WT and most null mutants, except for *nce103Δ*. This difference may be due to a compensatory effect by CO₂ from fungal metabolism (Hall et al., 2010). To better assess early growth, we repeated the assays with a lower starting OD_{600nm} of ~0.001. While WT strains showed faster initial growth, the final carrying capacity (K) was similar between WT and mutants. In the original Fig. 4D, E (**now Fig. 6D, E**), the infection time courses of 3 and 14 days with the high infection dose, should allow *C. auris* populations to reach their maximum biomass capacity at the expected time points (**Fig. R3**). Thus, we believe the differences in **Fig. 6D, E** are not due to growth bias. We provide the results here for the inspection by Reviewers and the Editor, but we do not add the data into the revised manuscript in the sense of brevity (**Fig. R3**).

Figure R3. Growth curve of *C. auris* in RPMI-MOPS medium, using starting OD₆₀₀ of 0.001 and 0.2. Fungal metabolism-derived CO₂ is likely to compensate for fitness defects of *rca1Δ* and *efg1Δ* mutants.

Regarding the fungal CFU burden on skin, we would like to clarify as follows. In low-CO₂ environments such as the skin surface, *Candida* relies on carbonic anhydrase to convert minute amounts of CO₂ into HCO₃⁻, which feeds into central metabolism to support fungal fitness (Bahn et al., 2005; Vandeputte et al., 2012). By contrast, high-CO₂ environments such as present in deeper skin tissues or internal organs, higher CO₂ levels of up to 5.5% can spontaneously convert to HCO₃⁻ in aqueous environments, reducing the need for carbonic anhydrases. Thus, defects in this CO₂-conversion system may impair fungal growth and contribute to lower CFUs on the skin. Note that RPMI medium mimics internal host environments, but it does not accurately reflect skin conditions. To address this, we used the infection model with extended infection durations (3 and 14 days) to reveal clinical relevance. The 14-day period approximates the typical window for *C. auris* acquisition during hospitalization and daily skin disinfection procedures reflects common hospital hygiene practices. This design aims to evaluate clinical relevance of CSP.

4) Unclear if AmpB sensitive isolates have genetic mutation in either Rca1, Nce103 and/or Efg1? Is it possible to perform a correlation analysis between the genetic mutation harbored in Rca1, Nce103 or Efg1 and the AmpB susceptibility?

Response: Thank you for this valuable remark. The AMB-sensitive isolate 2431/P/16 used in our study harbors a nonsense stop mutation in *RCA1*, which explains the reduced expression of *Nce103* (**Fig. 2B**). However, no mutations were found by sequencing in either *NCE103* or *EFG1*. To further probe this pathway, we used another AMB-sensitive strain, B8441 (AR387), in which *NCE103* was depleted via a Tet-Off system, leading to similarly diminished AMB MICs. These data fully support our conclusion that the CSP pathway provides a basal level of AMB tolerance in *C. auris*, potentially contributing to the elevated MICs observed in clinical isolates. Hence, targeting this pathway could enhance AmB efficacy across both resistant and sensitive strains, and may represent

a broader mechanism of intrinsic tolerance for other species of the *Candida haemulonii* complex (Francisco et al., 2023). To explore this further, we mined publicly available genome assemblies (NCBI nt/nt and WGS databases), covering *C. auris* isolates from diverse regions. Unfortunately, we do not have physical access to these strains for direct MIC susceptibility testing. Due to incomplete metadata for many strains retrieved via NCBI *blastn* and variability in MIC testing methods across studies, establishing precise genotype-phenotype correlations / predictions remain a formidable challenge. Additionally, the presence of complex acquired resistance mechanisms introduces further complexity that may preclude clear correlations (**Fig. R4B**) (Carolus et al., 2024).

To better address the concern from Reviewer #1, we have updated the analysis in **Fig. R4** to include recent data from NCBI, analyzing SNPs and indel variants in CSP-related genes from over 460 strains. Interestingly, the results show that *RCA1* harbors higher mutational variabilities when compared to downstream regulators *EFG1* and *NCE103*, as shown in our initial analysis (**Fig. R4A**). Of note, strains carrying indel mutations causing frameshifts in CSP genes are typically AMB-sensitive (**Fig. R4B** – Green strains). However, among acquired resistance mechanisms, such as in the loss-of-function mutations (LoF) in *ERG6* or *NCP1* (Carolus et al., 2024; Rybak et al., 2022), which may act independently and increase AMB resistance even in strains with CSP LoF mutations (**Fig. R4B**). We are aware and point out that these findings are based on *BLASTn* analysis against *C. auris* genome assemblies of varying qualities and should thus be interpreted with caution. The observed variants may provide preliminary insights and support future investigations, but validating these mutations is beyond the scope of this work. Importantly, our functional genetic data provides strong and compelling support that disruption of CSP in different clinical isolates contributes to AMB susceptibility.

Nonetheless, we attempted to validate these findings using whole-genome sequencing (WGS) analysis. However, due to the absence of some datasets, variability in data quality, and differences in sequencing platforms, we concluded that these limitations make the generation of robust data challenging. We are happy to publish the transparent peer review file, provided this aligns with the journal's policy. We should emphasize that we believe this to be a highly interesting observation for future investigation, especially as thousands of *C. auris* whole-genome sequencing datasets from clinical isolates are becoming available through NCBI. Hence, we provide the results here for the inspection of the Reviewers and the Editor (**Fig. R4**), but we do not add the data into the revised manuscript as explained above.

Figure R4. Genetic mutations in CSP across *C. auris* clinical isolates. **A.** Candidate genes within the CSP locus from reference strain B8441 were queried using *BLASTn* against *C. auris* genome assemblies present in the NCBI whole-genome shotgun (WGS) and nucleotide (nt/nt) databases. Putative cDNA-level variants identified across 460+ clinical isolates are visualized as lollipop plots. **B.** Truncated protein variants found among isolates were classified as AMB-susceptible (green) or AMB-resistant (pink); black indicates isolates for which MIC data were not available. Variants were predicted relative to primary sequences from strain B8441, and annotated domains were retrieved from the InterPro database. LoF: loss-of-function. As this analysis is based on assembled genomes without raw read validation, the results should be considered exploratory.

Minor comments

1.) Line 150 and 376... unique NCE103 gene in *C. auris*. The Nce103 orthologs are present in other *Candida* species. So, the Nce103 gene is not unique in *C. auris*, and this statement could be reframed throughout the manuscript. Additionally, synteny schema of Nce103 gene in *C. auris* with other *Candida* species. will give more insights into the gene level homology.

Response: Thank you very much, this is a very good idea and apologies for the confusing wording. We reworded and further added the synteny scheme for Nce103, Rca1 and Efg1 in **Figs. 2C & S2C** in the revised manuscript (**Fig. R5**).

Changes in revised manuscript, lines 143-144: The carbonic anhydrase is conserved across many fungal species and encoded by the *NCE103* (B9J08_000363) gene in *C. auris* (**Figs. 2C, S3E**).

Figure R5. The synteny schema illustrates the conservation of CSP genes and adjacent ORFs in fungal pathogens.

2.) Some of the mutants are aggregative Fig. S3G. In this case, it is difficult to enumerate the infection dose, and to quantify the recovered fungal burden by CFU plating. Consider sonicating the fungal cells before infection and CFU plating for precise quantification (Ref: Bing J, 2024 Nat Commun (2024); DOI: 10.1038/s41467-024-46786-8).

Response: Thank you for bringing up this point. We apologize for not being clear. We initially encountered challenges with the *efg1Δ* mutant due to its aggregative flocculation phenotype. In our first *in vivo* experiment, using a high inoculum (2×10^7 cells/100 μ l per 21.5 g of body weight), all mice ($n = 3$) died within one day, likely due to vascular blockage caused by cell aggregation. Therefore, we adjusted the protocol for systemic infections by reducing the inoculum (2×10^6 cells/100 μ l per 21.5 g of body weight). To quantify fungal cells accurately, we used a CASY cell counter after mild sonication of cell suspensions in CASY tubes (30 seconds) (Bing et al., 2024). Cell counts were then adjusted to need, and the suspensions were vigorously vortex-mixed prior to infection. To ensure accuracy, we verified cell counts using both the CASY and CFU counting for each experiment. The OD₆₀₀ measurement was used as another control. We did not see very much variation in initial infection doses between strains with this protocol. Therefore, we believe that concerns about data quality are unwarranted.

Of note, our data confirms that the flocculation phenotype led to higher CFU burden in brain tissues as reported earlier (Bing et al., 2024). Flocculation of *efg1Δ* mutants is most likely due to upregulation of *ALS4112* (Als4), as reported before for *C. auris* (Bing et al., 2023; Santana et al., 2023). To avoid a sonication-based fitness bias, the infection suspension was vortex-mixed. The Methods section has been updated for clarification.

Changes in revised manuscript, lines 836-840: For all experiments, initial infection doses were adjusted using a CASY cell counter after 30 seconds of sonication and further verified by CFU-counting. Cell density was also confirmed by recording OD_{600nm} whenever *efg1Δ* was used in the experiments. Prior to infection, cell suspensions were vigorously vortex-mixed to minimize clumping and maintain planktonic cell suspensions that ensure uniform distribution.

3.) The experiments with the bacterial co-culture with the *C. auris* strains are reasonable. The urease positive bacterial skin colonizers promote *C. auris* growth by CO₂ production. But the effect of ammonia and PH change in the skin microenvironment may also influence the *C. auris* growth. Additional in vitro or in vivo assay to show the net effect on the *C. auris* growth by these urease positive bacterial skin colonizers would be interesting.

Response: Thank you for your thoughtful suggestion. We provide the results of relevant experiments here for the inspection by Reviewers and the Editor (**Fig. R6**), but we do not add the data into the revised manuscript in the sense of brevity. In our preliminary *in vitro* experiments without pH-buffering, *Proteus mirabilis* raised the pH of the YPD medium above 10, completely inhibiting *Candida auris* growth. In contrast, *C. auris* grew well with *Klebsiella pneumoniae* and *Staphylococcus aureus*., highlighting the potential impact of pH changes on fungal growth (**Fig. R6**). Notably, the human skin surface maintains a relatively stable pH of 4.5–5.5 (Lambers et al., 2006). It is important to note that in *in vitro* experiments, the system is closed, and ammonia remains dissolved in the medium. However, on the skin as an open system, ammonia reaches ambient air. Additionally, skin urea levels are likely lower but more dynamic than in the initial 2% used in Christensen's agar, as urea is continuously released into sweat yielding dynamic concentration ranges. Some additional intriguing questions apply. What is happening inside the hair follicles or within the follicle shaft through which hairs egress? What does the pH change between the skin surface and hair follicles? Do other factors such as fatty acids released from sebaceous glands or sweat composition contribute to the overall fitness?

Figure R6. A experimental scheme for panel B. Urease agar was poured into two middle lanes for bacterial culture. YPD agar was added into two outside lanes for *C. auris* culture. **B.** Ammonia produced by *Proteus mirabilis* increased pH of YPD agar over 10 as evident from the pH indicator, which inhibited *C. auris* growth. **C.** *C. auris* alone and mixture with *P. mirabilis* were co-infected on mice back skin for 2 days before evaluating fungal burdens.

Of note, we have not observed significant differences in the mouse skin model using our standardized experimental setup (n = 6–7 mice per group – **Fig. R6C**). These negative results may be due to physiological differences between human and mouse skin. In humans, eccrine glands are widely distributed over the body, allowing for efficient thermoregulation via sweat release, whereas in most other mammals, including mice, the glands are restricted to paw pads, and heat dissipation relies on less efficient mechanisms such as panting, salivation, and increased blood flow to sparsely haired areas (Cui and Schlessinger, 2015). We also notice that *ex vivo* human skin models may need pharmacological stimulation to induce sweat secretion (Nakashima et al., 2023). One promising approach may be daily addition of synthetic sweat to better mimic native human skin *in vivo*. Additionally, employing immunocompromised skin models could help uncover interactions that are otherwise masked by host innate immune responses. While we are highly motivated to pursue some of these questions further, we need to emphasize that this would be far beyond the scope of this work.

Changes in revised manuscript, lines 410-413: Notably, while eccrine sweat glands are widely distributed in human skin, they are rare in fur-covered mammals such as mice⁷⁵. Therefore, there is a need to establish suitable animal models and *ex vivo* human skin systems to investigate interkingdom interactions on the skin surface⁷⁶.

4.) Fig. 2B. The susceptibility of these clinical isolates to AmpB is missing and needs to be discussed in the results.

Response: Indeed, we agree, and we addressed this concern above in Fig. R4.

5.) The flocculation phenotype of *rca1Δefg1Δ* mutants in the Fig. S3G is not shown. If the cells are aggregative, they could be separated by sonication for MIC assay (Ref: Bing J, 2024 Nat Commun (2024); DOI: 10.1038/s41467-024-46786-8).

Response: Thank you. We have updated **Fig. S3B (previously S3G)** to include the obvious flocculation phenotype of the *rca1Δ efg1Δ* strain. The flocculation observed in *efg1Δ* is likely mediated by Als4 (*ALS4112*), as previously described (Bing et al., 2023; Pelletier et al., 2024). The flocculation is reversible by dilution in distilled water (Pelletier et al., 2024). When *C. auris* was cultured in RPMI medium for 24 hours, we clearly observed that the *efg1Δ* strains showed pronounced aggregation (**Fig. R7C**). This indicates that the flocculation effect described is not only due to the initial inoculum preparation but occurs during the incubation period. To prepare stock suspensions for MIC assays, cells were first diluted in distilled water (Pelletier et al., 2024) and subsequently diluted 1:400 into RPMI medium. Therefore, we do not believe that the separation of aggregated cells impacts MIC outcomes. Instead, we propose that Als4-mediated flocculation is reversible and that reaggregation may occur upon dilution into RPMI or during proliferation. This re-aggregation may reduce drug accessibility, because cell clumps could suffer from limited exposure to amphotericin B (Du et al., 2020), thus enabling better growth at higher drug concentrations in *efg1Δ* when compared to *rca1Δ*. We also cannot exclude the possibility that Efg1 rewires downstream pathways that collectively contribute to antifungal susceptibilities (Glazier, 2022). Indeed, the deletion of *CYR1*, an upstream component of the cAMP-PKA pathway associated with Efg1, resulted in increased AMB^R (Kim Ji-Seok et al., 2021). We provide the results of relevant experiments here for the inspection by Reviewers and the Editor (**Fig. R7**), but we do not add all the data into the revised manuscript in the sense of brevity.

Figure R7. Adhesion (A) and flocculation (B, C) phenotypes of *efg1Δ* mutants. A–B. Fungal cells were cultured overnight and washed twice with dH₂O. Flocculation assays were performed in dH₂O for 15 minutes. C. Cells were cultured overnight on 12-well plates containing RPMI medium. Images were captured using bright-field microscopy (60x magnification) after 24 hours. Experiments were performed in three biological replicates.

Changes in revised manuscript, lines 114-117: Surprisingly, *efg1Δ* and the *rca1Δefg1Δ* double mutant showed a 2-fold increase in MICs when compared to the *rca1Δ* deletion (**Fig. 2A**). This may be attributed to the rewiring of multiple downstream pathways influencing AMB tolerance or to the flocculation phenotype of *efg1Δ* as previously observed in clinical isolates (**Fig. S3B**)⁴⁴.

6.) Line 109 to 114 is unclear. The results do not match the figures. For instance, *rca1Δefg1Δ* is only in Fig. 2A but, Fig. S3A was cited in line 112 to 114.

Response: Thank you for noticing and bringing us to this point. We apologize for the confusion and the lack of clarity. In Fig. 2A, we present the AMB MICs of mutants generated in the R1 background, including strains S and AR387 as controls. Fig. S3A provides the AMB MICs for *rca1Δ* and *efg1Δ* mutants generated in different clinical backgrounds. Our original intention was to avoid redundancy, which is why we referred to both figures together in lines 109–110. We modified the figure citations to better explain the details of each mutant more explicitly.

7.) Some of the figure legends are incomplete (Fig. S7D).

Response: Thank you and apologies - we corrected the relevant legends.

Changes in legend of Fig S6D: A representative picture illustrates the experimental setup for bacterial-fungal co-culture assay in a 12-well plate.

Reviewer #2 (Remarks to the Author):

Here the authors combined proteomics and transcriptome approaches and identified the carbonic anhydrase Nce103 and its regulators Rca1 and Efg1 as components of the carbonic sensing pathway that contributes to *Candida auris* AMB resistance (modest 2 to 4 fold increase). This is a quite exciting finding as it is different from the known ergosterol biosynthetic pathway. As expected, high level of CO₂, including CO₂ released by bacterial ureases, restored AMB tolerance in the nce103 mutant. The data to support NCE103's role in the modestly increased AMB resistance is strong. The authors further showed that CSP contributes to skin colonization, and connected CSP with mitochondrial dysfunction and ROS generation. The evidence to support later conclusions was not as convincing.

Response: We sincerely appreciate the insightful suggestions provided by Reviewer #2. We have made every effort to address the relevant concerns in a comprehensive manner. We were encouraged by the positive feedback from Reviewer #2, who emphasizes the importance of this study. Below, we provide detailed responses to each point.

General concerns:

1. I agree that Rca1 and Efg1 regulate NCE103. However, as the authors stated, these regulators likely control many other factors/pathways in addition to the carbonic anhydrase, the enzyme that converts CO₂ to bicarbonate. It is, therefore, not accurate to attribute the phenotypes of Rca1 and Efg1 mutations to the CSP pathway in some of the occasions. For example, Figure 4D: the competitive index for the rca1 mutant or the efg1 mutant during skin infection is very low, particularly for the efg1 mutant. In contrast, the difference between the control and the nce103 mutant is not even statistically significant. So the role of CSP's specific contribution to the intradermal infection is inconclusive at the best. There are other similar experiments/conclusions in this study that should be modified.

Response: Thank you for drawing our attention to this point. We agree that the effect of *rca1*Δ in the intradermal model is not striking, and we refined this section to reflect that more clearly. We also tend to believe that the CSP may not play a vital role in intradermal infections, likely because higher tissue CO₂ levels could compensate the lack of *RCA1* and *NCE103*. The slightly reduced fungal burden observed in the intradermal model may be due to the thin epidermal layer in mouse skin, which could affect CO₂ availability in this microenvironment.

Changes in revised manuscript, lines 282-284: Competitive quantitative fitness assays revealed that the R1 strain exhibited significantly higher fitness when compared to the *efg1*Δ mutant, but only a modest advantage over the *rca1*Δ and *nce103*Δ mutants (**Fig. 6D**).

Changes in revised manuscript, lines 293-296: Overall, our data demonstrate that the CSP is crucial for the initial colonization of native human and mouse skin (**Fig. 6H, I**). Moreover, Efg1 is required for fitness, virulence and long-term persistence on mouse skin, as well as for subsequent infection but independently of Nce103.

Indeed, both Rca1 and Efg1 affect the expression of *NCE103*, as well as over 100 additional transcription factors reflecting the underlying complexity (**Fig. S6G**). In *C. albicans*, Rca1 and Efg1 are well-characterized transcription factors involved in diverse cellular processes (Glazier, 2022; Tao et al., 2017; Vandeputte et al., 2012). In our manuscript, we focused primarily on the role of the carbonic anhydrase *NCE103*; however, we are well aware that Efg1 has numerous additional functions beyond the CSP (Glazier, 2022). For example, *efg1*Δ mutants display aggregation phenotypes, which influence virulence *in vivo*, as evidenced by elevated fungal burden in the brain when compared to the wild-type strain (**Fig. S6E**). This observation is fully consistent with reports demonstrating that flocculation mutants exhibit increased virulence in brain infections but reduced colonization of other organs (Bing et al., 2024).

2. The role of NCE103 and its homologs in fungal growth in ambient air has been demonstrated previously. The importance of bicarbonate for metabolism is also known. The fact that high levels of CO₂ can restore *nce103* mutant phenotype is expected. This is also consistent with no so drastically reduced fungal burden of the *nce103* mutant in the skin infection models (Figure 4D-E). This is also consistent with the lack of virulence phenotype of cryptococcal carbonic anhydrase mutants. Thus, it feels a bit forced for the authors to conclude that CSP is critical for skin infection.

Response: We agree with Reviewer #2 that the CSP does not have a significant impact in the intradermal infection model. However, it plays a more pronounced role in skin surface colonization. Indeed, this observation aligns very well with previous findings in *Cryptococcus*, as pointed out by Reviewer #2. We reworded relevant sections using “skin colonization” instead of “virulence” to avoid misunderstandings. The results in lines 282-284 and 293-296 also changed accordingly as mentioned above.

Regarding *nce103*Δ, in the short-term colonization model (48 hours, Fig. 6F) used before (Deng et al., 2024; Santana et al., 2023), we observed almost a pronounced 10-fold reduction in fungal burden. In contrast, we did not observe significant differences in the long-term colonization model (14 days), possibly because only skin-tolerant fungal populations can grow and persist on the surface over extended periods. In clinical settings, decolonization regimens are typically applied daily (Swan et al., 2016). Hence, the reduction in fungal burden observed at 48 hours (**Fig. 6F**) could represent a clinically relevant window for therapeutic interventions to achieve efficient skin disinfection. However, we note clinically useful *NCE103* inhibitors are as yet unavailable for fungal pathogens.

Our data show that fitness defects of *efg1*Δ are particularly striking in the long-term colonization model, suggesting that Efg1 is a promising target suitable for skin decolonization strategies. While transcription factors like Efg1 are generally considered undruggable owing to their pleiotropic functions and lacking defined binding pockets, recent advances in molecular glues-mediated targeted protein degradation (Bhat et al., 2024; Samarasinghe and Crews, 2021) may offer a promising direction for developing suitable therapeutics.

3. The authors showed that the sterol composition is similar between AMB resistant and sensitivity strains. It is recently shown that the inositolphosphorylceramide level has significant impact on AMB susceptibility. Given that both ergosterol and IPC are important components of fungal membrane and they affect various transporters and susceptibility to various drugs, the potential impact of CSP on both lipid components should be examined. The hypothesis that CSP may change the composition of membrane, which may indirectly affect mitochondrial function, should be considered. The evidence that CSP disruption impairs mitochondrial functions is not convincing based on the ROS evidence. The sensitivity to AA could also be an indirect effect because of altered membrane.

Response: We appreciate this insightful remark, and we agree with Reviewer #2 that IPC-related membrane lipids may be crucial for resistance development, as also elegantly shown recently for *Cryptococcus* (Chen et al., 2024). Of note, we explored several approaches to unravel underlying mechanisms. Identifying additional downstream CSP targets involved in AMB resistance has been extremely challenging, also reflected by the fact that we identified more than 100 transcriptional regulators to be implicated (**Fig. S6G**). We initially found that ROS accumulation might contribute to elevated sensitivity upon CSP disruption. However, treatment with Vitamin C only partially restored AMB-resistance phenotypes, implying involvement of additional mechanisms. To define the role of mitochondrial ROS, we constructed deletion mutants lacking the alternative oxidase *Aox2* or the superoxide dismutase *Sod2*. The mutants displayed severe sensitivity to antimycin A (AA) but had no effect on AMB susceptibility. Therefore, we conclude that while ROS is a contributing factor, other mechanisms must also play a role. As noted by Reviewer #2, expression of multiple transporter family members changes between the R1 and S strains, particularly when R1 is exposed to AMB, where the response is much stronger than in the S strain. Dysregulation of several additional membrane proteins is also evident (**Fig. R8**). Furthermore, Gcn4-mediated regulation of amino acid metabolism may also be implicated (**Fig. R8B**). Despite extensive efforts to functionally validate the highlighted genes (**bold-face genes in Fig. R8**), most single deletion mutants showed discernible effects, pointing towards genetic redundancy and combinatorial actions possibly driving these phenotypes.

Figure R8. AMB treatment alters the proteome of the resistant strain more significantly than that of the sensitive strain. A. Volcano plot showing comparative proteomic data of R1 and S strains under AMB treatment. Genes in bold were functionally validated either in previous literature or in this study. Among them, *PDR16* and *NCE103* contribute to AMB^R. Deletion of *HPA2*, *CFL4*, *SMF12*, *GCN4*, and *B9J08_000008* did not alter AMB MICs. **B.** Network of proteomic changes related to amino acid metabolism in response to AMB stress. These new data were not included in the initial manuscript but were now added to **Fig. S1F** of the revised manuscript.

Changes in revised manuscript, lines 170-171: In our proteomics experiment, AMB stress-induced Nce103 abundance in strain S, but failed to reach Nce103 levels in R1 (**Fig. S1F**).

With respect to mitochondrial function, we also investigated *GOA1*, a fungal-specific regulator of mitochondrial complex I (Bambach et al., 2009; Li et al., 2011). Despite multiple attempts to ablate *GOA1* in both the R1 and AR387 backgrounds, we were unable to obtain successful deletion mutants for this gene. We also applied the Tet-Off system, but after screening over 1,000 colonies, no correct genomic integrations were identified. However, we were able to generate an ectopic overexpression strain of *GOA1*, although it showed no significant change in the MIC for AMB. The eGoa1 strain exhibited slightly reduced fitness, with viability observed but not formally recorded. We provide the results of relevant experiments for the inspection by Reviewer #2 and the Editor (**Fig. R9**), but we have not included this data in the revised manuscript.

Figure R9. Overexpression of *GOA1* does not rescue the effects of CSP disruption on AMB susceptibility, suggesting that *Goa1* does not represent a relevant function for AMB^R.

The most striking effect observed in our phenotyping experiments was with antimycin A, which targets the cytochrome *bc1* complex (Salama et al., 2024; Vincent et al., 2016). To further investigate the underlying mechanism, we attempted to delete *RIP1*, a core component of cytochrome *bc1* – Complex III (Vincent et al., 2016). Despite several attempts, we were unable to generate a *rip1Δ* mutant in the R1 background, similar to our experience with *GOA1*. However, we successfully deleted *RIP1* in the AR387 background, and this new data is presented in the new **Fig. 5** of the revised manuscript. The *RIP1* deletion resulted in approximately a two-fold reduction in the AMB MIC in AR387 (**Fig. R10A**).

Due to the toxicity of the model compound antimycin A, which precludes its clinical application, we tested Inz-5, a fungal-specific cytochrome *bc1* inhibitor (**Fig. R10B**). Inz-5 effectively inhibited WT strains and had no significant additional effect on *rip1Δ*, confirming its on-target activity. Furthermore, combining Inz-5 with AMB resulted in a 2-4-fold reduction in the AMB MIC in other fungal species. Importantly, Inz-5 is well tolerated in mice, suggesting its potential as a lead compound for clinical translation to enhance AMB efficacy (Vincent et al., 2016). It is worth noting, however, that we cannot attribute all effects of the Inz-5 AMB combination solely to cytochrome *bc1* inhibition. This is particularly true since the enhancement of AMB efficacy in AR387 was even more pronounced than in the *rip1Δ* strain, despite Inz-5 being a relatively specific inhibitor of fungal cytochrome *bc1*. In conclusion, targeting the cytochrome *bc1* component Rip1 may offer a promising therapeutic strategy to potentiate AMB.

Figure R10. Targeting mitochondrial cytochrome *bc1* enhances AMB susceptibility. **A.** Deletion of *RIP1* encoding the cytochrome *bc1* complex active subunit increases susceptibility to AMB. MIC assays were performed for *C. auris* in RPMI, YPD and YP-2% glycerol (YPG). **B.** The cytochrome *bc1* inhibitor Inz-5 partially phenocopies the sensitivity of the *rip1Δ* mutant in a serial dilution spotting assay, both in the presence and absence of AMB. **C.** Inz-5 reduces the AMB MIC by approximately 2- to 4-fold across multiple fungal pathogens in checkerboard assays. Two to three biological replicates were performed (A-C). FICI: Fractional inhibitory concentration index.

Changes in revised manuscript, lines 234-246: Given that CSP dysfunction is linked to mitochondrial impairment via the antimycin A target cytochrome *bc1*, we attempted to delete *RIP1*, the catalytic subunit of complex III. Despite multiple efforts to ablate this gene in the R1 strain or its isogenic CSP mutants, we were unable to obtain correct genomic integrations. This may be due to the predominance of non-homologous end joining and the high frequency of random integration of drug resistance markers in certain clinical strains⁵³. However, we successfully obtained *rip1Δ* deletions in the AR387 (B8441) background strain. The *rip1Δ* mutants had an approximately 2-fold reduction in AMB^{MIC} (**Fig. 5A, B**). Owing to the antimycin A toxicity for humans, we employed the fungal-selective cytochrome *bc1* inhibitor Inz-5⁵⁴ to assess its potential in enhancing AMB efficacy in resistant strains. Indeed, Inz-5 reduced AMB^{MIC} in several *Candida* species, as well as in *Cryptococcus neoformans*. Checkerboard assays suggested that the effects were ranging from indifferent to synergistic based on the calculated FICIs (**Fig. 5C**). Therefore, the Rip1 mitochondrial cytochrome *bc1* component could be a promising target for potentiating AMB susceptibility in clinical settings.

As mentioned earlier, we fully agree with Reviewer #2 that it is crucial to consider changes in membrane lipids beyond sterols, such as inositol phosphorylceramide (IPC) (Chen et al., 2024). Notably, we have recently shown that increasing in gene dosage of *PDR16*, a phosphatidylinositol transfer protein, elevated AMB MICs by 4–8 fold (Phan-Canh et al., 2025a), although its deletion did not affect the AMB MIC. In response to Reviewer 2’s suggestion, we collaborated with a renowned lipidomics expert to perform global lipidomics analysis on the *nce103Δ* and *nce103ΔNCE103* (WT) strains, focusing specifically on IPC-related lipids. The results did not reveal significant differences in the overall lipid species between *nce103Δ* and WT (**Fig. R11A**). To account for biological variability, we generated data from 5–7 biological replicates. Although some tendencies are recognizable, the variations likely reflect the inherent noise typically associated with lipid quantification of low-abundance lipids from heterogenous cell populations (Hall et al., 2010). Similarly, no notable differences were observed in the overall lipid class distributions (**Fig. R11C**). However, PCA analysis revealed that data from *nce103Δ* mutant samples were much noisier and showed a slight separation from the WT following AMB treatment (**Fig. R11B**), suggesting that the dynamics of lipid changes may contribute to AMB susceptibility.

Furthermore, we investigated the combination of AMB with aureobasidin A (AbA), a specific inhibitor of the IPC biosynthesis pathway. It has been reported that AbA significantly decreases the MIC of AMB against *Cryptococcus neoformans* (Chen et al., 2024). Notably, both wild-type and mutant strains showed a 2-fold decrease in AMB MIC with AbA treatment. These findings suggest that mitochondrial function, rather than IPC levels, may be crucial in determining AMB^R. AbA appears to exert a fungistatic effect, as *Candida* cells recovered growth within 48 hours. We present these results for the scrutiny of Reviewer #2 and the Editor (**Fig. R11**), and have incorporated the new data into the revised manuscript as **Fig. 3**.

Figure R11. Lipidomics. **A–C.** Global lipid profiling of *C. auris* cultured in RPMI medium with or without AMB treatment. Fungal cells were grown in RPMI for 5 hours prior to treatment with 1 μg/mL AMB for 2 hours, followed by lipidomic analysis. Data represents the relative abundance of each lipid species as a proportion of the total lipid content. **A.** Heatmap displaying lipid species across samples scaled as Z-scores. **B.** PCA analysis of the lipidomics dataset was performed using the *prcomp* function in R. **C.** Proportion of sphingolipid classes within the total lipid pool. Certain classes were summed up data from their relative species. Cer: ceramide; HexCer: hexosylceramide; IPC: inositol phosphorylceramide; MIPC: mannosylinositol phosphorylceramide. Two-way ANOVA with Bonferroni correction was used for multiple comparisons. Five to seven biological replicates were performed. **D.** Drug combination assay of AMB and aureobasidin A (0–1 μg/mL) in *C. auris* using RPMI at 37 °C. The pink circles refer to MIC-reading wells. Two biological replicates were performed.

Changes in revised manuscript, lines 186-195: Growing evidence suggests that inositol phosphorylceramides (IPC) contribute to AMB tolerance⁵⁰, and dynamic changes in membrane lipid distribution may also affect AMB susceptibility^{17,28}. Therefore, we performed global lipidomics of the *nce103Δ* mutant and the *nce103Δ NCE103* control. We did not observe significant differences in membrane lipid species between the reconstituted and mutant strains (**Fig. 3C**). Similar results were observed for sphingolipids such as IPC (**Fig. 3D**). The data

phenocopied the effects of the inhibitor aureobasidin A (AbA), which showed no significant differences between WT and CSP mutants (**Fig. 3E**). However, PCA analysis revealed that samples from the *nce103Δ* mutant were more dispersed and showed a slight separation from the WT strain following AMB treatment (**Fig. 3F**), suggesting that dynamic lipid changes may contribute to AMB susceptibility.

Changes in revised manuscript, lines 351-363: Although the precise mode of AMB action remains ill-posed, the consensus suggests that the primary target of AMB is the lipid bilayer, where it binds or sequesters ergosterol-related lipids to exert its fungicidal activity. Surprisingly, our data revealed no significant differences in ergosterol levels or IPC-related lipids. It is tempting to speculate that the AMB – lipid interactions may cause dynamic changes in the lateral or transversal arrangement of ergosterol and phospholipids in the plasma membrane without affecting overall lipid levels. For instance, “sterol-sponges” may lead to membrane instability and the breakdown of the electrochemical gradient⁶⁹. Further, we hypothesize that the dynamic mosaic organization of lipid bilayers may be influenced by CSP and/or mitochondrial function. This is supported by our lipidomic analysis, which showed substantial fluctuations in lipid landscapes across biological replicates upon *NCE103* deletion (**Fig. 3F**). Additionally, deletion of *RCA1* or *EFG1* led to dysregulation of multiple membrane transporters, many of which have been implicated in multidrug resistance phenomena (**Fig. S7**). These observations suggest a potential mechanistic link between AMB^R and membrane lipid homeostasis.

Alterations in lipid composition may be highly dynamic and challenging to track or quantify, as changes can occur within seconds or even shorter time frames. However, *nce103Δ* mutant strains are highly sensitive to SDS, implying that membrane lipid as well as cell wall organization are involved. This dynamic behavior may also be observed in a fluorescein diacetate (FDA) uptake assay, which specifically measures protein-independent lipid-dependent membrane permeability (Shivarathri et al., 2022). Although we observed increased FDA uptake in some independent experiments with the mutants, we were unable to consistently reproduce these results across many replicates and strains. We provide these FDA results here for the inspection by Reviewer #2 and the Editor (**Fig. R12**), but we did not include this data into the revised manuscript for the above reasons.

Figure R12. Membrane lipid permeability was assessed by a fluorescein diacetate (FDA) uptake assay. Two-way ANOVA of mean fluorescence intensity (MFI) for time x strains followed by Tukey’s multiple comparison tests using R1 as control group. (data from 1 independent experiment).

4. One thing that is missing is the experiment to determine if AMB therapy will make a difference in vivo when the Nce103 is blocked.

Response: Thank you for this valuable suggestion. Initially, we did not perform this experiment because systemic infection occurs in a high-CO₂ environment, which could potentially compensate for CSP dysfunction. However, we agree with the Reviewer #2’s point and were also interested in exploring whether AMB, with its rapid fungicidal activity, might have a synergistic effect in this context. Hence, we conducted a systemic infection experiment with both WT and *nce103Δ* strains, with and without AMB treatment. We administered AMB at a dose of 5 mg/kg body weight, avoiding higher doses due to toxicity concerns. At this dose, we did not observe a striking therapeutic effect of AMB in the treatment groups, except for a reduction in fungal burden in the kidneys. Importantly, there were no significant differences in fungal burden between the *nce103Δ* and WT groups under these conditions. We provide these results here for the inspection by Reviewer #2 and the Editor (**Fig. R13**) and have added the new data into the new **Fig. S6F** of the revised manuscript.

Figure R13. Fungal burden in different organs following systemic *C. auris* infection in mice. Mice were infected intravenously with *C. auris* strains and, 24 hours post-infection, treated with AMB (5 mg/kg) once daily for 2 consecutive days. Fungal burden was assessed in various organs 3 days post-infection.

Note: A 3-day infection model was used because extended infection periods may be confounded by compensatory effects of metabolically produced CO₂, potentially masking the primary effect of CSP.

Changes in revised manuscript, lines 382-384: By sharp contrast, the lack of *RCA1* or *EFG1* or *NCE103* does not cause significant differences in fungal loads recovered from various organs after systemic infections even when mice were treated with AMB (Fig. S6E, F).

Specific concerns:

1. Figure 3E: why growth of the *rca1*, the *efg1*, and the double mutant is the best at intermediate AMB concentration with a relatively lower dose of Antimycin A?

Response: In Fig. 3E (now Fig. 4E), the relative growth of the mutants is shown. The growth of *rca1*Δ, *efg1*Δ, and the *rca1*Δ*efg1*Δ mutants is particularly influenced by CO₂ levels. We have observed that growth can vary across the same 96-well plate, with the center typically showing better growth, likely due to CO₂ from surrounding metabolic activity. However, these variations do not affect the MIC values, as we observed no significant changes even after 48 hours or longer. This highlights the challenge of working with *rca1*Δ and *efg1*Δ mutants, which are hypersensitive fluctuations in CO₂ levels, and their growth is tightly correlated with CO₂ availability.

2. Figure S5A: (1) Both S strain and *nce103* mutant in R1 strain background showed restored resistance to AMB at 5.5% CO₂ level. However, *efg1* sensitivity to AMB is not restored by CO₂, even with overexpression of *NCE103*. (2) The AMB sensitivity of *aox2* and *sod2* mutants did not change much in the presence of high levels of CO₂, arguing against the idea that they are affected by the bicarbonate levels.

Response: We appreciate this valid comment and agree that it raises an interesting point. Here is our response:

First, 5.5% CO₂ and overexpression of *NCE103* fully restore AMB^R in both *rca1*Δ and *efg1*Δ strains. However, while high CO₂ concentration and ectopic *eNCE103* partially restore the fitness of *efg1*Δ mutants in the presence of antimycin A (Fig. S4A), this suggests that *Efg1* contributes to fitness through both CSP-dependent and CSP-independent mechanisms. Notably, *Efg1* is involved in regulating multiple pathophysiological processes in *C. albicans*, including mitochondrial function (Glazier, 2022; Tao et al., 2017).

Second, ectopic overexpression of *NCE103* fully rescued the growth of *rca1*Δ to wild-type levels in the presence of antimycin A, but only partially rescued *efg1*Δ fitness. This may indicate that *Rca1* plays a key role in regulating *NCE103*, while *Efg1* influences its expression. As shown in Fig. R1, deletion of *RCA1* and *EFG1* reduces *NCE103* expression under both low and high CO₂ conditions. *In silico* predictions suggest that *Rca1* and *Efg1* may bind to the promoter regions of *NCE103*.

Third, the deletion of *AOX2* and *SOD2* does indeed significantly affect fitness in the presence of antimycin A, but this appears to be independent of the CSP. As discussed earlier, our aim was to dissect the downstream mechanisms by which mitochondrial function influences AMB^R. We selected *AOX2* and *SOD2* as potential targets based on previous findings showing that elevated expression of *AOX2* is associated with increased AMB resistance in *C. auris* (Chauhan et al., 2025). Unexpectedly though, the deletion of these genes did not impact on the AMB MIC, suggesting genetic redundancy. Taking together, our data strongly support the conclusion that targeting the cytochrome *bc1* complex III impacts AMB susceptibility in *C. auris* (Fig. R10).

3. It would be useful to survey the AMB resistant isolates used in Figure 2C to examine how common NCE103 is involved in AMB resistance (e.g. by qPCR) as RCA1 and EFG1 ultimately need to alter the function of NCE103 for their CSP specific effect. Given that Nce103 is essential in ambient air, the likely impact is on its expression level.

Response: Thank you for this insightful suggestion. As recommended, we analyzed the mRNA levels and AMB resistance phenotypes of 22 clinical strains representing various clades. Our analysis revealed that a subset within clade I exhibited high AMB resistance, which was associated with elevated *NCE103* expression (**Fig. R14**). However, the overall Pearson correlation coefficient was moderate ($R^2 < 0.5$), suggesting a potential but not strong correlation. This could be due to the narrow effective window of AMB or the presence of additional mutations that might mask the effect. For example, a loss-of-function mutation in *ERG6* or *NCP1* could potentially mask the effect of CSP on AMB^R, as highlighted in **Fig. R4** of the response letter.

Changes in revised manuscript, lines 156-159: Therefore, we reasoned that mRNA expression or gene dosage of *NCE103* might affect AMB^R traits. Indeed, a subset of AMB-resistant strains from Clade I exhibited high *NCE103* expression (**Fig. 2F**), and the correlation between *NCE103* mRNA levels and AMB^R was statistically significant but modest (Pearson's $R^2 = 0.19$, $p = 0.045$).

Figure R14. Pearson correlation between *NCE103* mRNA expression and AMB resistance among clinical *C. auris* isolates. AMB resistance levels were semi-quantified using serial dilution assays on AMB-containing plates. Two AMB concentrations (1 $\mu\text{g/ml}$ and 2 $\mu\text{g/ml}$) were used to capture a broad range of susceptibilities. Spotting assays were chosen over broth dilution MIC methods due to their higher resolution in evaluating compounds with a narrow effective window as AMB. Two biological replicates were performed, yielding consistent results.

4. In Table S1, CLSI and EUCAST methods generated similar MICs (mostly within 2-4 fold). However, there are some large differences between the two methods. For example, MICs for VRC of the sensitive strain showed greater than 32 fold difference. Similarly, MICs for CAS of the strains AR285 and AR1097 are more than 32 fold different. Explanation?

Response: Thank you for this question. One key difference between the CLSI and EUCAST is the use of 2% glucose in the EUCAST protocol, which can influence antifungal susceptibility by affecting metabolism, cell wall structure and composition (Ene et al., 2012). Moreover, discrepancies in caspofungin MIC-testing are notorious and well-reported across laboratories (Pfaller et al., 2014). Hence, substituting caspofungin with micafungin for susceptibility testing is often recommended to reduce variability (Pfaller et al., 2014).

5. Figure S3C: The authors stated that “both *rca1* Δ and *efg1* Δ mutants showed slightly increased susceptibility to 5-fluorocytosine (5FC), although not significant to caspofungin (CAS) or voriconazole (VRC) (Fig. S3C).” The CAS test did not reach the concentration that causes significant reduced growth for R1, and thus it is hard to tell. However, it looks like that there should be at least 2 or 4 fold difference with *rca1* or *efg1* mutations. Based on the graph of relative growth, both *rca1* Δ and *efg1* Δ mutants showed dramatically increased sensitivity to 5FC (8 fold for *efg1* and 4 fold for *rca1*). This is more than they observed with AMB, the focus of this story. Are these changes in drug resistance/sensitivity related to bicarbonate levels?

Response: Thank you for highlighting this point. It is well-accepted that 5FC resistance is typically linked to mutations in the 5-fluorouracil (5FU) conversion pathway, as also demonstrated in our recent work (Phan-Canh et al., 2025b) and by others (Samantha et al., 2022; Rhodes et al., 2024). Resistance usually manifests itself as a high-level MIC shift (>64 $\mu\text{g/ml}$) (Chowdhary et al., 2018). The strain used in our study is not intrinsically resistant

to 5FC, so the observed MIC reduction is unlikely to have a strong clinical relevance, especially as 5FC resistance is primarily acquired. We also noted a pronounced trailing effect in 5FC MIC assays. The inhibition was not fungicidal but more likely fungistatic, as mutant strains showed regrowth at supra-MIC concentrations after 48 hours. Given the absence of a robust phenotype, we chose not to pursue this observation further.

6. Ptc2 functions in sensing high levels of CO₂ to control fungal response to high levels of CO₂ (like morphogenesis in *Candida albicans*). It is obviously not the same as the CSP pathway where conversion of CO₂ to bicarbonate is critical when the CO₂ level is very low. It is unclear how including this adds to this study.

Response: Thank you for this thoughtful comment. Indeed, we included this experiment. While the result was negative, it supports our conclusion that AMB^R is mediated by *NCE103*, rather than through direct Ptc2-mediated CO₂-sensing, implying that the two pathways are not epistatic and thus operate independently.

7. Have the authors tested e*NCE103* in the R1 strain without other mutations as a control for all the e*NCE103* strains in various mutant backgrounds?

Response: No, we did not generate an ectopic *NCE103* overexpression strain in the R1 background. However, growth under 5.5% CO₂ can serve as a functional proxy for *NCE103* overexpression in this strain. Under these conditions, we observed no significant changes in AMB MICs when compared to low CO₂, suggesting that increased *NCE103* overexpression alone does not further enhance resistance in R1. High CO₂ may still exert a modest to moderate effect on WT strains as shown in **Fig. R15**.

Figure R15. 5.5% CO₂ enhances AMB tolerance in a subset of clinical *C. auris* strains (highlighted in pink). Spot dilution assays were performed using 5-fold serial dilutions of *C. auris* cells spotted onto YPD agar with or without AMB.

Changes in revised manuscript, lines 155-156: We observed that supplying 5.5% CO₂ can enhance AMB^R in several clinical strains (**Fig. 2E**).

8. “Although there was no significant enhancement of AMB MIC in mutants, strain AR387 showed a 2-fold increase when treated with 250-1000 μM VitC. These findings suggest that, in addition to the ROS response, mitochondrial functions may control downstream effects resulting from CSP disruption.” It is not clear to me how the evidence leads to the conclusion that ROS is an important factor in CSP’s impact on mitochondrial function.

Response: Thank you for pointing out this ambiguity, and we apologize for the unclear wording. Our intention was not to imply that ROS is a central component of the CSP pathway. Rather, our data indicates that Vitamin C only partially rescues the AMB-resistance phenotype in *RCA1* and *EFG1* null mutants, suggesting that ROS only contributes modestly. The pronounced sensitivity observed under antimycin A hints a broader role of mitochondrial dysfunction beyond ROS. Furthermore, our new data on the ablation of *RIP1* provides compelling evidence that mitochondrial function plays a pivotal role in amphotericin B susceptibility.

Changes in revised manuscript, lines 223-224: These findings suggest that ROS accumulation alone cannot fully explain the AMB sensitivity that results from disabling the CSP (**Fig. 4B**).

9. Figure 4A. Have the authors tested preculturing the strains in high level of CO₂ before the growth assay? Will such pre-adaptation reduce the difference in MM media?

Response: To address this question, we conducted additional spot assays. *C. auris* was initially grown on YPD agar at 5.5% CO₂ for two days. Serial 5-fold dilutions were then prepared and pre-incubated for 2 hours at 5.5% CO₂ prior to spotting onto minimal medium (MM) agar (**Fig. R16**). The resulting growth patterns were consistent with those observed in our original experiment shown in **Fig. 4C (now Fig. 6C)**. We provide these results here for the inspection by Reviewer2 and the Editor (**Fig. R16**) but do not include them into the revised manuscript.

Figure R16. *C. auris* was pre-grown in 5.5% CO₂ atmosphere before spot cell suspensions onto MM agar and incubation for 3 days.

Note: The differences between this picture and the related panel in Fig. 6C is due to variation among independent replicates and across culture plates.

10. Figure 4D-E: Could authors clarify if the *nce103*Δ*NCE103* used in animal models is a complemented strain with *NCE103* driven by its native promoter or it is an *NCE103* overexpression strain in the *nce103*Δ mutant background?

Response: Yes, the *NCE103* gene present in the *nce103*Δ *NCE103* strain is expressed under the control of its native promoter.

Reviewer #2 (Remarks on code availability):

They used standard pipelines for proteomics and RNA seq analyses.

Response: Thank you for confirming.

Reviewer #3 (Remarks to the Author):

The manuscript investigates the role of the carbonic sensing pathway (CSP) in antifungal resistance and fitness of *Candida auris*, focusing on the carbonic anhydrase Nce103 and its regulators Rca1 and Efg1. The authors provide compelling evidence that CSP is crucial for amphotericin B resistance (AMBR), mitochondrial function, and fungal adaptation to skin environments. They propose that bacterial urease-derived CO₂ may enhance *C. auris* colonization, suggesting potential therapeutic strategies targeting CSP components.

The experimental design is robust, combining transcriptomics, proteomics, gene deletion, pharmacological inhibition, and in vivo/in vitro models. The findings are largely supported by data, and conclusions appear justified. The manuscript does not have apparent fatal flaws that would prohibit publication, though some aspects require additional clarification. In particular, the proteomics and RNA-seq experiment section would benefit from further details about the following:

Response: We highly appreciate the thoughtful and encouraging feedback from Reviewer #3. We appreciate the recognition of our experimental approach and the resulting conclusions. Of note, we have performed additional experiments and included additional data where appropriate or requested to address specific concerns.

• What statistical test was employed for the differential expression analysis?

Response: Thank you for the question and we apologize for the lack of detail. We have updated the Method section accordingly. For RNA-seq data, differential expression analysis was performed based on EdgeR using quasi-likelihood F-test (glmQLFTest).

Changes in revised manuscript, lines 620-621: Differential expression analysis was conducted using EdgeR v3.40.2⁹⁰, including the quasi-likelihood F-test (glmQLFTest) approach.

For proteomics datasets, the differential analysis was performed using the LIMMA (Linear Models for Microarray Data) package in R integrated into the Cassiopeia_LFQ package. Cassiopeia does statistical comparisons of groups using LIMMA from the R Bioconductor repository. Similar to classical t-tests, LIMMA tests for the equality of norm intensity means in two different groups for each protein. This is described in lines 573-580 of the main text. A full description is available at https://github.com/moritzmadern/Cassiopeia_LFQ.

• Was the data normalised and/or log transformed before statistics? If yes, what method of normalisation was used?

Response: We updated the methods section accordingly. Yes, RNA-seq data were normalized using the TMM method via calcNormFactors() in EdgeR. Counts were not log-transformed prior to testing, as EdgeR uses raw counts with appropriate modeling (e.g., glmQLFTest), which is the default approach in EdgeR.

Changes in revised manuscript, lines 620-627: Differential expression analysis was conducted using EdgeR v3.40.2⁹⁰, including the quasi-likelihood F-test (glmQLFTest) approach. Raw read counts were normalized using the TMM (Trimmed Mean of M-values) method via calcNormFactors() in EdgeR. The false discovery rate (FDR) was controlled by adjusting p-values using the Benjamini–Hochberg correction. Principal component analysis (PCA) was conducted using CPM-normalized expression values obtained from the *cpm* function in EdgeR, applying TMM normalization (normalized.lib.size=TRUE). Genes with zero expression across all samples were removed, and PCA was conducted using the *prcomp* function in R (stats v3.4.1) with centering and scaling.

For proteomics analysis, the data were normalized using the MaxLFQ function in MaxQuant (Cox et al., 2014). Output data from MaxQuant were further filtered, log-transformed and statistically analyzed as described in lines 568-570 of the revised manuscript.

Changes in revised manuscript, lines 568-570: For label-free quantification, we activated “match between runs” and “MaxLFQ”, including normalization - all other parameters were used at default settings.

• After splitting the data into single CV and processing each one separately through MaxQuant, how were the results combined and normalised?

Response: Thank you for bringing this up, and apologies for the incomplete description in the original manuscript. The split raw files were processed together in a single search, but specified as two different fractions in the experimental design (CV -45V and CV -60V). We have now added a proper description in the Methods section.

Changes in revised manuscript, lines 570-572: Split raw files were processed in a single combined search, with each file assigned a fraction number corresponding to its CV value so as to ensure that match-between-runs was performed only within the same CV group.

• Add a reference for the contaminants database.

Response: We completely agree that this important information is often missing in publications. Therefore, we have recently released the contaminant database on our GitHub page (<https://github.com/maxperutzlabs-ms/perutz-ms-contaminants>). It is based on the contaminant file delivered with MaxQuant, but was further curated and improved over time. We have now added the link to the GitHub repository to the manuscript (line 565).

• The actual code for proteomics data analysis is not disclosed in github or elsewhere.

Response: Thank you for pointing this out – we apologize for the omission. The code used for the proteomics analyses is now available under the link <https://doi.org/10.5281/zenodo.5758974>, and this link has now been included in both the Methods and Code Availability section to provide full access (lines 887-888).

• Although figure S1 mentions biological triplicates, more details on replicates for proteomics and RNA-seq in the methods section would be helpful.

Response: Thank you, we added the requested information into the Methods section.

Changes in revised manuscript, lines 528-529: Three biological replicates were performed for each experimental group.

Changes in revised manuscript, line 604: At least three biological replicates were subjected to RNA-seq.

I was not able to access the data in PRIDE as the provided username and password did not work.

Response: We are sorry for this problem. The datasets are available at <https://www.ebi.ac.uk/pride/archive/projects/PXD048342/>

On line 76, I suggest you remove the statement that “proteomics may identify key genes” and leave it at key pathways.

Response: We have revised the statement as requested.

On line 454, change wording to “Fungal cells were grown...”

Response: The wording has been corrected as suggested.

In fig. 1B., the authors have used a picture of a randomly-selected Fusion Lumos mass spectrometer. It is advisable (but not strictly necessary) to use a picture of the machine that was actually used in the experiments, that is the Exploris 480.

Response: Thank you for the helpful suggestion. We have updated the image to show the Exploris 480 used in our experiments.

Other potential points to address may include:

- Are the fitness defects of *rca1*Δ and *efg1*Δ mutants fully attributable to mitochondrial dysfunction?

Response: Based on our current data, the fitness defect in the *rca1*Δ mutant appears to be largely attributable to loss of *NCE103*. In contrast, *NCE103* overexpression only partially rescued the defect in the *efg1*Δ mutant, suggesting additional as yet unknown mechanisms. At this stage, we are unable to determine the extent to which mitochondrial dysfunction contributes to the fitness defects observed in these mutants. Addressing this will require generating double mutants with key mitochondrial genes in future efforts.

While we have attempted to delete several such genes (*AOX2*, *SOD2*, *GOA1*, and *RIP1*), the complexity of mitochondrial function necessitates a systematic approach. We successfully generated *rip1*Δ mutants in the AR387 background, which exhibited severely impaired fitness (**Fig. R10**). These findings support the idea that mitochondrial function and metabolism are promising avenues for antifungal development (Chauhan et al., 2025; Salama et al., 2024).

- How do CSP-related AMB resistance mechanisms compare to those in clinical isolates with known resistance mutations?

Response: We appreciate the opportunity to clarify this point. Our findings suggest that CSP-related mechanisms contribute to the intrinsic high AmB tolerance of *C. auris*, and potentially in other members of the *C. haemulonii* complex. Notably, this intrinsic trait is not commonly seen in other *Candida* species (Ben-Ami et al., 2017; Francisco et al., 2023). In contrast, in-depth knowledge about mechanisms of clinical AMB resistance in *C. auris* has been scarce. Typically, mutations in the ergosterol biosynthesis pathway usually acquired under drug pressure (Carolus et al., 2024; Rybak et al., 2022) can explain AMB^R. However, acquired mutations alone cannot fully account for the elevated baselines in AMB MICs frequently observed in many clinical isolates. Although we have not yet generated double mutants, combining CSP deletions with mutations in the ERG and/or ICP pathways may advance the field. Of note, both R1 and R2 strains used in our study harbor the *Erg11*^{K143R} mutation – commonly associated with azole resistance but not with AMB resistance (Rybak et al., 2022). We provide these results here for the inspection by Reviewer #3 and the Editor (**Fig. R17**) but do not include them into the revised manuscript.

Figure R17. Visualization of RNA-seq data using the IGV revealed the presence of the K143R mutation in the *ERG11* gene in strains R1 and R2.

Interestingly, based on published NCBI genome data, we identified a *C. auris* strain carrying loss-of-function mutations in both CSP genes and *ERG6* or *NCP1*, which resulted in AMB^R (**Fig. R4B**). This suggests that rarely acquired resistance mechanisms through the ERG pathway may mask the effects of CSP disruption. We include this genomic finding as exploratory data, and we believe that further research will benefit from the ever-growing number of publicly available *C. auris* WGS datasets. It will be necessary to systemically assess the interplay between CSP and mechanisms of acquired AMB resistance.

- Could the CSP be targeted in therapeutic interventions without disrupting host CO₂ metabolism?

Response: Thank you for this important question. We agree that directly targeting the CSP may not be suitable for systemic infections, owing to elevated CO₂ levels (e.g., 5.5%) in internal organs that may restore AmB resistance. Further, such interventions could also potentially interfere with host CO₂ homeostasis (**Fig. R13**).

Instead, we propose that mitochondrial function (**Fig. R10**) and sphingolipid metabolism (**Fig. R11D**) may offer better therapeutic targets.

For example, we tested Inz-5, a fungal-specific cytochrome *bc1* (complex III) inhibitor, which enhanced AMB activity with a 2-4-fold reduction of MICs across multiple fungal species. Similarly, the deletion of *RIP1*, a core subunit of cytochrome *bc1*, resulted in a 2-fold MIC reduction in the AR387 strain.

In line with a remark from Reviewers, we also tested Aureobasidin A (AbA), a specific inhibitor of inositol phosphorylceramide (IPC) biosynthesis. AbA showed a 2-fold reduction in the AMB MIC during combinatorial treatments (**Fig. R11D**). However, we observed fungal regrowth after 48 hours, indicating a fungistatic rather than fungicidal activity. While this regrowth was not formally recorded, it suggests that IPC-targeting compounds may require optimization or combination strategies to achieve durable antifungal effects.

• Were urease-negative bacterial strains tested for other factors that might affect fungal growth?

Response: This is a very interesting point, indeed. However, our work primarily aimed to demonstrate that bacteria-derived CO₂ can promote the growth of *C. auris* via the CSP. To establish this proof-of-concept, we focused interkingdom interactions involving urease activity by employing a *Proteus mirabilis ureC* null mutant alongside the *C. auris nce103Δ* mutants. This approach enabled us to specifically probe the role of CSP in mediating fungal response to bacterial CO₂ on the skin surface microenvironment.

While this *in vitro* interaction was clearly demonstrated, translating it into an *in vivo* context will be more complex. Factors such as community composition, microbiome dynamics, as well as physiological differences such as the absence of eccrine sweat glands in murine back skin (Cui and Schlessinger, 2015) are most likely influencing the outcome. These confounding effects may explain the lack of a significant phenotype in our human vs mouse skin models (**Fig. R6**). Nonetheless, our data are fully in line with clinical observations reporting frequent co-occurrence of *Proteus mirabilis* and *Klebsiella pneumoniae* in *C. auris*-positive samples (Proctor et al., 2021).

We also recognize that urease-negative bacteria may influence *C. auris* through other mechanisms. One promising area is lipid metabolism: microorganisms that secrete extracellular lipases may support *C. auris* growth in skin niches, particularly those mimicking human sweat (Horton et al., 2020). A recent study further demonstrated that lipase activity is essential for *C. auris* proliferation under such conditions (Nicklas et al., 2025), highlighting a potential avenue for future investigation into CSP-independent microbial interactions.

In conclusion, the study presents novel insights into the CSP's role in AMBR and fungal colonization. The integrated proteo-transcriptomics approach is an appropriate preliminary screen, and the *in vivo* skin colonization model strengthens translational relevance. The conclusions align well with the data and the claim that CSP can be a therapeutic target is reasonable, but potential off-target effects of inhibitors should be acknowledged. The study is strong and has substantial merit for publication after addressing the minor points as mentioned above.

Response: We are delighted and grateful for the enthusiastic comments and insightful discussion provided by Reviewer #3. We appreciate the recognition of our integrated proteo-transcriptomics approach and the relevance of our *in vivo* skin colonization model. In response to Reviewer suggestions, we have included a discussion of potential therapeutic strategies, including the use of the lead compound Inz-5 (Vincent et al., 2016). We believe our results provide a strong basis for future translational efforts to improve the efficacy of AMB against *C. auris* and related species within the *C. haemulonii* complex.

Reviewer #3 (Remarks on code availability):

I have not reviewed the RNA-seq code because this is not my area of expertise. I would be happy to review the proteomics data analysis code if the authors provide it. I would also be happy to review the PRIDE submission if you give me a working username and password.

Response: We apologize for the inconvenience. The PRIDE dataset is now publicly available. The code used for the proteomic analysis is available at <https://doi.org/10.5281/zenodo.5758974> and also accessible via GitHub at https://github.com/maxperutzlabs-ms/Cassiopeia_LFQ. We used the zenodo system to comply with *Nature's* requirements for code availability, which specify that a DOI must be provided. This pipeline has been developed and is routinely applied by our proteomics facility (Bestehorn et al., 2025; Zhu et al., 2023). We appreciate your support in verifying our analysis and are pleased to have the opportunity to improve our work through the revisions.

REFERENCES

1. Avelar Gabriela M., Pradhan Arnab, Ma Qinxu, Hickey Emer, Leaves Ian, Liddle Corin, et al. A CO₂ sensing module modulates β -1,3-glucan exposure in *Candida albicans*. *mBio* 2024;15:e01898-23. <https://doi.org/10.1128/mbio.01898-23>.
2. Bahn Y-S, Cox GM, Perfect JR, Heitman J. Carbonic anhydrase and CO₂ sensing during *Cryptococcus neoformans* growth, differentiation, and virulence. *Curr Biol* 2005;15:2013–20. <https://doi.org/10.1016/j.cub.2005.09.047>.
3. Bambach A, Fernandes MP, Ghosh A, Kruppa M, Alex D, Li D, et al. Goa1p of *Candida albicans* localizes to the mitochondria during stress and is required for mitochondrial function and virulence. *Eukaryot Cell* 2009;8:1706–20.
4. Ben-Ami R, Berman J, Novikov A, Bash E, Shachor-Meyouhas Y, Zakin S, et al. Multidrug-resistant *Candida haemulonii* and *C. auris*, Tel Aviv, Israel. *Emerg Infect Dis* 2017;23:195.
5. Bestehorn A, von Wirén J, Zeiler C, Fesselet J, Didusch S, Forte M, et al. Cytoplasmic mRNA decay controlling inflammatory gene expression is determined by pre-mRNA fate decision. *Mol Cell* 2025;85:742-755.e9. <https://doi.org/10.1016/j.molcel.2025.01.001>.
6. Bhat S, Palepu K, Hong L, Mao J, Ye T, Iyer R, et al. *De novo* design of peptide binders to conformationally diverse targets with contrastive language modeling. *Sci Adv* n.d.;11:eadr8638. <https://doi.org/10.1126/sciadv.adr8638>.
7. Bing J, Guan Z, Zheng T, Ennis CL, Nobile CJ, Chen C, et al. Rapid evolution of an adaptive multicellular morphology of *Candida auris* during systemic infection. *Nat Commun* 2024;15:2381. <https://doi.org/10.1038/s41467-024-46786-8>.
8. Bing J, Guan Z, Zheng T, Zhang Z, Fan S, Ennis CL, et al. Clinical isolates of *Candida auris* with enhanced adherence and biofilm formation due to genomic amplification of ALS4. *PLoS Pathog* 2023;19:e1011239.
9. de Boer CG, Hughes TR. YeTFaSCo: a database of evaluated yeast transcription factor sequence specificities. *Nucleic Acids Res* 2012;40:D169–79. <https://doi.org/10.1093/nar/gkr993>.
10. Carolus H, Sofras D, Boccarella G, Sephton-Clark P, Biriukov V, Cauldron NC, et al. Acquired amphotericin B resistance leads to fitness trade-offs that can be mitigated by compensatory evolution in *Candida auris*. *Nat Microbiol* 2024;9:3304–20. <https://doi.org/10.1038/s41564-024-01854-z>.
11. Chauhan A, Carolus H, Sofras D, Kumar M, Kumar P, Nair R, et al. Multi-Omics analysis of experimentally evolved *Candida auris* isolates reveals modulation of sterols, sphingolipids, and oxidative stress in acquired amphotericin B resistance. *Mol Microbiol* 2025;n/a. <https://doi.org/10.1111/mmi.15379>.
12. Chen L, Tian X, Zhang L, Wang W, Hu P, Ma Z, et al. Brain glucose induces tolerance of *Cryptococcus neoformans* to amphotericin B during meningitis. *Nat Microbiol* 2024;9:346–58. <https://doi.org/10.1038/s41564-023-01561-1>.
13. Chowdhary A, Prakash A, Sharma C, Kordalewska M, Kumar A, Sarma S, et al. A multicentre study of antifungal susceptibility patterns among 350 *Candida auris* isolates (2009–17) in India: role of the *ERG11* and *FKS1* genes in azole and echinocandin resistance. *J Antimicrob Chemother* 2018;73:891–9. <https://doi.org/10.1093/jac/dkx480>.
14. Cottier F, Raymond M, Kurzai O, Bolstad M, Leewattanapasuk W, Jimenez-Lopez C, et al. The *bZIP* transcription factor Rca1p is a central regulator of a novel CO₂ sensing pathway in yeast. *PLoS Pathog* 2012;8:e1002485.
15. Cox J, Hein MY, Lubner CA, Paron I, Nagaraj N, Mann M. Accurate proteome-wide label-free quantification by delayed normalization and maximal peptide ratio extraction, termed MaxLFQ *. *Mol Cell Proteomics* 2014;13:2513–26. <https://doi.org/10.1074/mcp.M113.031591>.
16. Cui C-Y, Schlessinger D. Eccrine sweat gland development and sweat secretion. *Exp Dermatol* 2015;24:644–50. <https://doi.org/10.1111/exd.12773>.
17. Deng Y, Xu M, Li S, Bing J, Zheng Q, Huang G, et al. A single gene mutation underpins metabolic adaptation and acquisition of filamentous competence in the emerging fungal pathogen *Candida auris*. *PLOS Pathog* 2024;20:e1012362. <https://doi.org/10.1371/journal.ppat.1012362>.
18. Du H, Bing J, Hu T, Ennis CL, Nobile CJ, Huang G. *Candida auris*: Epidemiology, biology, antifungal resistance, and virulence. *PLoS Pathog* 2020;16:e1008921.
19. Ene IV, Adya AK, Wehmeier S, Brand AC, MacCallum DM, Gow NAR, et al. Host carbon sources modulate cell wall architecture, drug resistance and virulence in a fungal pathogen. *Cell Microbiol* 2012;14:1319–35. <https://doi.org/10.1111/j.1462-5822.2012.01813.x>.
20. Francisco EC, de Jong AW, Colombo AL. *Candida haemulonii* species complex: A mini-review. *Mycopathologia* 2023;188:909–17. <https://doi.org/10.1007/s11046-023-00748-8>.
21. Glazier VE. *EFG1*, everyone's favorite gene in *Candida albicans*: A comprehensive literature review. *Front Cell Infect Microbiol* 2022;12:855229.
22. Hall RA, De Sordi L, MacCallum DM, Topal H, Eaton R, Bloor JW, et al. CO₂ acts as a signalling molecule in populations of the fungal pathogen *Candida albicans*. *PLOS Pathog* 2010;6:e1001193. <https://doi.org/10.1371/journal.ppat.1001193>.
23. Horton MV, Johnson CJ, Kernien JF, Patel TD, Lam BC, Cheong JZA, et al. *Candida auris* forms high-burden biofilms in skin niche conditions and on porcine skin. *mSphere* 2020;5:e00910-19. <https://doi.org/10.1128/mSphere.00910-19>.
24. Jacobs Samantha E., Jacobs Jonathan L., Dennis Emily K., Taimur Sarah, Rana Meenakshi, Patel Dhruv, et al. *Candida auris* pan-drug-resistant to four classes of antifungal agents. *Antimicrob Agents Chemother* 2022;66:e00053-22. <https://doi.org/10.1128/aac.00053-22>.
25. Kim Ji-Seok, Lee Kyung-Tae, Lee Myung Ha, Cheong Eunji, Bahn Yong-Sun. Adenylyl cyclase and protein kinase play redundant and distinct roles in growth, differentiation, antifungal drug resistance, and pathogenicity of *Candida auris*. *mBio* 2021;12:10.1128/mbio.02729-21. <https://doi.org/10.1128/mbio.02729-21>.
26. Krysan Damian J., Zhai Bing, Beattie Sarah R., Misel Kara M., Wellington Melanie, Lin Xiaorong. Host carbon dioxide concentration is an independent stress for *Cryptococcus neoformans* that affects virulence and antifungal susceptibility. *mBio* 2019;10:10.1128/mbio.01410-19. <https://doi.org/10.1128/mbio.01410-19>.
27. Lambers H, Piessens S, Bloem A, Pronk H, Finkel P. Natural skin surface pH is on average below 5, which is beneficial for its resident flora. *Int J Cosmet Sci* 2006;28:359–70. <https://doi.org/10.1111/j.1467-2494.2006.00344.x>.
28. Li D, Chen H, Florentino A, Alex D, Sikorski P, Fonzi WA, et al. Enzymatic dysfunction of mitochondrial complex I of the *Candida albicans* Goa1 mutant is associated with increased reactive oxidants and cell death. *Eukaryot Cell* 2011;10:672–82.

29. Nakashima K, Kato H, Kurata R, Qianwen L, Hayakawa T, Okada F, et al. Gap junction-mediated contraction of myoepithelial cells induces the peristaltic transport of sweat in human eccrine glands. *Commun Biol* 2023;6:1175. <https://doi.org/10.1038/s42003-023-05557-9>.
30. Nicklas JP, Deming C, Lee-Lin S-Q, Conlan S, Shen Z, Blaustein R, et al. *Candida auris* metabolism and growth preferences in physiologically relevant skin-like conditions. *bioRxiv* 2025:2025–05.
31. Pelletier C, Shaw S, Alsayegh S, Brown AJP, Lorenz A. *Candida auris* undergoes adhesin-dependent and -independent cellular aggregation. *PLOS Pathog* 2024;20:e1012076. <https://doi.org/10.1371/journal.ppat.1012076>.
32. Pfaller Michael A., Messer Shawn A., Diekema Daniel J., Jones Ronald N., Castanheira Mariana. Use of micafungin as a surrogate marker to predict susceptibility and resistance to caspofungin among 3,764 clinical isolates of *Candida* by use of clsi methods and interpretive criteria. *J Clin Microbiol* 2014;52:108–14. <https://doi.org/10.1128/jcm.02481-13>.
33. Phan-Canh Trinh, Bitencourt Tamires, Kuchler Karl. Gene dosage of *PDR16* modulates azole susceptibility in *Candida auris*. *Microbiol Spectr* 2025a;0:e02659-24. <https://doi.org/10.1128/spectrum.02659-24>.
34. Phan-Canh Trinh, Nguyen-Le Duc-Minh, Luu Phuc-Loi, Khunweeraphong Narakorn, Kuchler Karl. Rapid *in vitro* evolution of flucytosine resistance in *Candida auris*. *mSphere* 2025b;0:e00977-24. <https://doi.org/10.1128/msphere.00977-24>.
35. Pohlrs Susann, Martin Ronny, Krüger Thomas, Hellwig Daniela, Hänel Frank, Kniemeyer Olaf, et al. Lipid signaling via Pkh1/2 regulates fungal CO₂ sensing through the kinase Sch9. *mBio* 2017;8:10.1128/mbio.02211-16. <https://doi.org/10.1128/mbio.02211-16>.
36. Proctor DM, Dangana T, Sexton DJ, Fukuda C, Yelin RD, Stanley M, et al. Integrated genomic, epidemiologic investigation of *Candida auris* skin colonization in a skilled nursing facility. *Nat Med* 2021;27:1401–9. <https://doi.org/10.1038/s41591-021-01383-w>.
37. Rhodes Johanna, Jacobs Jonathan, Dennis Emily K., Manjari Swati R., Banavali Nilesh K., Marlow Robert, et al. What makes *Candida auris* pan-drug resistant? Integrative insights from genomic, transcriptomic, and phenomic analysis of clinical strains resistant to all four major classes of antifungal drugs. *Antimicrob Agents Chemother* 2024;68:e00911-24. <https://doi.org/10.1128/aac.00911-24>.
38. Rybak Jeffrey M., Barker KS, Muñoz JF, Parker JE, Ahmad S, Mokaddas E, et al. In vivo emergence of high-level resistance during treatment reveals the first identified mechanism of amphotericin B resistance in *Candida auris*. *Clin Microbiol Infect* 2022;28:838–43. <https://doi.org/10.1016/j.cmi.2021.11.024>.
39. Rybak Jeffrey M, Cuomo CA, David Rogers P. The molecular and genetic basis of antifungal resistance in the emerging fungal pathogen *Candida auris*. *Curr Opin Microbiol* 2022;70:102208. <https://doi.org/10.1016/j.mib.2022.102208>.
40. Salama EA, Elgammal Y, Wijeratne A, Lanman NA, Utturkar SM, Farhangian A, et al. Lansoprazole interferes with fungal respiration and acts synergistically with amphotericin B against multidrug-resistant *Candida auris*. *Emerg Microbes Infect* 2024;13:2322649.
41. Samarasinghe KTG, Crews CM. Targeted protein degradation: A promise for undruggable proteins. *Cell Chem Biol* 2021;28:934–51. <https://doi.org/10.1016/j.chembiol.2021.04.011>.
42. Santana DJ, Anku JAE, Zhao G, Zarnowski R, Johnson CJ, Hautau H, et al. A *Candida auris*-specific adhesin, Scf1, governs surface association, colonization, and virulence. *Science* 2023;381:1461–7. <https://doi.org/10.1126/science.adf8972>.
43. Shivarathri Raju, Jenull Sabrina, Chauhan Manju, Singh Ashutosh, Mazumdar Rounik, Chowdhary Anuradha, et al. Comparative transcriptomics reveal possible mechanisms of amphotericin B resistance in *Candida auris*. *Antimicrob Agents Chemother* 2022;66:e02276-21. <https://doi.org/10.1128/aac.02276-21>.
44. Swan JT, Ashton CM, Bui LN, Pham VP, Shirkey BA, Blackshear JE, et al. Effect of chlorhexidine bathing every other day on prevention of hospital-acquired infections in the surgical ICU: a single-center, randomized controlled trial. *Crit Care Med* 2016;44.
45. Tao L, Zhang Y, Fan S, Nobile CJ, Guan G, Huang G. Integration of the tricarboxylic acid (TCA) cycle with cAMP signaling and Sfl2 pathways in the regulation of CO₂ sensing and hyphal development in *Candida albicans*. *PLOS Genet* 2017;13:e1006949. <https://doi.org/10.1371/journal.pgen.1006949>.
46. Vandeputte Patrick, Pradervand Sylvain, Ischer Françoise, Coste Alix T., Ferrari Sélène, Harshman Keith, et al. Identification and functional characterization of Rca1, a transcription factor involved in both antifungal susceptibility and host response in *Candida albicans*. *Eukaryot Cell* 2012;11:916–31. <https://doi.org/10.1128/EC.00134-12>.
47. Vincent BM, Langlois J-B, Srinivas R, Lancaster AK, Scherz-Shouval R, Whitesell L, et al. A fungal-selective cytochrome *bc1* inhibitor impairs virulence and prevents the evolution of drug resistance. *Cell Chem Biol* 2016;23:978–91. <https://doi.org/10.1016/j.chembiol.2016.06.016>.
48. Zhu C, Stolz V, Simonovic N, Al-Rubaye O, Vcelkova T, Moos V, et al. Targeting the catalytic activity of HDAC1 in T cells protects against experimental autoimmune encephalomyelitis. *bioRxiv* 2023:2023–04.

Response to Reviewers

Reviewer #1 (Remarks to the Author):

Authors performed additional experiments and addressed majority of critical comments. I recommend the manuscript for publication in this journal!

Response: We are delighted and appreciate the positive feedback from Reviewer #1.

Reviewer #2 (Remarks to the Author):

I applaud the authors' effort in addressing the reviewers' concerns. I again believe that the finding of CSP contribution to *Candida auris* AMB resistance is a quite exciting as it is different from the known ergosterol biosynthetic pathway. That said, I do not think this discovery will have much impact much on antifungal therapy given the modest contribution of CSP to AMB resistance and the fact that once *C. auris* is disseminated, the high levels of CO₂ would compensate for the loss of CSP. The potential combination therapy of ABA with AMB and also anti-mitochondrial drugs with AMB have been proposed in other studies. It is important for the authors to acknowledge those in the manuscript without diminishing the novelty the findings reported here.

General Response: We really appreciate the enthusiastic feedback by Reviewer #2 on the revision, whose feedback has been critical but constructive and helped to improve our work through the revisions. We apologize for the missing information and lack of clarity noted by Reviewer #2. We wish to emphasize that the requested inclusion of new data into the first revision led to exceeding the word limit past the applicable limit set by the journal. However, the toggling between the revised manuscript and response letter and the efforts to cut the word count might have led to some unintentional omissions. We agree with the reviewer that CSP likely does not play a major role in AMB resistance in invasive candidiasis, which we have been briefly discussed in the revised manuscript. We rephased the relevant part to avoid misunderstandings:

Lines 328-330: By sharp contrast, lack of *RCA1* or *EFG1* or *NCE103* does not cause significant differences in fungal burden recovered from various organs after systemic infections, even when mice were treated with AMB (**Extended Data Fig. 7E, F**).

In fact, this is precisely the reason why we highlighted inositolphosphorylceramide synthase and cytochrome *bc1* as potential antifungal targets. While we have referenced the relevant publication in the revised manuscript before, we have now added a better acknowledgment of this data supporting our findings.

Lines 167-170: The fungal-specific IPC synthase inhibitor aureobasidin A (AbA) enhances the efficacy of AMB against *Cryptococcus neoformans*⁴⁰. AbA similarly enhanced AMB activity in *C. auris*, but it showed no significant differences between WT and CSP mutants (**Extended Data Fig. 4E**).

Lines 206-207: Lansoprazole can synergize with AMB by inhibiting cytochrome *bc1* in *C. auris*⁴³, which is consistent with our data for AR387 and R1, yielding FICI values of 0.625 and 0.75, respectively.

There are few specific concerns after reading the revised version.

1. “One thing that is missing is the experiment to determine if AMB therapy will make a difference in vivo when the Nce103 is blocked.” I was surprised that the authors did systemic candidiasis model to address this concern. This is clearly a wrong animal model to test the idea as authors demonstrated so well that the elevated CO₂ levels in systemic infection would restore the AMB resistance due to lack of NEC103 in vitro. They need to test the AMB therapy in a different animal model, as they have done in Figure 6.

Response: Indeed, we did not perform this experiment in the initial submission, because existing data allowed for a precise prediction of the outcome. Hence, we ultimately decided that the use of the skin *in vivo* models is not needed for several reasons. First, data from human and mouse models already demonstrated a reduced burden of the *nce103*Δ strain after 24-48 hours. Second, in clinical practice, AMB is primarily used for systemic infections through intravenous applications using highly specialized drug formulations, while topical applications are usually not employed (Pappas et al. 2016; Lionakis Michail S. and Chowdhary Anuradha 2024). There are currently no clinical guidelines that recommend the use of AMB for skin disinfection or prevention of fungal colonization. Moreover, numerous effective and cost-efficient disinfectant products are available to reduce or prevent skin colonization by *C. auris* (Rutala et al. 2019; Lionakis Michail S. and Chowdhary Anuradha 2024). Importantly, using AMB as a skin disinfectant not only comes at high cost, but it would also most likely further facilitate the emergence of AMB-resistant mutants (Carolus et al. 2024), especially given the plasticity of *C. auris* as shown in our recent work (Phan-Canh et al. 2025). We believe that the current data presented in the revised manuscript are sufficient to fully support our conclusions, and the handling senior editor has in fact confirmed in the decision letter that such new *in vivo* experimental data are not needed or required.

2. I am hoping that the authors can specifically conclude that CSP’s specific contribution to the intradermal infection is inconclusive. The way it is written is misleadingly vague. The authors have proposed some likely reasons and those should be included in the revised manuscript rather than just in the rebuttal text for the reviewers.

Response: Thank you, and yes, we agree with Reviewer #2. We have rephrased the relevant conclusion as follows:

Lines 247-249: ..., but there were no differences between R1 and the *rca1*Δ or *nce103*Δ mutants (Fig. 5D). Therefore, the CSP does not contribute to fitness when *C. auris* reaches deeper skin compartments as by intradermal infection.

Lines 254-257: After growing for 14 days on mouse skin, *C. auris* was mainly seen around or within hair follicle shafts⁴⁷, where higher CO₂ levels emerging from epithelial metabolism and bacterial skin microbiome components may further support fungal growth⁴⁸.

3. I think it would be useful to include the FDA assay in the manuscript to bolster the conclusion that CSP may affect membrane integrity, particularly given that the authors did not find much changes in ergosterol or other lipids in the lipodomics experiments.

Response: We did not do this owing to the need for brevity, but we included this data in the Extended Data Fig. S8C as requested.

Lines 315-319: Of note, fluoresceine diacetate (FDA)-based assays suggest increased membrane fluidity, confirming protein-independent changes in bilayer permeability²¹. Although we observe increased FDA uptake in some experiments (Extended Data Fig. 8C), these results are variable across independent replicates similar to what we observe for lipodomics data in *nce103*Δ.

4. Although I am not the reviewer 1, I think the authors should include some of the data in the manuscript rather than just presenting in this rebuttal letter for the reviewers here. Examples include Fig. R2, R3.

Response: We have included Fig. R2 into the Extended Fig. S8C. Of note, Fig. R3 has been included in the supplementary material of the first revision.

Reviewer #3 (Remarks to the Author):

I would like to thank the authors for addressing all of my concerns and for their detailed answers. A few very minor points:

Response: Thank you, and we are delighted to receive such positive feedback.

1. I was able to access the PRIDE submission this time and I am satisfied with the shared raw mass spec files and MaxQuant outputs. However, both the zenodo link and the PRIDE submission contain the generic Cassiopeia-LFQ code, not the actual script that was implemented for analysing this particular data set. I'll leave it to the editor to decide if that is acceptable.

Response: Our proteomics collaborator has developed this analysis pipeline as a standardized workflow for automatically handling datasets generated by the proteomics facility. While we are happy to respond to specific questions upon request, we have chosen not to publish individual scripts for each dataset, as doing so in the past led to unnecessary confusion for readers. Instead, we provide clarification as needed to ensure transparency and to avoid misunderstandings. Of note, this standardized data analysis workflow has been applied in multiple publications from the facility (Licheva et al. 2025; Khan et al. 2025; Kavaklioglu et al. 2025), including Nature publications.

2. It was interesting to see the additional lipidomics experiments. They are described very well, although unlike the other omics, these data was not shared.

Response: We now added the lipidomics data into the supplementary data file, thus complying with Nature Microbiology policy (*"Data Availability Is the Golden Rule in Research" 2024*).

Lines 804-805: Lipidomics datasets are provided in supplementary data.

3. What does the blue line in fig. S1F indicate? Could you please write it in the legend?

Response: We apologize for the confusion. Indeed, we use it for indicating a cut-off for gene deletion validation. We removed the blue line to avoid confusion.

I look forward to seeing this paper published.

Response: We thank Reviewer 3 for the formidable feedback and his/her support!

REFERENCES

1. Carolus, Hans, Dimitrios Sofras, Giorgio Boccarella, et al. 2024. "Acquired amphotericin b resistance leads to fitness trade-offs that can be mitigated by compensatory evolution in *candida auris*." *Nature Microbiology* 9 (12): 3304–20. <https://doi.org/10.1038/s41564-024-01854-z>.
2. "Data availability is the golden rule in research." 2024. *Nature Microbiology* 9 (4): 879–879. <https://doi.org/10.1038/s41564-024-01676-z>.
3. Kavaklioglu, Gülnihal, Alexandra Podhornik, Terezia Vcelkova, et al. 2025. "The domesticated transposon protein L1td1 associates with its ancestor L1 ORF1P to promote line-1 retrotransposition." *eLife* 13 (March): RP96850. <https://doi.org/10.7554/eLife.96850>.
4. Khan, Matarr, Marlis Alteneder, Wolfgang Reiter, et al. 2025. "Single-cell and chromatin accessibility profiling reveals regulatory programs of pathogenic TH2 cells in allergic asthma." *Nature Communications* 16 (1): 2565. <https://doi.org/10.1038/s41467-025-57590-3>.
5. Licheva, Mariya, Jeremy Pflaum, Riccardo Babic, et al. 2025. "Phase separation of initiation hubs on cargo is a trigger switch for selective autophagy." *Nature Cell Biology* 27 (2): 283–97. <https://doi.org/10.1038/s41556-024-01572-y>.
6. Lionakis Michail S. and Chowdhary Anuradha. 2024. "Candida auris infections." *New England Journal of Medicine* 391 (20): 1924–35. <https://doi.org/10.1056/NEJMra2402635>.
7. Pappas, Peter G., Carol A. Kauffman, David R. Andes, et al. 2016. "Clinical practice guideline for the management of candidiasis: 2016 update by the infectious diseases society of America." *Clinical Infectious Diseases* 62 (4): e1–50. <https://doi.org/10.1093/cid/civ933>.
8. Phan-Canh, Trinh, Sabrina Jenull, Tamires Bitencourt, et al. 2025. "White-Brown switching controls phenotypic plasticity and virulence of *Candida auris*." *Cell Reports* 44 (7): 115976. <https://doi.org/10.1016/j.celrep.2025.115976>.
9. Rutala, William A., Hajime Kanamori, Maria F. Gergen, Emily E. Sickbert-Bennett, and David J. Weber. 2019. "Susceptibility of *Candida auris* and *Candida albicans* to 21 germicides used in healthcare facilities." *Infection Control & Hospital Epidemiology* 40 (3): 380–82. Cambridge Core. <https://doi.org/10.1017/ice.2019.1>.
10. Williams, Rebecca, Michael P. Philpott, and Terence Kealey. 1993. "Metabolism of freshly isolated human hair follicles capable of hair elongation: a glutaminolytic, aerobic glycolytic tissue." *Journal of Investigative Dermatology* 100 (6): 834–40. <https://doi.org/10.1111/1523-1747.ep12476744>.